# XDomainBench: Diagnosing Reasoning Collapse in High-Dimensional Scientific Knowledge Composition

**Zhiren Gong** [1 2]  **Tiantong Wu** [1]  **Jiaming Zhang** [1]  **Fuyao Zhang** [1]  **Che Wang** [1]  **Yurong Hao** [1]  **Yikun Hou** [1 3]
**Foo Ping** [1]  **Yilei Zhao** [1]  **Fei Huang** [4]  **Chau Yuen** [5]  **Wei Yang Bryan Lim** [1]

## Abstract

Large Language Models (LLMs) are increasingly deployed for knowledge synthesis, yet their capacity for **compositional generalization in scientific knowledge** remains under-characterized. Existing benchmarks primarily focus on single-turn restricted scenarios, failing to capture the capability boundaries exposed by real-world interactive scientific workflows. To address this, we introduce XDOMAINBENCH, a diagnostic benchmark for **interactive interdisciplinary scientific reasoning**. We formalize the composition order and mixture structure to enable systematic stress-testing from single-discipline to inter-disciplinary, comprising 8,598 interactive sessions across 20 domains and 4 task categories, with 8 realistic trajectory patterns covering difficulty and domain-mixture dynamics, simulating real AI4S scenarios. Large-scale evaluation of LLMs reveals a systematic *reasoning collapse* as composition order increases, stemming from two root causes: (i) *direct* difficulty increases induced by domain composition, and (ii) *indirect* interaction-amplified failures where trajectory patterns trigger error accumulation, reasoning breaks, and domain confusion, ultimately leading to session collapse. We have code in GitHub repository, project page in XDomainBench, and dataset in Hugging Face.

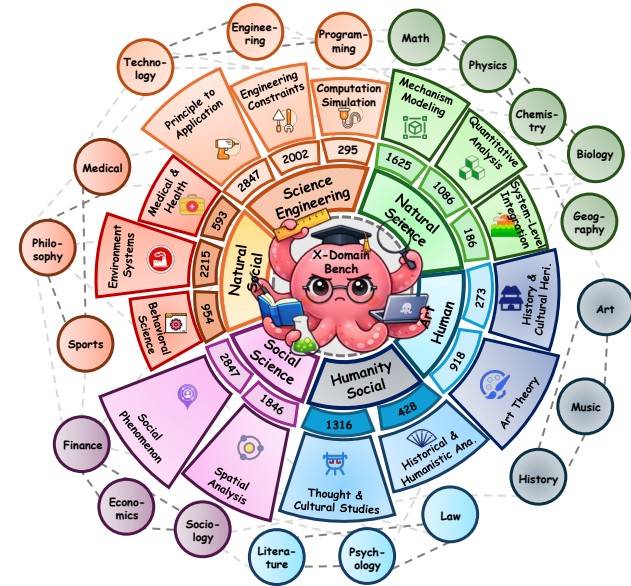

*Figure 1.* **XDOMAINBENCH domain taxonomy and data scale** covering 20 domains organized into 6 major scientific categories.

## 1 Introduction

The emerging paradigm of AI for Science (AI4S) and interdisciplinary studies promises to transform Large Language Models (LLMs) from passive knowledge retrievers into active research assistants capable of cross-domain integration (Zhang et al., 2024; Zheng et al., 2025). In practice, realistic scientific discovery is rarely a stationary inference; it is an inherently interactive, multi-constraint, and interdisciplinary process. For instance, material design typically requires synthesizing constraints from compositional analytical *chemistry*, solid-state *physics*, and *financial* cost assessment; similarly, medicinal analysis often interleaves molecular *biology* with *mathematical* estimation and experimental *engineering* (Zhang et al., 2024; Zheng et al., 2025). Such scenarios fundamentally demand that models possess capabilities **Compositional Generalization in Scientific Knowledge** —*the ability to systematically recombine knowledge from disparate domains to solve complex, interactive problems* (Lake & Baroni, 2018; Keysers et al., 2020; Gong et al., 2026).

---

[1]College of Computing and Data Science, Nanyang Technological University, Singapore [2]Interdisciplinary Graduate Programme, Nanyang Technological University, Singapore [3]Department of Mathematics and Mathematical Statistics, Umeå University, Sweden [4]Alibaba Group, China [5]School of Electrical and Electronic Engineering, Nanyang Technological University, Singapore. Correspondence to: Zhiren Gong <zhiren001@e.ntu.edu.sg>, Jiaming Zhang <jiaming.zhang@ntu.edu.sg>, Wei Yang Bryan Lim <bryan.limwy@ntu.edu.sg>.

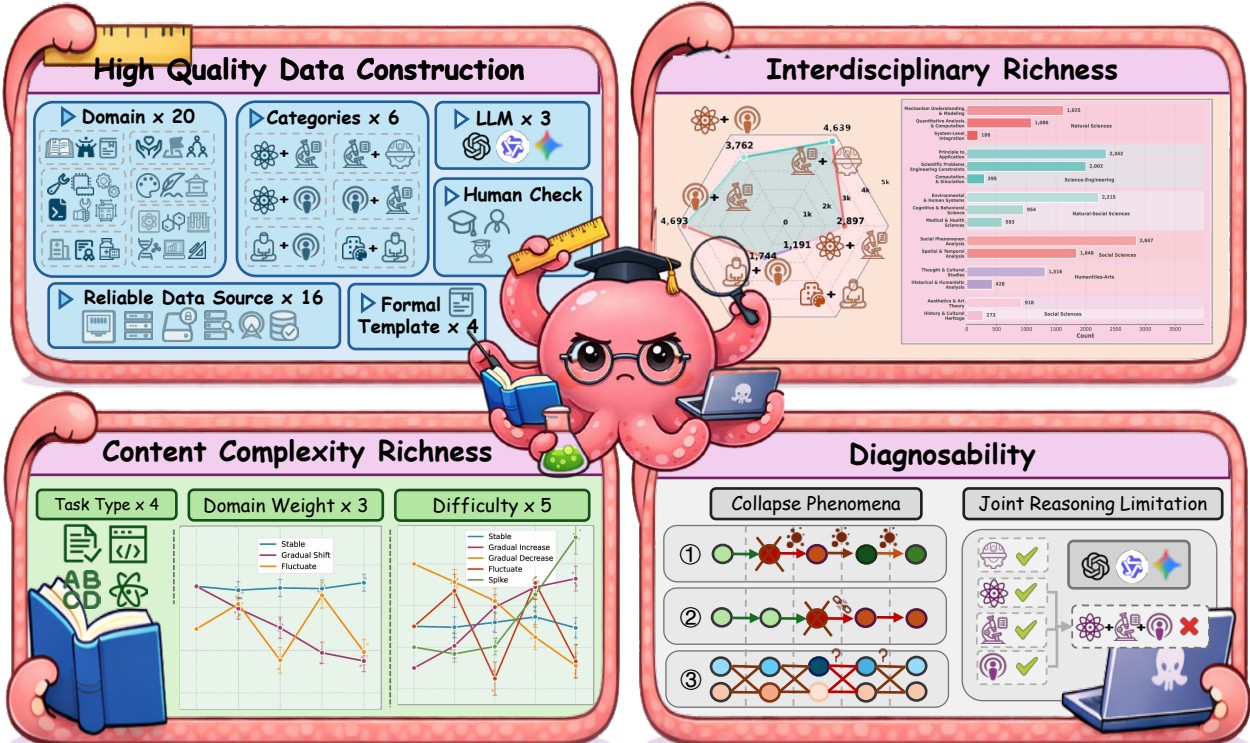

*Figure 2.* **Overview of the XDOMAINBENCH Framework.** The framework generates multi-turn trajectories across 20 domains and 4 task archetypes, annotated with diagnostic signals to evaluate reasoning robustness.

Despite this real-world demand, a critical evaluation gap persists. Current benchmarks struggle to capture the compositional nature of interdisciplinary scientific workflows. Existing evaluations typically focus on static, single-domain reasoning or treat queries as isolated one-shot instances, failing to evaluate the dynamic integration and cross-domain composition required as a research trajectory evolves. Furthermore, they lack a structured construction of interdisciplinary scenarios, making it difficult to diagnose *whether failures stem from missing domain knowledge or limitations in joint reasoning*. Consequently, we lack a systematic understanding of how and why LLMs degrade when multiple scientific constraints collide.

To bridge this gap, we introduce XDOMAINBENCH, a diagnostic framework designed to stress-test compositional generalization in scientific reasoning as shown in Figure 1. We formalize scenario construction through two controllable dimensions: **(i) Composition Order**: the number of distinct domains involved, measuring the *breadth* of integration. **(ii) Mixture Structure**: the logical dependency among domains within a session, measuring the *depth* of joint reasoning. Unlike static datasets, XDOMAINBENCH operates within a standardized interactive controller, transforming reasoning tasks into dynamic trajectories. By annotating each session with **trajectory-level signals** (e.g., difficulty fluctuations and domain-mixture shifts), we can move beyond simple accuracy scores to diagnose exact failure mech-

anisms. In summary, XDOMAINBENCH is characterized by the following key features as shown in Figure 2:

- **Interdisciplinary richness**: We formalize dataset construction through composition order and mixture structure, enabling systematic control over interdisciplinary reasoning complexity and releasing XDOMAINBENCH with **8,598** interactive sessions across **20** domains and **4** task categories.
- **Content complexity richness**: We construct sessions with 8 realistic trajectory patterns covering difficulty and domain-mixture dynamics, approximating AI4S-style interactive reasoning scenarios1 with controlled complexity evolution.
- **Diagnosability**: We provide rich trajectory-level diagnostic annotations linking performance degradation to 3 interpretable failure mechanisms, enabling systematic diagnosis of reasoning collapse.

Our extensive experiments demonstrate a systematic reasoning collapse as composition order increases: accuracy drops from 38.7% to 27.1% as composition order increases, with degradation accelerating non-linearly at higher orders. This collapse stems from two root causes: *(i)* direct complexity overhead by domain composition, and *(ii)* indirect interaction-amplified failures, where trajectory patterns trigger error accumulation, reasoning breaks, and domain confusion, ultimately leading to session collapse.

*Table 1.* **Benchmark positioning and property comparison.**

| Category | Benchmark | Brief description | Sci | Comp | Inter | Turn | Traj |
|---|---|---|---|---|---|---|---|
| **General** | **MMLU** | Broad subject exam-style evaluation (mostly one-shot). | △ | ✗ | ✗ | ✗ | ✗ |
| | **BIG-bench** | Large heterogeneous task suite for capability probing. | △ | ✗ | ✗ | ✗ | ✗ |
| | **HELM** | Holistic evaluation across scenarios and metrics. | △ | ✗ | ✗ | △ | ✗ |
| **CompGen** | **CFQ** | Compositional generalization benchmark (semantic parsing). | ✗ | ✗ | ✗ | △ | ✗ |
| **Long ctx** | **LongBench** | Benchmarking long-context understanding and retrieval. | ✗ | ✗ | ✗ | △ | ✗ |
| **Robust** | **Dynabench** | Dynamic benchmark construction for robustness evaluation. | ✗ | ✗ | ✗ | △ | ✗ |
| **Sci QA** | **ScienceQA** | Multimodal science QA benchmark (static instances). | ✓ | △ | ✗ | ✗ | ✗ |
| | **GPQA** | Expert-level scientific QA (primarily one-shot). | ✓ | ✗ | ✗ | ✗ | ✗ |
| | **SciBench** | Static domain expertise probing (isolated fields). | ✓ | ✗ | ✗ | ✗ | ✗ |
| | **MRMR** | Multidisciplinary multimodal retrieval (23 domains). | ✓ | △ | ✗ | ✗ | ✗ |
| **Diagnostic** | **XDOMAINBENCH** | Interactive interdisciplinary reasoning with knowledge composition. | ✓ | ✓ | ✓ | ✓ | ✓ |

**Notes: Sci**: designed for scientific/AI4S reasoning; **Comp**: supports cross-domain composition *within an instance/session*; **Inter**: session-level interactive evaluation; **Turn**: turn-level diagnostic annotations; **Traj**: trajectory-level diagnostic annotations across turns. △ indicates partial/limited support.

*Table 2.* **Comparison of SciQA benchmarks on dataset scale and composition.**

| Benchmark | #Dom | #Comb | #Theme | Samples | #Type |
|---|---|---|---|---|---|
| **ScienceQA** | 3 | - | 21 | 10.2 | 1 |
| **GPQA** | 3 | - | - | 0.4 | 1 |
| **SciBench** | 3 | - | 9 | 0.7 | 1 |
| **MRMR** | 23 | - | 6 | 1.4 | 1 |
| **OURS** | 20 | 62 | 15 | 8.6 | 4 |

**Notes: #Dom**: number of domains; **#Comb**: number of domain combinations; **#Theme**: number of interdisciplinary themes; **Samples (K)**: sample size in thousands; **#Type**: number of question types. Empty cells indicate not applicable.

## 2 Related Work

### 2.1 Interdisciplinary Benchmarking

As summarized in Table 1, existing benchmarks generally fall into two categories, yet both capture the compositional nature of interdisciplinary scientific workflows. The first category includes **General Evaluations**: MMLU (Hendrycks et al., 2021), BIG-bench (Srivastava et al., 2022), HELM (Liang et al., 2022) and **Specialized Benchmarks**: CFQ (Keysers et al., 2020), LongBench (Bai et al., 2024), Dynabench (Kiela et al., 2021), which typically treat each query as an isolated one-shot instance and not explicit control over *how domains are composed* within a single reasoning process. These circumstances are particularly problematic for AI4S, where *interdisciplinary reasoning* often requires reconciling coupled constraints and paradigm differences *within a single problem-solving process*, which remains largely unformalized as an explicitly controllable evaluation dimension. The second category focuses on **scientific Reasoning Benchmarks**, as presented in Table.2: ScienceQA (Lu et al., 2022), GPQA (Rein et al., 2024), and SciBench (Wang et al., 2024) advance scientific competence by probing deep knowledge in isolated fields ("silos"), but they focus primarily on evaluating static instances and do not test the dynamic integration required in real research. Recent work on multidisciplinary benchmarks, such as MRMR (Zhang et al., 2025), addresses cross-domain challenges in multimodal retrieval across 23 domains, yet focuses on retrieval tasks rather than interactive reasoning, and does not provide trajectory-level diagnostic annotations to understand failure mechanisms. Moreover, existing benchmarks do not characterize how performance degrades as cross-domain compositions become higher-order, making it difficult to diagnose whether failures stem from missing domain knowledge or limitations in joint reasoning.

### 2.2 Interdisciplinary Knowledge Composition

Cross-domain interdisciplinary reasoning can be viewed as *compositional generalization*, where models must recombine learned knowledge under novel compositions rather than merely recall isolated facts (Lake & Baroni, 2018; Keysers et al., 2020). In our setting, *knowledge composition* refers to composing *domain theories and constraints* under a controlled mixture, rather than simply covering multiple subjects. This perspective motivates treating interdisciplinary reasoning as a high-dimensional *composition space* whose complexity can be scaled and measured via composition order and mixture structure. However, this formalization remains absent in existing AI4S-oriented evaluations.

### 2.3 Interactive Reasoning and Diagnostic Evaluation

Scientific workflows are inherently interactive: questions are refined, intermediate results are reused, and reasoning must adapt as new evidence and constraints appear. Long-context evaluations, such as LongBench (Bai et al., 2024), demonstrate that performance can degrade when crucial evidence is embedded in lengthy contexts (Bai et al., 2024; Liu et al., 2024). However, existing benchmarks typically do not provide trajectory-level diagnostic signals (e.g., difficulty fluctuations and domain-mixture shifts) to diagnose *how* and *why* models fail under realistic AI4S workflows. Robustness-oriented evaluation frameworks (Kiela et al., 2021; Ribeiro et al., 2020) highlight that average-case performance scores can obscure systematic failures, yet they lack the composition-space view needed for diagnosing the collapse of interdisciplinary reasoning.

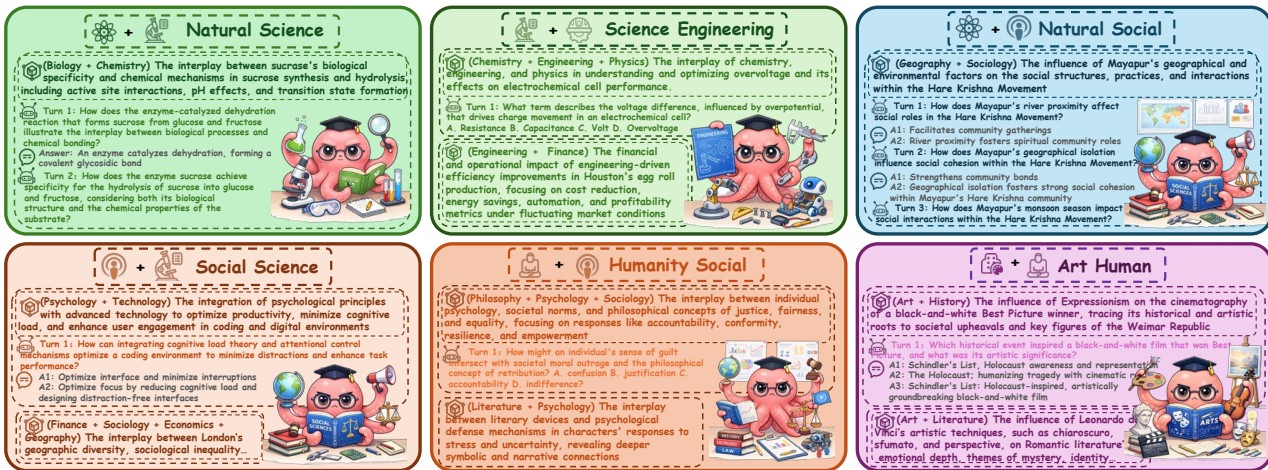

*Figure 3.* **Scientific categories overview of XDOMAINBENCH** covering 6 major interdisciplinary categories, 62 combinations.

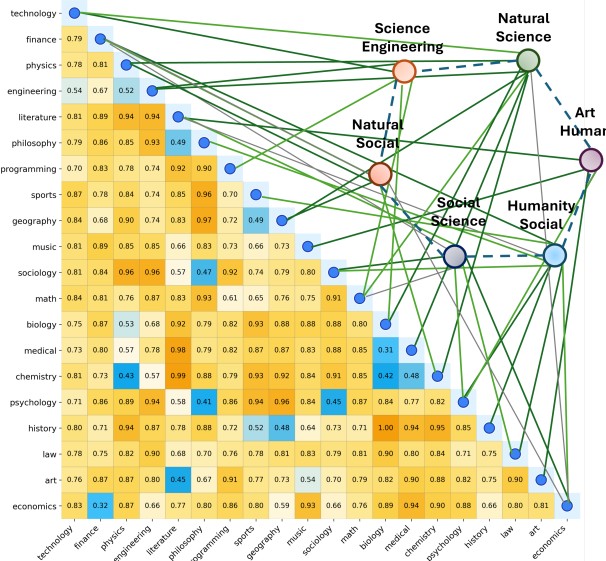

*Figure 4.* **Inter-domain semantic similarity matrix with domain taxonomy** shows cosine similarities between 20 domains, while the network graph categorizes domains into 6 major categories.

## 3 Benchmark Design

### 3.1 Overview and design principles

Realistic AI4S workflows are inherently interactive: models must reason across multiple turns where each turn $x_t$ builds upon previous context through a history mechanism, forming a multi-turn *session* $\{(x_t, y_t)\}_{t=1}^{T}$ where later turns may reuse intermediate results, introduce new constraints, or require reconciliation across domains. Figure 3 summarizes the 6 major interdisciplinary categories of XDOMAIN-BENCH, where the unit of evaluation is a multi-turn session rather than a single query.

Our design is organized around three core principles: (i) **Interdisciplinary richness**: systematic selection and composition of cross-domain scenarios through controlled composition order and mixture structure (Sec. 3.2); (ii) **Content**

complexity richness: controlled difficulty and domain-mixture trajectories with diverse task types to simulate realistic AI4S assistance (Sec. 3.3); (iii) **Diagnosability**: trajectory-level annotations linking performance degradation to interpretable failure mechanisms (Sec. 3.4).

### 3.2 Interdisciplinary richness

We formalize dataset construction through two controllable dimensions: composition order $k$ (how many domains) and domain selection policy (which domains to combine).

**Domain coverage and composition types.** XDOMAIN-BENCH covers 20 scientific domains and constructs six major types of interdisciplinary connections. Complete domain lists and composition selection criteria are in Appendix F.

**Composition order and domain selection.** The composition order $k$ controls the number of distinct domains in a session: $\mathcal{D} = \{d_1, \ldots, d_k\}$. We construct sessions at $k \in \{1, 2, 3, 4\}$, yielding 20 single-domain pools, 24 two-domain combinations, 10 three-domain combinations, and 8 four-domain combinations. To ensure that multi-domain combinations reflect realistic interdisciplinary connections, we select domain sets using embedding-based semantic similarity. For each domain $d$, we compute a domain representation $e(d) = \frac{1}{|\mathcal{S}(d)|} \sum_{x \in \mathcal{S}(d)} \text{Emb}(x)$ by averaging embeddings of normalized prompts/queries from its single-domain seed pool, where $\mathcal{S}(d)$ is the single-domain seed pool and $\text{Emb}(\cdot)$ is a sentence embedding model (Reimers & Gurevych, 2019). We then compute inter-domain distances using cosine similarity and select nearest-neighbor combinations for $k \in \{2, 3, 4\}$, followed by lightweight human sanity checks. Figure 4 visualizes the inter-domain similarity matrix, highlighting the semantic relationships that guide our composition selection.

### 3.3 Content complexity richness

We construct sessions with controlled difficulty and domain-mixture trajectories, where conversations evolve through extension, expansion, and deepening around an interdis-

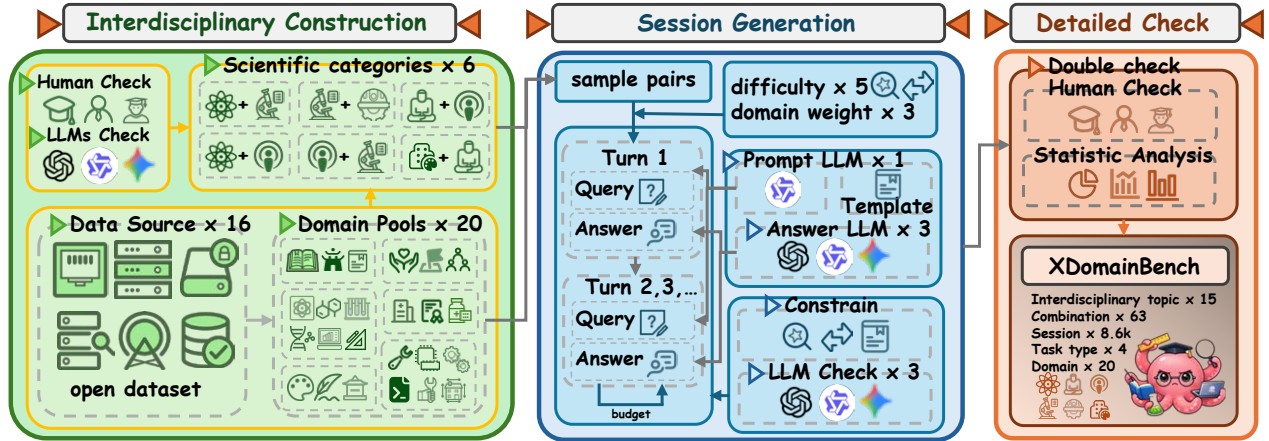

*Figure 5.* **Design framework of XDOMAINBENCH** constructing datasets by controlling composition order and mixture structure, generates interactive multi-turn sessions with trajectory-level diagnostic annotations with detailed human check.

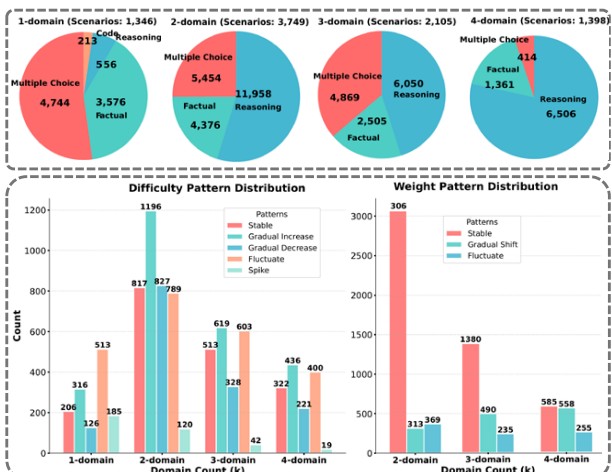

*Figure 6.* **Distribution of different patterns and task tpyes** showing content complexity and diagnosability of XDOMAINBENCH

ciplinary topic, simulating realistic AI4S assistance. The distribution is shown in Figure 6. Detailed definitions and thresholds are in Appendix C. We also validate the reliability $\delta_t$ through detailed human checks and statistical analysis in Appendix B.

### 3.3.1 DIFFICULTY TRAJECTORY AND PATTERNS

Each session is annotated with per-turn *difficulty scores* $\delta_t \in [0,1]$ to capture how the session's difficulty pattern evolves. We estimate $\delta_t$ using a hybrid approach: (i) performance-based signals from 3 fixed pilot models to avoid model bias and (ii) semantic complexity proxies.

Each session is assigned one of five difficulty patterns: STABLE, GRADUAL INCREASE, GRADUAL DECREASE, SPIKE, and FLUCTUATE. Pattern assignment uses interpretable statistics (mean level $\mu_\delta$, volatility $\sigma_\delta$, and trend scores).

### 3.3.2 DOMAIN-MIXTURE TRAJECTORY AND PATTERNS

At each turn $t$, we specify a *domain-mixture distribution* $w_t \in \Delta^{k-1}$ over the selected domains, which governs how strongly the turn should rely on each domain. We

quantify per-turn mixture dispersion using Shannon entropy: $H(w_t) = -\sum_{i=1}^{k} w_{t,i} \log w_{t,i}$. We measure drift between consecutive turns using Jensen–Shannon divergence: $d_t = \text{JSD}(w_t \parallel w_{t+1})$ for $t = 2, \ldots, T$.

Each session is annotated with one of three mixture patterns with the trajectory $w_{1:T}$: STABLE ($w_t \equiv w_0$), GRADUAL SHIFT (interpolation between endpoint mixtures), and FLUCTUATE (bounded random walk on the simplex).

### 3.3.3 TASK TYPES

Sessions incorporate four task types to reflect diverse query forms in real AI4S scenarios: **Multiple Choice** (selection from options), **Factual QA** (direct knowledge recall), **Reasoning** (step-by-step derivation), and **Code** (programmatic solutions). [1] Each task type has a template family and domain-specific seed pools.

### 3.4 Diagnosability

We attribute performance degradation to two root causes: (i) *direct*: insufficient capability for handling complex high-dimensional scientific knowledge composition; (ii) *indirect*: performance degradation induced by session pattern dynamics in multi-turn interactive scenarios.

### 3.4.1 DIRECT MECHANISM: TURN-LEVEL COMPOSITIONAL DIFFICULTY

The direct mechanism manifests as immediate difficulty increases when domain combinations increase, as higher-dimensional compositions raise cognitive load and joint reasoning ability. We diagnose this using the difficulty trajectory $\{\delta_t\}_{t=1}$ with its relationship to composition order $k$, since the initial step questions are directly sampled from single-domain pools during our session construction.

---

[1]The selection of task types is random and depends on the range of task types available for this cross-disciplinary topic. And to prevent bias in model performance evaluation caused by different distributions of task types, we normalize task types in all subsequent average performance evaluations.

*Table 3.* **Overall scaling results as composition order increases** ($k = 1 \to 4$). **Left:** Deterministic problems (R: Recall; F1: F1-score; S@t: SessionSuccess@$\tau$). **Right:** Open-world problems. To prevent bias caused by task type, the data has been normalized.

| Model | Deterministic Problems | | | | | | | | | | | | Open-world Problems | | | | | | | | | | | |
|---|---|---|---|---|---|---|---|---|---|---|---|---|---|---|---|---|---|---|---|---|---|---|---|---|
| | $k=1$ | | | $k=2$ | | | $k=3$ | | | $k=4$ | | | $k=1$ | | | $k=2$ | | | $k=3$ | | | $k=4$ | | |
| | R | F1 | S@t | R | F1 | S@t | R | F1 | S@t | R | F1 | S@t | R | F1 | S@t | R | F1 | S@t | R | F1 | S@t | R | F1 | S@t |
| **Large Models** | | | | | | | | | | | | | | | | | | | | | | | | |
| GPT-5.2 | 30.8 | 22.0 | 42.5 | 25.9 | 11.4 | 28.8 | 22.8 | 8.2 | 25.2 | 18.0 | 4.8 | 26.9 | 25.9 | 9.3 | 29.6 | 21.7 | 7.0 | 28.2 | 28.0 | 5.8 | 21.4 | 13.0 | 5.8 | 35.3 |
| Claude-4.5 Sonnet | 30.4 | 22.8 | 48.5 | 28.6 | 14.0 | 31.3 | 21.0 | 7.3 | 26.7 | 19.3 | 4.5 | 24.4 | 30.2 | 8.0 | 40.7 | 23.4 | 10.3 | 21.4 | 13.5 | 5.5 | 21.4 | 13.3 | 4.0 | 14.7 |
| Claude-4.5 Haiku | 30.6 | 23.0 | 41.8 | 29.7 | 13.7 | 31.1 | 26.9 | 12.3 | 30.7 | 23.0 | 9.5 | 37.7 | 19.4 | 7.0 | 40.7 | 22.4 | 11.0 | 21.4 | 32.6 | 13.8 | 23.2 | 19.1 | 11.5 | 35.3 |
| Gemini-2.5 Flash | 27.1 | 20.4 | 36.6 | 21.4 | 5.3 | 24.3 | 18.4 | 3.0 | 25.2 | 13.8 | 3.1 | 21.3 | 18.8 | 6.9 | 33.3 | 22.1 | 4.1 | 34.2 | 13.5 | 3.3 | 21.4 | 12.4 | 3.7 | 34.5 |
| Gemini-2.0 Flash | 27.2 | 12.9 | 26.9 | 22.4 | 5.6 | 25.8 | 20.7 | 5.0 | 24.3 | 15.0 | 2.8 | 20.3 | 26.8 | 1.8 | 22.2 | 16.2 | 2.8 | 24.8 | 12.6 | 4.6 | 16.1 | 11.9 | 2.3 | 15.4 |
| Qwen2.5-72B | 26.8 | 17.8 | 35.8 | 27.0 | 12.8 | 27.3 | 25.0 | 11.5 | 28.2 | 19.9 | 8.9 | 32.3 | 31.3 | 8.4 | 25.9 | 21.9 | 11.2 | 20.5 | 16.1 | 10.2 | 21.4 | 13.2 | 10.3 | 32.4 |
| **Mean** | 28.8 | 19.8 | **38.7** | 25.8 | 10.5 | **28.1** | 22.5 | 7.9 | **26.7** | 18.2 | 5.6 | 27.1 | 25.4 | 6.9 | 32.1 | 21.3 | 7.7 | **25.1** | 19.4 | 7.2 | **20.8** | 13.8 | 6.3 | 27.9 |
| **Small Models** | | | | | | | | | | | | | | | | | | | | | | | | |
| GPT-5-mini | 23.1 | 12.8 | 25.2 | 25.3 | 14.4 | 21.3 | 22.4 | 2.2 | 20.0 | 11.4 | 1.3 | 20.0 | 22.0 | 2.8 | 10.0 | 23.1 | 3.4 | 20.0 | 3.2 | 3.3 | 10.0 | 3.2 | 3.3 | 10.0 |
| Qwen2.5-14B | 27.2 | 19.1 | 39.6 | 25.5 | 11.3 | 26.6 | 23.3 | 9.1 | 27.7 | 18.5 | 7.6 | 29.9 | 18.2 | 3.9 | 18.5 | 27.1 | 10.4 | 25.6 | 28.1 | 9.3 | 10.7 | 11.6 | 7.5 | 20.6 |
| Qwen2.5-7B | 26.4 | 18.4 | 31.3 | 25.8 | 11.5 | 26.8 | 23.3 | 9.8 | 28.7 | 18.9 | 7.3 | 28.1 | 50.2 | 3.7 | 18.5 | 22.6 | 8.6 | 23.9 | 11.1 | 7.6 | 8.9 | 14.5 | 6.8 | 17.6 |
| Llama-3.1-8B | 26.2 | 18.8 | 35.8 | 24.7 | 11.4 | 23.8 | 23.3 | 9.9 | 28.2 | 16.5 | 8.1 | 25.1 | 15.2 | 4.1 | 14.8 | 23.9 | 9.8 | 29.9 | 28.9 | 10.3 | 17.9 | 19.0 | 7.9 | 29.4 |
| Llama-3.2-3B | 26.8 | 16.3 | 31.3 | 24.5 | 10.5 | 23.8 | 23.2 | 8.7 | 28.7 | 20.9 | 5.5 | 29.9 | 19.6 | 3.4 | 25.9 | 22.8 | 8.9 | 25.6 | 47.1 | 6.9 | 23.2 | 14.3 | 5.1 | 26.5 |
| Gemma-2-2B-IT | 27.3 | 18.6 | 30.6 | 26.3 | 12.1 | 27.6 | 20.6 | 7.4 | 21.8 | 18.1 | 4.3 | 22.6 | 18.4 | 4.6 | 7.4 | 20.7 | 8.5 | 18.8 | 45.0 | 7.4 | 17.9 | 9.5 | 2.8 | 17.6 |
| **Mean** | 26.2 | 17.3 | **32.3** | 25.3 | 11.9 | **25.0** | 22.7 | 7.9 | **25.9** | 17.4 | 5.7 | **25.9** | 23.9 | 3.8 | **15.9** | 23.4 | 8.3 | **24.0** | 27.2 | 7.5 | **14.8** | 12.0 | 5.6 | **20.3** |
| **MoE Models** | | | | | | | | | | | | | | | | | | | | | | | | |
| Qwen3-Next-80B | 63.8 | 65.7 | **89.6** | 46.3 | 40.1 | **51.1** | 41.7 | 36.4 | **52.0** | 29.5 | 26.3 | 34.5 | 10.8 | 15.0 | 40.7 | 29.8 | 27.9 | 28.2 | 26.3 | 23.9 | 46.4 | 25.7 | 24.1 | 41.2 |
| Mixtral-8x7B | 56.4 | 22.4 | 74.6 | 51.9 | 18.0 | **46.4** | 51.0 | 18.9 | 48.5 | 41.0 | 19.5 | 49.4 | 24.1 | 21.3 | 40.7 | 41.9 | 25.7 | 35.9 | 42.0 | 25.5 | 53.6 | 39.9 | 23.7 | 55.9 |
| **Mean** | 60.1 | 44.0 | **82.1** | 49.1 | 29.1 | **48.8** | 46.4 | 27.6 | **50.2** | 35.2 | 22.9 | **42.0** | 17.5 | 18.1 | **40.7** | 35.9 | 26.8 | **32.0** | 34.1 | 24.7 | **50.0** | 32.8 | 23.9 | **48.5** |

### 3.4.2 INDIRECT MECHANISM: SESSION-LEVEL INTERACTIVE DYNAMICS

The indirect mechanism operates through difficulty and domain-mixture dynamics in multi-turn interactive sessions. We formalize this via three interaction-level failure phenomena to mediate how trajectories with different patterns lead to session-level degradation: **Error accumulation** (sustained performance decline after an onset turn), **Reasoning breaks** (sudden accuracy drops indicating model failure to maintain reasoning continuity), and **Domain confusion** (realized domain reliance deviates from intended mixture). Each phenomenon is detected via reproducible heuristics combining standardized signals (Judge scores $J_t$, ParseFail indicators $PF_t$, mixture mismatches $d_t$) with judge-assisted verification when needed. Operational definitions, detector rules, and signature curves are detailed in Appendix C.

### 3.5 Dataset construction pipeline

Following the construction pipeline shown in fig. 5, we summarize the three-stage process:

**Stage 1: Interdisciplinary construction.** We first select a domain set $\mathcal{D} = \{d_1, \ldots, d_k\}$ using embedding-based nearest-neighbor policy (Sec. 3.2). We also use LLM to determine whether the selected domains can form a realistic interdisciplinary scenario, followed by human sanity checks.

**Stage 2: Session generation.** For each session with a fixed topic, we first formalize a target configuration cfg $= (k, P^\delta, P^w)$, where $P^\delta$ and $P^w$ are pattern labels

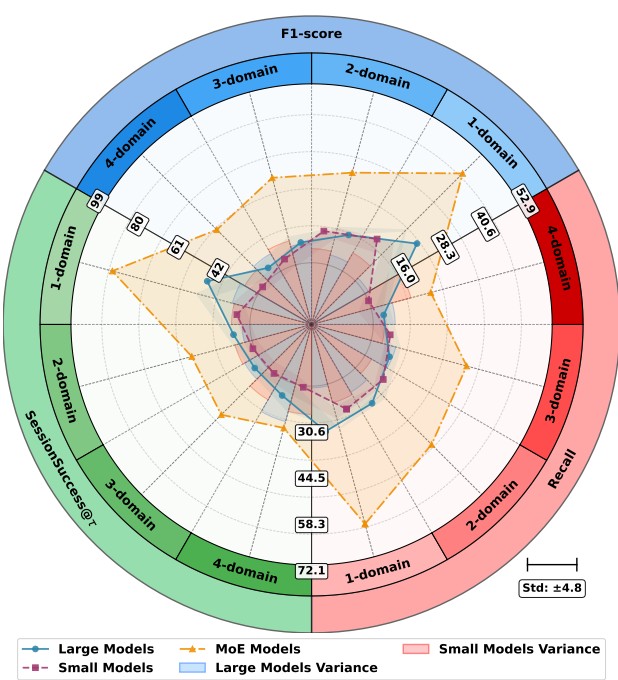

*Figure 7.* **Multi-metric scaling patterns across composition orders.** Variance bands illustrate that small models exhibit greater sensitivity to composition order.

related to difficulty and domain mixture (section 3.3). We then instantiate target diagnostic signals: (i) difficulty trajectory $\{\delta_t\}_{t=1}^T$ with pattern $P^\delta$ and (ii) mixture trajec-

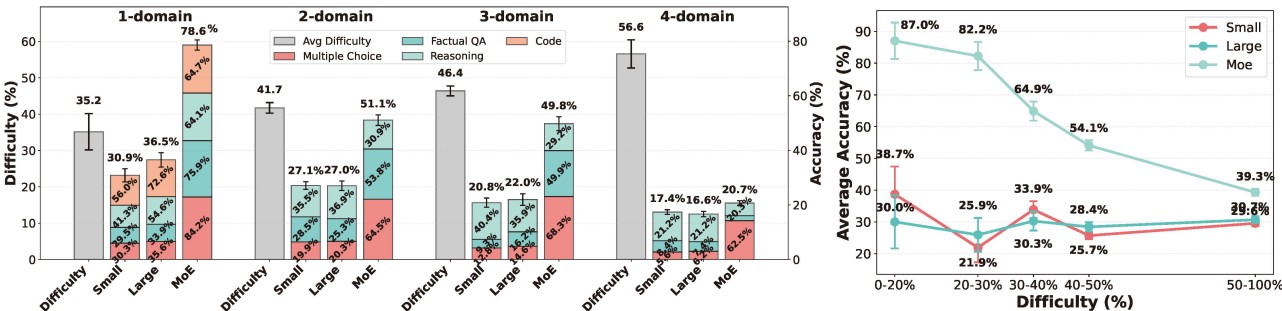

*Figure 8.* Left: **Performance breakdown by composition order and task type at Turn 1.** Each group contains average difficulty and performance for small, large, and MoE models. Right: **Performance vs. difficulty by model type.**

tory $\{w_t^{\text{target}}\}_{t=1}^T$ with regime $P^w$. For turn 1, an LLM generates a cross-domain question $(x_1, y_1)$ conditioned on $(\mathcal{D}, \delta_1, w_1^{\text{target}})$ and construction templates. Three state-of-the-art models independently assess the realized domain weight $\hat{w}_1^{\text{realized}}$ and difficulty $\hat{\delta}_1$. For subsequent turns ($t \geq 2$), generation is conditioned on the previous 1–2 turns to ensure topic coherence, with each turn validated by the same three-model ensemble. If validation fails (e.g., $\hat{w}_t^{\text{realized}}$ deviates from $w_t^{\text{target}}$ beyond tolerance), the turn is regenerated until constraints are met or a retry budget is exhausted. To ensure accuracy, three models provide answers for each turn, while identical answers are merged and samples with large differences in model responses will be deleted.

**Stage 3: Detailed check.** After all turns are generated, we verify that the realized session matches the intended difficulty pattern $P^\delta$ and mixture regime $P^w$. If constraints are violated, we resample affected turns or resample targets, then rerun validation under a bounded session retry budget. Finally, all accepted sessions undergo detailed human quality checks and expert review for critical cases.

**Decoupling construction and evaluation.** Construction-time LLMs are used only to instantiate, validate, and calibrate candidate sessions under fixed diagnostic targets. They do not define the evaluation target and are not used to score the tested models in the default protocol. The evaluated quantity is the model's degradation pattern under controlled composition orders and trajectory regimes, not agreement with the construction models. This separation reduces the risk that benchmark performance merely reflects generator imitation.

**Role of public seed pools.** Public QA sources are used only as seed pools for anchoring feasible domain concepts and supervision formats. They are not directly reused as the evaluated multi-turn sessions. Each released session is newly instantiated through controlled generation and validation, where later turns vary in topic realization, task form, difficulty, and domain-mixture trajectory while remaining coherent under the same interdisciplinary scenario. Thus, XDOMAINBENCH evaluates newly constructed interactive sessions rather than recycled public benchmark items.

## 4 Experiments and Analysis

This section empirically validates the core hypothesis of XDOMAINBENCH: as knowledge compositions increase in order and structural complexity, LLM performance degrades in interactive interdisciplinary scientific reasoning, and the degradation can be diagnosed through trajectory patterns and the failure phenomena they precipitate.

### 4.1 Evaluation setup

**Evaluation suite and interactive protocol.** We evaluate models on a sampled evaluation suite of XDOMAINBENCH episodes, constructed with stratified sampling to ensure coverage across domains, compositions, and trajectory patterns. All experiments use a standardized interactive protocol with a history mechanism: each turn's context includes previous turns, allowing later turns to be influenced by earlier ones. We evaluate diverse LLMs spanning frontier proprietary systems and open-weight models, all queried in a zero-shot setting. Detailed sampling quotas, coverage statistics, and protocol details are in Appendices F, L, and H.

**Metrics.** For **turn-level** correctness, we use **Recall (R)** and **F1-score (F1)**. For **session-level** success, we use **SessionSuccess@$\tau$**, which marks an episode successful if its turn-correctness rate exceeds a fixed threshold $\tau$. Detailed metric definitions, scoring procedures, and implementation details are provided in Appendix J.

### 4.2 Main results: degradation with composition

Table 3 and Figure 7 summarize overall performance as composition order increases from $k = 1$ to $k = 4$.

**Composition order effects.** Across all models, higher-order compositions consistently degrade performance: turn-level accuracy (Recall and F1) and session success rate (SessionSuccess@$\tau$) decline as $k$ increases from 1 to 4. The degradation accelerates non-linearly at higher $k$ values, indicating that cross-domain integrative reasoning becomes increasingly challenging as the composition space dimension grows. Meanwhile, the cognitive load of combining multiple domains compounds rather than accumulates linearly, like large models show moderate drops from $k = 1$ to $k = 2$ but steeper declines at $k = 3$ and $k = 4$. Complete

*Table 4.* Semantic-evaluation robustness under increasing composition order, showing judge-based evaluation preserves the same degradation trend as token-level Recall.

| Metric | $k=1$ | $k=2$ | $k=3$ | $k=4$ | Drop Var. |
|---|---|---|---|---|---|
| Token Recall | 31.3 | 28.0 | 24.1 | 21.9 | 0.021 |
| Judge Acc. | 37.1 | 33.0 | 29.0 | 26.8 | 0.018 |

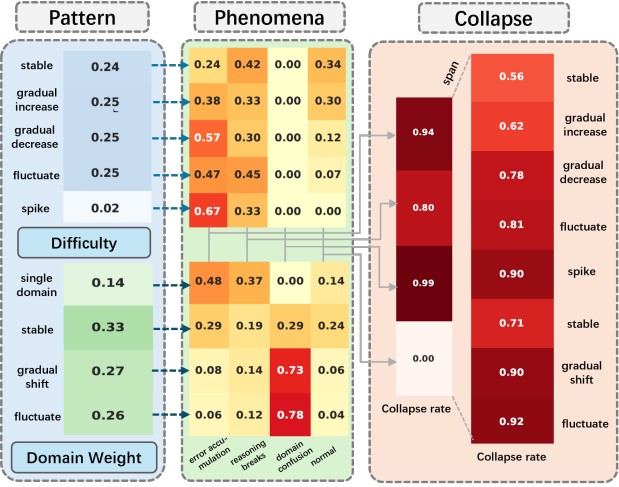

*Figure 9.* **Mechanism summary: pattern distribution, pattern → phenomena, and pattern/phenomena → collapse associations.** All statistics are averaged across all models.

breakdowns by exact domain sets, task types, and trajectory patterns are provided in Appendix N.

**Model-type differences.** Large models maintain relatively stable performance with moderate variance across $k$, while small models exhibit steeper degradation, with variance increasing as $k$ grows. Meanwhile, MoE models demonstrate exceptional performance, consistently outperforming both large and small models across all $k$ values and metrics, with particularly strong performance in SessionSuccess@$\tau$. This pattern indicates that smaller models struggle more with the cognitive load of higher-dimensional domain compositions, while MoE architectures excel at handling complex compositional reasoning tasks.

To verify that the observed collapse is not an artifact of lexical overlap, we evaluate model outputs using a judge-based semantic correctness score. As shown in Table 4, the semantic judge yields higher absolute scores, as expected under a more relaxed criterion, but preserves the same monotonic degradation as the composition order increases. This suggests that token-level Recall provides a stricter measurement of the same underlying failure pattern, rather than manufacturing the collapse through surface-form brittleness.

### 4.3 Diagnostic analysis

The aggregate degradation combines two sources: (i) **direct difficulty increase** induced by domain composition, and (ii) **indirect interaction-amplified** failures arising from particular difficulty and mixture trajectory patterns.

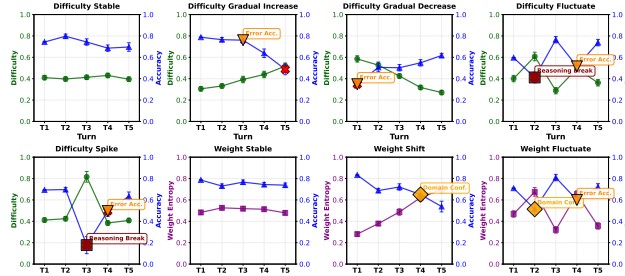

*Figure 10.* **Trajectories of pattern and performance for eight pattern types.** Each panel shows difficulty or weight entropy and accuracy evolution across turns, illustrating how pattern characteristics trigger specific failure phenomena.

#### 4.3.1 DIRECT MECHANISM

The direct mechanism reflects *insufficient capability for handling compositional problems* as domain combinations increase. Following the construction method in Section 3.5, the first turn of each composed episode is instantiated from the same single-domain seed pools as corresponding $k=1$ episodes, enabling direct comparison. For a $k$-domain composition ($k \in \{2,3,4\}$), we compare turn-1 correctness on composed episodes against $k$ single-domain baselines, defining the *composition drop* as the difference between composed turn-1 performance and the mean of constituent baselines. Figure 8 visualizes turn-1 performance and difficulty across composition orders and task types.

Across all models, composing domains induces immediate performance decline, with magnitude increasing as $k$ grows, indicating that combining disciplines imposes an integrative load *prior to* any multi-turn error propagation. The decline is consistent across task types, with difficulty increasing systematically. Detailed turn-1 before/after comparisons for all domain sets are reported in Appendix O.

#### 4.3.2 INDIRECT MECHANISM

The indirect mechanism reflects *performance degradation induced by domain-mixture dynamics in multi-turn interactive scenarios*. We analyze how interactive trajectory patterns in difficulty $\{\delta_t\}$ and domain-mixture $\{w_t\}$ signals indirectly induce failure phenomena, ultimately leading to session-level collapse. We group episodes into representative pattern types using drift and volatility statistics, and examine the chain from trajectory patterns to failure phenomena to session collapse.

Quantificationally, Figure 9 embodies the mechanism across all models: pattern distributions, pattern-to-phenomena associations, phenomena-to-collapse links, and pattern-to-collapse probabilities. The middle columns map patterns to failure phenomena, showing that different patterns trigger different phenomena at varying rates (e.g., gradual decrease patterns strongly associate with error accumulation, while fluctuate patterns trigger both error accumulation and reasoning breaks). The rightmost columns show that all

three failure phenomena lead to high collapse rates, and that pattern-to-collapse probabilities follow the expected trend: stable patterns have lower collapse rates while more volatile patterns (fluctuate, spike) have higher rates.

Qualitatively, Figure 10 illustrates trajectories for each pattern type, showing how difficulty or weight entropy evolves across turns alongside accuracy. These trajectories demonstrate how specific pattern characteristics lead to different failure phenomena: gradual decrease patterns show early high difficulty causing error accumulation, spike patterns show sudden difficulty spikes triggering reasoning breaks that contaminate subsequent turns, and fluctuate patterns exhibit multiple phenomena due to volatility.

**Diagnostic interpretation.** Our analysis should be interpreted as controlled diagnostic attribution rather than a full causal intervention study. The paired turn-1 comparison isolates a direct composition overhead before multi-turn propagation, while the pattern-balanced analyses associate trajectory regimes with recurring failure signatures. Together, these controls support the direct/indirect mechanism interpretation, but we avoid claiming that they exhaust all causal pathways behind model failure.

### 4.4 Implications and Future Directions

Our diagnostic findings reveal fundamental limitations in compositional scientific reasoning, with degradation mechanisms that are both *predictable* (via trajectory patterns) and *addressable* (through targeted interventions). The systematic collapse suggests that scaling alone is insufficient; models need **compositional reasoning architectures** that explicitly handle domain mixtures and maintain coherence across multi-turn interactions.

XDOMAINBENCH enables several promising directions. Our trajectory-level annotations support **pattern-aware training** strategies where models learn to recognize and adapt to difficulty spikes, mixture shifts, and error accumulation signals. The controlled composition space enables **systematic curriculum learning** from low-order ($k = 1, 2$) to high-order ($k = 3, 4$) compositions. MoE models' superior performance suggests **specialized expert routing** for domain mixtures, while the indirect mechanism's prominence motivates **memory-augmented architectures** that prevent error propagation across turns. XDOMAINBENCH further facilitates systematic evaluation and interpretable failure analysis, supporting research into retrieval-augmented generation, few-shot adaptation, and multi-agent collaboration.

### 4.5 Limitations and Scope

XDOMAINBENCH is designed as a diagnostic benchmark rather than a complete simulator of scientific discovery. First, our default protocol is closed-book: models receive the current query and interaction history, but not retrieval tools, external memories, or laboratory feedback. This choice isolates intrinsic multi-turn interdisciplinary reasoning, but

does not measure the full performance of tool-augmented scientific agents. Second, although our construction pipeline separates generation, validation, and evaluation, and includes human auditing and multi-model checks, the benchmark still reflects the coverage and assumptions of its seed pools, templates, and selected domain combinations. Third, our mechanism analysis provides controlled diagnostic attribution, not exhaustive causal intervention: direct composition overhead and interaction-amplified failures explain major observed trends, but other factors such as domain familiarity, prompt sensitivity, and model-specific decoding behavior may also contribute. Finally, some task types, especially code, are less broadly instantiated in the current release. Future versions can extend XDOMAINBENCH toward retrieval-augmented, tool-assisted, multimodal, and larger-scale scientific workflows while preserving the same controllable composition-space design.

## 5 Conclusion

We introduced XDOMAINBENCH, a diagnostic benchmark for interactive interdisciplinary scientific reasoning. By controlling composition order and mixture structure, XDOMAINBENCH reveals consistent and non-linear degradation in performance as domains are increasingly composed. Meanwhile, our trajectory-level annotations enable mechanism-oriented diagnosis, revealing two root causes: (i) *direct* difficulty increases induced by domain composition, and (ii) *indirect* interaction-amplified failures where specific trajectory patterns trigger failure phenomena that lead to session collapse. In summary, XDOMAINBENCH will provide a controllable testbed for developing more robust and adaptive AI4S and interdisciplinary assistants.

## Acknowledgements

This research is supported by the RIE2025 Industry Alignment Fund – Industry Collaboration Projects (IAF-ICP) (Award I2301E0026), administered by A*STAR, as well as supported by Alibaba Group and NTU Singapore through Alibaba-NTU Global e-Sustainability CorpLab (ANGEL). This research is also supported by the Ministry of Education, Singapore, under its Academic Research Fund Tier 2 (Award MOE-T2EP20125-0005). Yikun Hou was supported by the Wallenberg AI, Autonomous Systems, and Software Program (WASP), funded by the Knut and Alice Wallenberg Foundation. Partial computations were enabled by the Berzelius resource provided by the Knut and Alice Wallenberg Foundation at the National Supercomputer Centre.

## Impact Statement

This paper presents work whose goal is to advance the field of Machine Learning. There are many potential societal consequences of our work, none which we feel must be specifically highlighted here.

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

# Appendix Overview

This appendix provides design extensions, construction and quality-control details, evaluation protocol and scoring specifications, as well as extended results and mechanism analyses for XDOMAINBENCH. To keep the appendix navigable, we organize it into four *Parts* according to the paper's pipeline.

**Part A: Dataset Construction & Diagnostic Signals.** This part consolidates all dataset construction details, extending Sections 3.2–3.4 and Section 3.5 with formal definitions, implementation-facing details, and end-to-end construction pipeline. We organize the content into (i) *metric definitions and signal instantiation*, (ii) *validation and quality control*, and (iii) *construction pipeline*.

- Appendix A: Composition metrics and calibration (entropy, drift distances, session-level summaries; includes domain-distance stratification used for nearest-neighbor composition selection).
- Appendix B: Trajectory annotation and instantiation rules (difficulty $\{\delta_t\}$ estimation/normalization; target mixture regimes $\{w_t\}$ specification used during construction).
- Appendix C: Taxonomy of representative trajectory regimes and assignment rules used in mechanism analyses, including hybrid validation for diagnostic signals (LLM rubric + lexical/embedding checks) with tolerance constraints and retry policy.
- Appendix D: Session construction pipeline and coherence/validity filters.

**Part B: Dataset Statistics and Release Artifacts.** This part documents the static dataset characteristics, quality control policies, and release artifacts.

- Appendix E: Curation policies, arbitration, audit procedures, and contamination/leakage safeguards.
- Appendix F: Dataset statistics and coverage across $(k, \text{entropy}, \text{pattern})$.
- Appendix G: Release schema (JSON fields, validators, and directory structure).

**Part C: Standardized Interactive Evaluation Protocol, Scoring, Hyperparameters, and Evaluation Suite.** This part provides the exact controller, prompts, parsing contract, scoring rules, all benchmark hyperparameters, and evaluation suite sampling strategy.

- Appendix H: Reference controller with history mechanism, context injection, and truncation.
- Appendix I: Prompt templates and required output formats by task type.
- Appendix J: Token-level scoring (Recall and F1), judge prompts, calibration/adjudication, and SessionSuccess@$\tau$.
- Appendix K: Model list, versions, inference configuration, retries/timeouts.
- Appendix L: Stratified sampling quotas for the evaluation suite.
- Appendix M: Benchmark hyperparameters (construction/evaluation knobs, validation thresholds, sampling ratios).

**Part D: Extended Results and Mechanism Analyses.** This part deepens the analyses behind the $k$-scaling results by separating (i) global, $k$-level diagnostic shifts, (ii) the controlled *turn-1* direct composition effect (before vs. after), and (iii) session-level results by exact domain sets. We further provide full session-level tables by exact domain sets as *across-model aggregates*.

- Appendix N: Global summary & sanity checks across $k$ and task-type controls.
- Appendix O: Turn-1 direct composition effect under paired controls, including full before/after entry-point tables and difficulty–drop coupling with protocol details.
- Appendix P: Full session-level tables by exact domain sets for $k=1/2/3/4$ as across-model aggregates.

# Part A: Dataset Construction & Diagnostic Signals.

## A    Composition Metrics and Calibration

This section specifies the metrics used to quantify the *mixture structure* and *mixture drift* of the turn-level domain mixture trajectory $\{w_t\}_{t=1}^{T}$, and describes how we select domain combinations for cross-domain compositions.

### A.1    Notation

A session instantiates a domain set $\mathcal{D} = \{d_1, \ldots, d_k\}$ of order $k$ and length $T$. At turn $t$, the intended domain mixture is a probability vector $w_t \in \Delta^{k-1}$, where $w_{t,i} \geq 0$ and $\sum_{i=1}^{k} w_{t,i} = 1$. For specific calculation methods of $w_t$, refer to the mixture trajectory section below.

### A.2    Domain distance

Cross-domain compositions are stratified by *inter-domain distance* to support controlled construction of semantically coherent multi-domain scenarios. Our implementation derives domain representations from the single-domain prompt/query pools, maps them into an embedding space, and selects nearest-neighbor domain combinations for $k \in \{2, 3, 4\}$, followed by a lightweight human sanity check.

**Embedding-based domain representation.**    For each domain $d$, we collect its single-domain prompt/query set and remove generic function words and domain-agnostic boilerplate tokens using a fixed normalization rule. We encode each normalized prompt/query with a sentence embedding model (e.g., Sentence-BERT) (Reimers & Gurevych, 2019). To obtain a robust domain vector, we average embeddings over the single-domain prompt/query set:

$$e(d) = \frac{1}{|\mathcal{S}(d)|} \sum_{x \in \mathcal{S}(d)} \text{Emb}(x). \tag{1}$$

**Distance and nearest-neighbor composition selection.**    We define domain distance by cosine distance:

$$\text{Dist}(d, d') = 1 - \cos(e\,(d)\,, e\,(d'))\,. \tag{2}$$

We compute the full pairwise distance matrix across domains and construct candidate compositions by nearest-neighbor selection: for $k = 2$, we pick top-$M$ nearest neighbors per anchor domain. For $k = 3, 4$, we expand greedily from a near pair by adding domains that minimize the average pairwise distance within the set. This procedure yields semantically coherent compositions and enables controlled stress by distance, which is consistent with prior discussions that larger domain divergence can hinder cross-domain transfer and robustness (Kashyap et al., 2021). Finally, we manually spot-check sampled combinations/sessions to ensure the resulting scenarios are meaningful.

Figure 11 visualizes the inter-domain distance matrix for all 20 domains in our benchmark, reordered by hierarchical clustering to reveal semantically similar domain clusters. This distance matrix guides our nearest-neighbor composition selection strategy, ensuring that cross-domain scenarios are constructed from domains that are semantically coherent yet sufficiently distinct to test compositional reasoning capabilities.

### A.3    Turn-level mixture structure

**Mixture entropy.**    We quantify how "blended" a turn is by Shannon entropy (Shannon, 1948):

$$H(w_t) = -\sum_{i=1}^{k} w_{t,i} \log w_{t,i}. \tag{3}$$

For a fixed $k$, $H(w_t) \in [0, \log k]$. We also report a *normalized* entropy

$$\bar{H}(w_t) = \frac{H(w_t)}{\log k} \in [0, 1], \tag{4}$$

which allows comparisons across different composition orders.

**Effective composition order.**    Entropy can be mapped to an "effective number of active domains" via

$$k_{\text{eff}}(t) = \exp(H\,(w_t)) \in [1, k]. \tag{5}$$

Intuitively, $k_{\text{eff}}(t)$ is close to 1 when the mixture is sharply concentrated and approaches $k$ when the mixture is near-uniform.

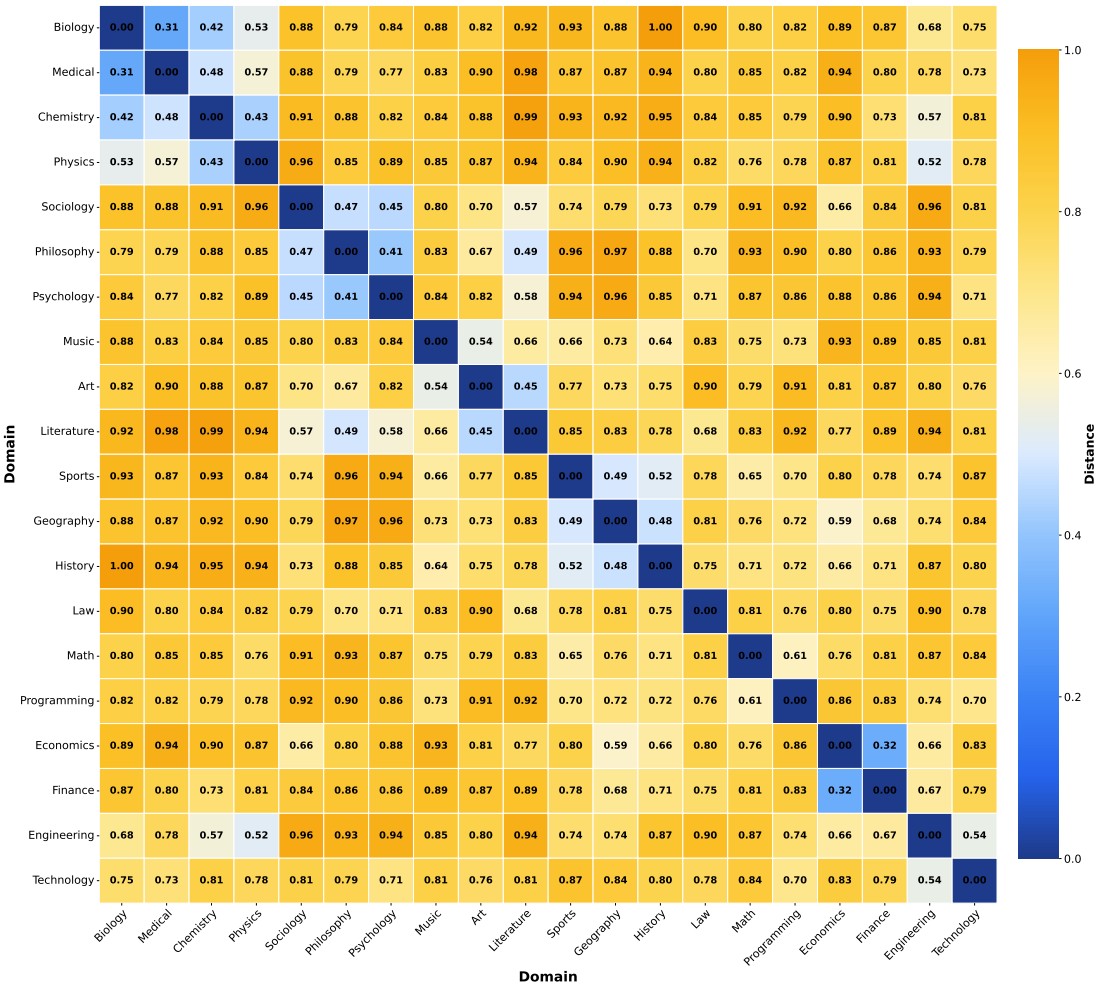

*Figure 11.* Inter-domain distance matrix used for nearest-neighbor composition selection. The matrix shows pairwise distances between 20 domains, reordered by hierarchical clustering to reveal near-domain blocks.

## A.4   turn-to-turn-level mixture drift

To quantify how mixture focus shifts across turns, we compute a symmetric divergence between consecutive mixtures. Let $m_t = \frac{1}{2}(w_{t-1} + w_t)$. The Jensen–Shannon divergence is (Lin, 1991)

$$\text{JSD}(w_{t-1}, w_t) = \frac{1}{2}\text{KL}\left(w_{t-1}\|m_t\right) + \frac{1}{2}\text{KL}\left(w_t\|m_t\right), \tag{6}$$

where $\text{KL}\left(p\|q\right) = \sum_i p_i \log \frac{p_i}{q_i}$. We define the per-turn drift magnitude as

$$d_t = \text{JSD}\left(w_{t-1}, w_t\right), \quad t = 2, \ldots, T. \tag{7}$$

In practice, $\{d_t\}$ captures whether a session maintains a stable mixture, shifts gradually, or fluctuates in domain reliance.

## A.5   Session-level summaries

For each session, we summarize the mixture structure and drift by compact statistics. These summaries are used for (i) reporting and slicing analyses (Section 4.3.2), and (ii) validating that the realized trajectory matches the intended regime.

**Mixture complexity summaries.**   We compute the mean and volatility of normalized entropy:

$$\mu_H = \frac{1}{T} \sum_{t=1}^{T} \bar{H}(w_t), \tag{8}$$

$$\sigma_H = \sqrt{\frac{1}{T} \sum_{t=1}^{T} \left(\bar{H}\left(w_t\right) - \mu_H\right)^2}. \tag{9}$$

**Mixture drift summaries.** We compute the mean and peak drift, plus drift volatility:

$$\mu_d = \frac{1}{T-1} \sum_{t=2}^{T} d_t, \tag{10}$$

$$d_{\max} = \max_{t=2,\ldots,T} d_t, \tag{11}$$

$$\sigma_d = \sqrt{\frac{1}{T-1} \sum_{t=2}^{T} (d_t - \mu_d)^2}. \tag{12}$$

**Regime-oriented indicators.** When we need a simple scalar to reflect "stability vs. fluctuation", we use $\sigma_d$ and $d_{\max}$ as volatility proxies; this aligns with the intuition that high oscillation yields larger turn-to-turn divergence. We do *not* treat these indicators as model features, as they are session diagnostics and construction/validation controls.

## B Trajectory Annotation: Difficulty and Mixture

This section specifies how XDOMAINBENCH constructs two trajectory-level diagnostic signals: the difficulty trajectory $\{\delta_t\}_{t=1}^{T}$ and the domain-mixture (weight) trajectory $\{w_t\}_{t=1}^{T}$. Both are *construction-time metadata* and are defined independently of the evaluated model.

**Design requirements.** We target (i) controllability (pattern instantiation), (ii) comparability (normalized scales), (iii) auditability (rule-based pattern assignment), and (iv) model-agnosticism, consistent with robustness- and diagnosis-oriented evaluation principles (Liang et al., 2022; Kiela et al., 2021; Ribeiro et al., 2020).

### B.1 Difficulty trajectory $\{\delta_t\}$

#### B.1.1 DEFINITION AND DECOMPOSITION

We define a per-turn difficulty score $\delta_t \in [0,1]$ as a convex combination of: (i) a rubric-guided LLM-based difficulty component (70%), and (ii) an embedding-space semantic complexity component (30%) computed after removing generic words.

$$\delta_t = \lambda_{\text{LLM}} \, \delta_t^{\text{LLM}} + (1 - \lambda_{\text{LLM}}) \, \delta_t^{\text{sem}}. \tag{13}$$

#### B.1.2 LLM-BASED DIFFICULTY SCORING

**Three-criterion scoring.** The LLM-based difficulty component is derived from a construction-time rubric score composed of three criteria: ACCURACY, QUALITY, and PERPLEXITY-style fluency/consistency signals (where "perplexity-style" denotes a normalized fluency/consistency proxy scored on a fixed rubric scale, rather than raw LM perplexity). For each judge model $j$ and query $x_t$ of each turn $(x_t, y_t)$, we obtain criterion scores $r_{j,t}^{(\text{acc})}, r_{j,t}^{(\text{qual})}, r_{j,t}^{(\text{ppl})} \in [0,1]$ and combine them with fixed weights:

$$s_{j,t}^{\text{diff}} = \beta_{\text{acc}} \, r_{j,t}^{(\text{acc})} + \beta_{\text{qual}} \, r_{j,t}^{(\text{qual})} + \beta_{\text{ppl}} \, r_{j,t}^{(\text{ppl})}, \qquad \beta_{\text{acc}} + \beta_{\text{qual}} + \beta_{\text{ppl}} = 1. \tag{14}$$

Rubric prompts and score normalization follow LLM-judge best practices and are specified in Appendix J (Zheng et al., 2023; Chiang et al., 2024; Laskar et al., 2024).

**Three state-of-the-art judge ensemble.** To ensure robust and reliable difficulty assessment, we use a judge ensemble $\mathcal{J} = \{j_1, j_2, j_3\}$ consisting of three state-of-the-art large language models and compute:

$$s_t^{\text{diff}} = \sum_{j \in \mathcal{J}} \gamma_j^{\text{diff}} s_{j,t}^{\text{diff}}, \qquad \sum_{j \in \mathcal{J}} \gamma_j^{\text{diff}} = 1, \qquad \mathcal{J} = \{j_1, j_2, j_3\}. \tag{15}$$

This ensembling affects only construction-time metadata and does not depend on the evaluated model.

**From "correctness" to difficulty.** We convert the fused rubric score to difficulty:

$$\delta_t^{\text{LLM}} = 1 - s_t^{\text{diff}}. \tag{16}$$

---

**Algorithm 1** Mixture-trajectory sampler (implementation-agnostic)

---

**Require:** order $k$, length $T$, regime $\mathsf{R} \in \{\texttt{stable}, \texttt{gradual\_shift}, \texttt{fluctuate}\}$; minimum mass $\epsilon_{\min}$; max step change
$\quad\quad \Delta_{\max}$

**Ensure:** mixture trajectory $w_{1:T}$, $w_t \in \Delta^{k-1}$

  1: Sample (or choose) a baseline mixture $\bar{w} \in \Delta^{k-1}$ that matches the target entropy bucket

  2: **if** $\mathsf{R} = \texttt{stable}$ **then**

  3:      **for** $t = 1$ to $T$ **do**

  4:          $w_t \leftarrow \mathrm{Proj}_\Delta(\bar{w} + \xi_t)$                                      $\triangleright \, \xi_t$ small noise

  5:          Enforce $w_{t,i} \geq \epsilon_{\min}$ by truncation+renormalization

  6:      **end for**

  7: **else if** $\mathsf{R} = \texttt{gradual\_shift}$ **then**

  8:      Choose source mixture $w^{(s)}$ and target mixture $w^{(g)}$ (both in the same entropy bucket)

  9:      **for** $t = 1$ to $T$ **do**

10:          $\alpha_t \leftarrow \frac{t-1}{T-1}$

11:          $w_t \leftarrow (1 - \alpha_t) w^{(s)} + \alpha_t w^{(g)}$

12:          Enforce per-step change $\|w_t - w_{t-1}\|_1 \leq \Delta_{\max}$ for $t \geq 2$

13:      **end for**

14: **else if** $\mathsf{R} = \texttt{fluctuate}$ **then**

15:      Choose a baseline $\bar{w}$ and an oscillation amplitude schedule $\eta_{2:T}$

16:      **for** $t = 1$ to $T$ **do**

17:          $w_t \leftarrow \mathrm{Proj}_\Delta(\bar{w} + \eta_t \cdot u_t)$                            $\triangleright \, u_t$ alternates directions

18:          Enforce $\|w_t - w_{t-1}\|_1 \leq \Delta_{\max}$ for $t \geq 2$

19:      **end for**

20: **end if**

---

### B.1.3   EMBEDDING-BASED DIFFICULTY SCORING

We compute $\delta_t^{\mathrm{sem}} \in [0, 1]$ from the semantic complexity of the query $x_t$ of each turn $(x_t, y_t)$, where normalization removes generic function words and domain-agnostic boilerplate using a fixed rule. We embed the normalized text using a sentence embedding model (Reimers & Gurevych, 2019) and derive a scalar complexity proxy (dispersion or uncertainty under a fixed neighborhood statistic).

### B.1.4   DIFFICULTY PATTERNS (5 TYPES)

Each session is assigned one of five difficulty patterns used in mechanism analyses (see Appendix C.2 for the detailed taxonomy and assignment rules): STABLE, GRADUAL INCREASE, GRADUAL DECREASE, SPIKE, FLUCTUATE. Pattern assignment uses interpretable statistics (mean level $\mu_\delta$, volatility $\sigma_\delta$, and a trend score such as Spearman correlation).

### B.2   Mixture trajectory $\{w_t\}$

### B.2.1   CONSTRUCTING MIXTURE TRAJECTORIES

Algorithm 1 shows the reference constructor that generates a target mixture trajectory. Here, $\mathrm{Proj}_\Delta(\cdot)$ denotes Euclidean projection onto the probability simplex to enforce non-negativity and unit sum. In our implementation, this is done deterministically and does not depend on the evaluated model. It is intentionally lightweight: (i) it guarantees $w_t \in \Delta^{k-1}$; (ii) it supports three canonical regimes; and (iii) it exposes only a small number of hyperparameters (step size, noise scale, oscillation period). Note that $w_{1:T}$ is a *benchmark control signal* used to instantiate and diagnose sessions, not a specialized modeling component.

### B.2.2   TARGET MIXTURE REGIMES (3 TYPES)

We instantiate three canonical regimes for the *target* mixture $w_{1:T}$ (see Appendix C.3 for the detailed taxonomy and assignment rules): STABLE, GRADUAL SHIFT, and FLUCTUATE.

**Stable.**    $w_t \equiv w_0$ for $t = 1, \ldots, T$.

**Gradual shift.**    We interpolate between endpoint mixtures $w^{(s)}, w^{(g)} \in \Delta^{k-1}$:

$$w_t = (1 - \lambda_t) \, w^{(s)} + \lambda_t \, w^{(g)}, \lambda_t = \frac{t-1}{T-1}. \tag{17}$$

**Fluctuate.** We generate a bounded random walk on the simplex:

$$w_{t+1} = \text{Proj}_{\Delta}\left(w_t + \eta\,\xi_t\right), \xi_t \sim \mathcal{N}(0, I). \tag{18}$$

### B.2.3  OBSERVED MIXTURE ESTIMATION

To validate that the turn content reflects the intended mixture, we estimate an *observed* mixture $\hat{w}_t$ by combining: (i) an LLM attribution score ($\lambda_{\text{LLM}}^w = 50\%$), (ii) keyword matching ($\lambda_{\text{key}}^w = 35\%$), and (iii) embedding similarity ($\lambda_{\text{emb}}^w = 15\%$):

$$\hat{w}_t = \lambda_{\text{LLM}}^w\,\hat{w}_t^{\text{LLM}} + \lambda_{\text{key}}^w\,\hat{w}_t^{\text{key}} + \lambda_{\text{emb}}^w\,\hat{w}_t^{\text{emb}}. \tag{19}$$

**LLM attribution** For each judge model $j \in \mathcal{J} = \{j_1, j_2, j_3\}$ (the same three state-of-the-art models as used for difficulty assessment), we prompt the judge to output a domain-proportion vector $\hat{w}_{j,t}^{\text{LLM}} \in \Delta^{k-1}$ indicating how much each domain contributes to the knowledge required/used at turn $t$. We then fuse them by fixed weights:

$$\hat{w}_t^{\text{LLM}} = \sum_{j \in \mathcal{J}} \gamma_j^w\,\hat{w}_{j,t}^{\text{LLM}}, \qquad \sum_{j \in \mathcal{J}} \gamma_j^w = 1, \qquad \mathcal{J} = \{j_1, j_2, j_3\}. \tag{20}$$

Rubric prompts, parsing constraints, and judge calibration details are specified in the validation subsection below and in Appendix J.

**Keyword matching and embedding similarity.** $\hat{w}_t^{\text{key}}$ is derived from normalized domain keyword hits, and $\hat{w}_t^{\text{emb}}$ from embedding similarity to domain seed pools. Both are deterministically computed from released resources; details appear in the validation subsection below.

## C  Pattern Taxonomy for Trajectories

This section specifies the *rule-based taxonomy* used to assign each session to (i) one of five **difficulty patterns** and (ii) one of three **mixture regimes**. These labels are used for stratified reporting and mechanism analyses in the main text. We emphasize that pattern labels are derived from the *constructed target trajectories* $\{\delta_t\}_{t=1}^T$ and $\{w_t\}_{t=1}^T$ (defined above) and validated independently (see validation subsection below), following controlled-factor evaluation principles for robust, interpretable diagnosis (Liang et al., 2022; Kiela et al., 2021; Ribeiro et al., 2020).

### C.1  Session-level features used by the taxonomy

**Difficulty-derived features.** Given $\{\delta_t\}_{t=1}^T$, we compute compact descriptors: (i) level $\mu_\delta$ and volatility $\sigma_\delta$; (ii) trend score $\rho_\delta$ (Spearman correlation between $\delta_t$ and $t$); (iii) maximum step change

$$\Delta_\delta^{\max} = \max_{t=2,\ldots,T} |\delta_t - \delta_{t-1}|; \tag{21}$$

and (iv) a spike strength statistic

$$S_\delta = \max_t \left(\delta_t - \text{median}\left(\delta_{1:T}\right)\right). \tag{22}$$

We also record the number of sign changes in first differences $\text{sgn}\left(\delta_t - \delta_{t-1}\right)$ as an oscillation proxy.

**Mixture-derived features.** Mixture dynamics are characterized by the turn-to-turn drift sequence $\{d_t\}_{t=2}^T$ defined above ((7)) via Jensen–Shannon divergence (Lin, 1991), together with the entropy sequence $\{H(w_t)\}$ ((3)) (Shannon, 1948). We reuse the session summaries defined above: $(\mu_d, \sigma_d, d_{\max})$ for drift and $(\mu_H, \sigma_H)$ for mixture structure. The taxonomy below uses *drift summaries* to assign the regime (STABLE, GRADUAL SHIFT, FLUCTUATE). The structure bins, based on entropy, remain defined and reported in Appendix F.

### C.2  Difficulty pattern taxonomy (5 types)

Each session is assigned exactly one label:

$$P^\delta \in \{\text{STABLE, GRADUAL INCREASE, GRADUAL DECREASE, SPIKE, FLUCTUATE}\}.$$

The assignment is intentionally *rule-based* for auditability.

**STABLE.** Low-variance, trend-free difficulty:

$$\sigma_\delta \le \tau_\sigma^\delta \wedge |\rho_\delta| \le \tau_\rho^\delta \wedge \Delta_\delta^{\max} \le \tau_\Delta^\delta. \tag{23}$$

**GRADUAL INCREASE / GRADUAL DECREASE.** Monotonic drift with bounded volatility, operationalized via a trend gate and a step gate:

$$\left(\rho_\delta \geq \tau_{\rho,+}^\delta\right) \text{ or } \left(\rho_\delta \leq -\tau_{\rho,+}^\delta\right), \Delta_\delta^{\max} \leq \tau_{\Delta,\text{grad}}^\delta, \tag{24}$$

together with a mild-volatility constraint $\sigma_\delta \leq \tau_{\sigma,\text{grad}}^\delta$. The sign of $\rho_\delta$ determines INCREASE vs. DECREASE.

**SPIKE.** A localized burst in difficulty:

$$S_\delta \geq \tau_S^\delta \quad \wedge \quad \sigma_{\delta,\neg t^\star} \leq \tau_{\sigma,\text{base}}^\delta, \tag{25}$$

where $t^\star = \arg\max_t \delta_t$ and $\sigma_{\delta,\neg t^\star}$ is the standard deviation computed with the spike turn excluded. This ensures the session is not globally volatile (otherwise it is classified as FLUCTUATE).

**FLUCTUATE.** Sessions not matched by the above but exhibiting high volatility or oscillatory behavior, e.g.,

$$\sigma_\delta > \tau_{\sigma,\text{fluc}}^\delta \quad \text{or} \quad N_{\text{flip}}\left(\delta_{1:T}\right) \geq \tau_{\text{flip}}^\delta, \tag{26}$$

where $N_{\text{flip}}$ counts sign changes of $(\delta_t - \delta_{t-1})$.

**Priority and tie-breaking.** To avoid ambiguity, we apply a fixed priority order: SPIKE $\succ$ GRADUAL $\succ$ STABLE $\succ$ FLUCTUATE. Specifically, we first check the SPIKE gate, then the GRADUAL INCREASE/DECREASE gate, and finally the STABLE gate. If no fire, we assign FLUCTUATE. This ordering matches the intended semantics: a spike is a more specific structure than generic volatility.

### C.3 Mixture regime taxonomy (3 types)

Each session is assigned a mixture regime $P^w \in \{\text{STABLE}, \text{GRADUAL SHIFT}, \text{FLUCTUATE}\}$ based on drift summaries $(\mu_d, \sigma_d, d_{\max})$ defined above.

**STABLE.** Low average drift and low volatility:

$$\mu_d \leq \tau_\mu^w \quad \wedge \quad \sigma_d \leq \tau_\sigma^w. \tag{27}$$

**GRADUAL SHIFT.** Sustained but smooth drift with limited burstiness:

$$\mu_d \geq \tau_{\mu,\text{grad}}^w \wedge \sigma_d \leq \tau_{\sigma,\text{grad}}^w \wedge d_{\max} \leq \tau_{\max,\text{grad}}^w. \tag{28}$$

**FLUCTUATE.** Sessions that violate smoothness via volatile or bursty drift:

$$\sigma_d > \tau_{\sigma,\text{fluc}}^w \quad \text{or} \quad d_{\max} > \tau_{\max,\text{fluc}}^w. \tag{29}$$

It is noted that mixture *structure* (entropy bins) and mixture *dynamics* (regimes above) are treated as orthogonal axes: entropy/structure is defined and bucketed above, while this section only assigns the *dynamic regime* $P^w$.

### C.4 Joint labels, coverage, and released fields

**Joint trajectory type.** For mechanism analysis, we use the Cartesian product label

$$P = \left(P^\delta, P^w\right), \tag{30}$$

yielding $5 \times 3 = 15$ trajectory types. We report the coverage of these types (and entropy bins) in Appendix F and enforce balanced quotas during evaluation-suite sampling.

**Consistency with validation.** Pattern labels are assigned on targets $(\delta_{1:T}, w_{1:T})$, while the validation subsection below validates that realized estimates $(\hat{\delta}_{1:T}, \hat{w}_{1:T})$ remain within tolerance of the targets. Sessions that fail regime confirmation after turn-level validation are regenerated or dropped under bounded budgets (see validation subsection below).

**Released metadata.** For each session we release: (i) per-turn targets $(\delta_t, w_t)$; (ii) pattern labels $(P^\delta, P^w)$ and the joint label $P$; (iii) session summaries $(\mu_\delta, \sigma_\delta, \rho_\delta, S_\delta)$ and $(\mu_d, \sigma_d, d_{\max}, \mu_H, \sigma_H)$; and (iv) validation logs (retry counts and gate statistics), enabling third-party reproduction of pattern-based slicing without re-running construction.

## C.5 Hybrid Validation for Diagnostic Signals

This subsection specifies the *hybrid validation* that checks whether each constructed session follows its intended diagnostic signals: (i) the difficulty trajectory $\{\delta_t\}_{t=1}^T$ and (ii) the domain-mixture trajectory $\{w_t\}_{t=1}^T$. The validator runs during construction and combines heterogeneous signals (LLM rubric, lexical matching, embedding similarity) to mitigate failure modes of any single validator (Zheng et al., 2023; Laskar et al., 2024; Deng et al., 2024; Chiang et al., 2024).

**Turn-level validation gates.** For each turn $t$, we compute realized estimates $(\hat{\delta}_t, \hat{w}_t)$ using the same estimators defined above. A turn is accepted if:

- **Difficulty gate:**

$$\left| \hat{\delta}_t - \delta_t \right| \le \epsilon_\delta \tag{31}$$

  (where $\hat{\delta}_t$ uses the same decomposition and aggregation as defined above).

- **Mixture gate:**

$$d(\hat{w}_t, w_t) \le \epsilon_w \quad \text{for } k \ge 2 \tag{32}$$

  (where $d$ is Jensen–Shannon divergence and $\hat{w}_t$ uses the hybrid estimator defined above).

- **Agreement checks:** Judge disagreement and component-agreement gates ensure stability (enabled when variance is high).

Failures trigger bounded retries under fixed budgets ($R_{\text{turn}}$ per turn).

**Session-level confirmation.** After all turns pass, we confirm that the realized trajectories match the intended difficulty pattern and mixture regime labels using the rule-based statistics and thresholds defined above. Violations trigger resampling/regeneration under a bounded session retry budget ($R_{\text{ep}}$). Candidates that exceed budgets are dropped from the released pool.

**Parsing and retry policy.** LLM judge outputs must follow a strict machine-readable schema (e.g., $k$-dimensional nonnegative vector summing to 1 for mixture attribution). If parsing fails, we apply deterministic repairs (stripping trailing text, clipping negatives, renormalizing). Persistent failures trigger re-judging or turn regeneration.

**Realized mixture alignment.** Although the turn-level mixture gate uses a tolerance threshold for bounded retries, the released sessions are substantially better aligned with their intended targets in practice. Across the released benchmark, the realized mixture error $d(\hat{w}_t, w_t)$ has mean/median/95th-percentile values of $0.118/0.102/0.274$, respectively. Thus, the mechanism analyses are conducted on a pool whose realized mixture trajectories remain close to the intended diagnostic regimes, rather than merely passing a permissive boundary.

# D  Session Construction Pipeline

This section documents the high-level end-to-end *session construction pipeline* of XDomainBench, from seed pools and template families to the layered quality-control (QC) gates used to ensure that each released session is (i) evaluable, (ii) topically coherent across turns, and (iii) consistent with its intended diagnostic configuration. Building on the diagnostic signal definitions and estimators defined above, we focus on the *construction loop and interfaces* here.

## D.1 Construction inputs and reusable resources

**Seed pools (single-domain).** For each domain $d$, we maintain a single-domain seed pool $\mathcal{S}(d)$ of atomic QA items, each associated with a task type (e.g., factual, reasoning, code, multiple-choice) and minimal provenance metadata. Sessions are constructed by selecting one or more seeds and expanding them into multi-turn trajectories under controlled diagnostic targets.

**Template families (by task type).** We use a small number of prompt templates per task type to standardize generation outputs (question format, reference answer format, and required fields). Full templates and output contracts are provided in Appendix I. This separation keeps construction controllable while allowing the evaluation controller (Appendix H) to remain stable across benchmark versions.

**Domain lexicons and embeddings.** Construction-time validators rely on (i) a domain keyword lexicon $\mathcal{K}(d)$ per domain and (ii) a domain embedding representation (centroid) computed from single-domain seed pools. These reusable resources are also used by the domain-mixture validation module and by the domain-distance stratification.

**Construction-time judges (three state-of-the-art models).** Some QC components require rubric-guided LLM judgments (e.g., difficulty rubric scoring and mixture attribution). To ensure robust and reliable assessment, we use a fixed ensemble of three state-of-the-art large language models and fuse their scores with fixed weights. Importantly, these judges are used *only* during construction/validation and are decoupled from the evaluated models.

### D.2 Session specification as a *construction request*

A session is instantiated from a configuration tuple

$$\mathsf{cfg} = \left(k,\ \mathcal{D},\ T,\ \tau_{1:T},\ P^\delta,\ P^w,\ w_{1:T}\right), \tag{33}$$

where $\mathcal{D} = \{d_1, \ldots, d_k\}$ is the domain set (order $k$), $T$ is the turn budget, $\tau_{1:T}$ is a task-type schedule, $P^\delta$ is a difficulty-drift pattern label, and $P^w$ together with $w_{1:T}$ specifies the target mixture regime/trajectory. The benchmark-wide *sampling quotas* over these configurations (coverage across $k$, entropy/drift bins, pattern labels, etc.) are described in Appendix L, and the resulting empirical coverage is reported in Appendix F.

### D.3 Single-domain session builder ($k = 1$)

Single-domain sessions are generated by iteratively expanding a seed into a multi-turn trajectory. At a high level:

1. **Seed selection.** Sample a domain $d$ and a seed $(x_1, y_1) \in \mathcal{S}(d)$ consistent with the desired task type $\tau_1$.

2. **Turn expansion.** For $t = 2, \ldots, T$, generate $(x_t, y_t)$ conditioned on prior turns, the desired task type $\tau_t$, and the target difficulty control for turn $t$.

3. **Step-wise QC.** After each generated turn, apply the *format gate*, *topic-coherence gate* (Section D.5), and *difficulty-consistency gate*. Only accepted turns are appended to the history.

This strict step-wise gating prevents error accumulation during construction and ensures that the resulting session remains evaluable and coherent before advancing to later turns.

### D.4 Cross-domain session builder ($k \in \{2, 3, 4\}$)

Cross-domain sessions require controlling the per-turn mixture trajectory $w_{1:T}$ and ensuring that the generated turns genuinely demand knowledge from intended domains, proportionally consistent with $w_t$.

**Domain combination selection.** We first select a domain set $\mathcal{D}$ using embedding-based semantic similarity over single-domain pools, while maintaining diversity via a *used-combinations* set that prevents over-reusing the same domain tuples. (Details of domain representations and distances appear in Appendix A.2.)

**Mixture trajectory instantiation.** Given $\mathcal{D}$, we sample a target mixture regime and instantiate $w_{1:T}$. The generation prompt at each turn explicitly includes the intended domain mixture $w_t$ and the intended difficulty target, and asks the generator to produce $(x_t, y_t)$ whose required knowledge matches these controls.

**Cross-domain QC.** Each turn is validated by (i) the same *format* and *topic-coherence* gates as in the single-domain case, and additionally (ii) a *mixture-consistency* gate that estimates an observed mixture $\hat{w}_t$ from the realized turn and checks it against $w_t$ within tolerance. At the session level, we verify that the realized drift regime statistics match the intended regime before acceptance.

### D.5 Topic coherence gate

Beyond diagnostic signals, we enforce a *topic-coherence* constraint to ensure that turns within a session form a connected conversation rather than a bag of independent QA items. Given current text $t$ (typically the current query, including the reference answer) and a set of previous turn texts $T_{\mathrm{prev}}$, we compute a coherence score

$$s_{\mathrm{topic}} = 0.7\, s_{\mathrm{emb}} + 0.3\, s_{\mathrm{kw}}, \tag{34}$$

where $s_{\mathrm{emb}}$ is the maximum embedding cosine similarity between $t$ and any $t' \in T_{\mathrm{prev}}$ (after stopword removal), and $s_{\mathrm{kw}}$ is the Jaccard overlap between extracted keyword sets.

**Rationale.** This hybrid gate is designed to detect abrupt topic shifts (low lexical overlap) while remaining robust to surface paraphrases (high semantic similarity), providing a lightweight but effective coherence control.

### D.6 Quality assurance and error handling

We employ a layered QC stack during construction. The full auditing and filtering policy is detailed in Appendix E.3.

---

**Algorithm 2** Full pipeline for XDOMAINBENCH: construction → validation → evaluation → analysis.

---

**Require:** Domain library and single-domain pools (Part B); quota/slice specification (Appendix L); hyperparameters/tolerances and retry budgets; hybrid diagnostic validators; controller, prompts, and scoring rules (Part C); analysis specifications (Part D).

**Ensure:** Released evaluation suite with metadata and logs, plus evaluation/analysis outputs.

    **Phase 0: Preprocessing (construction-time utilities).**

1: Build domain representations from single-domain pools and compute the inter-domain distance matrix.

2: Build/normalize released lexical resources (e.g., domain keywords) and embedding indices used by validators.

    **Phase 1: Construction → validation (closed loop under quotas).**

3: **while** evaluation suite not filled under quota constraints **do**

4:     Sample a target slice cfg under quota constraints (e.g., $k$, entropy/drift bins, difficulty pattern, mixture regime, task-type schedule).

5:     Select a domain set $\mathcal{D}$ of order $k$ (nearest-neighbor policy).

6:     Instantiate target diagnostic signals for this session: (i) target difficulty $\delta_{1:T}$ and its pattern label; and (ii) target mixture $w_{1:T}$ and its regime label.

7:     Initialize retry counters: $r_{\mathrm{ep}} \leftarrow 0$.

    **Turn loop (generate–validate–retry).**

8:     **for** $t = 1$ **to** $T$ **do**

9:         $r_{\mathrm{turn}} \leftarrow 0$.

10:         **repeat**

11:             Generate a candidate turn instance (query, reference, required fields) conditioned on $(\mathcal{D}, \delta_t, w_t)$ and construction templates.

12:             Apply **hard-format/evaluability gates** (schema/contract; required fields); reject immediately if violated.

13:             Compute realized diagnostics $(\hat{\delta}_t, \hat{w}_t)$ and apply **turn-level hybrid validation**: (i) difficulty gate w.r.t. $\delta_t$; (ii) mixture gate w.r.t. $w_t$ (only if $k \geq 2$); and (iii) parsing/consistency handling and tolerance checks.

14:             **if** any turn-level gate fails **then**

15:                 $r_{\mathrm{turn}} \leftarrow r_{\mathrm{turn}} + 1$.

16:             **end if**

17:         **until** all turn-level gates pass **or** $r_{\mathrm{turn}} > R_{\mathrm{turn}}$

18:         **if** $r_{\mathrm{turn}} > R_{\mathrm{turn}}$ **then**

19:             **Reject** this session candidate; **continue** the outer **while** loop.

20:         **end if**

21:     **end for**

    **Session-level confirmation and QC.**

22:     Compute session summaries (entropy/drift; difficulty statistics) and confirm that: (i) the realized session matches the intended difficulty pattern, (ii) the realized session matches the intended mixture regime.

23:     **while** session-level constraints violated **and** $r_{\mathrm{ep}} < R_{\mathrm{ep}}$ **do**

24:         $r_{\mathrm{ep}} \leftarrow r_{\mathrm{ep}} + 1$.

25:         Resample affected turns or resample targets, then rerun the turn and session confirmation.

26:     **end while**

27:     **if** session-level constraints still violated **then**

28:         **Reject** this session candidate; **continue** the outer **while** loop.

29:     **end if**

30:     Apply additional dataset QC filters and audits; drop if any hard constraint fails.

31:     Commit the session to the released pool.

32: **end while**

    **Phase 2: Standardized interactive evaluation (consume released suite).**

33: **for** each evaluated model $m$ (Appendix K) **do**

34:     **for** each session $e$ in the released suite **do**

35:         Run the reference controller with the required prompt templates (Appendices H and I).

36:         Parse outputs per contract.

37:         Score per turn and per session (Appendices J and K).

38:     **end for**

39: **end for**

    **Phase 3: Aggregation and analyses (reporting + mechanisms).**

40: Produce full result tables and slices (Appendices N, O, and O.2).

41: Run mechanism analyses using released signals and labels (Appendices C and N).

42: **return** Release artifacts (suite + metadata) and analysis outputs.

---

# Part B: Dataset Statistics and Release Artifacts.

## E   Data Quality, Auditing, and Contamination Controls

This section documents the *quality control (QC)* and *risk control* policies used during XDOMAINBENCH construction, focusing on (i) *evaluability* and formatting correctness, (ii) *auditing and arbitration* for ambiguous or unstable cases, and (iii) *contamination/leakage* safeguards. Based on the definitions of diagnostic signals and their hybrid validators defined above, we specify how those components are *operationalized* in dataset curation. The closed-loop construction/validation procedure and retry budgets follow Algorithm 2.

### E.1   Design goals and threat model

**Evaluability.**   Every released turn must admit a well-defined scoring procedure under the standardized protocol (Part C), meaning that the prompt is self-contained, the expected output format is deterministic, and reference supervision (gold/judge rubric/unit-testable contract) is available.

**Controllability.**   Released sessions must respect the intended drift controls (e.g., difficulty patterns and mixture regimes) and remain topically coherent across turns. We enforce these via *gates* and *retry budgets* in the closed-loop pipeline (Algorithm 2).

### E.2   Evaluability criteria and hard filters

We apply *hard filters* that a candidate's turn/session must satisfy before entering the released pool. These checks are **model-agnostic** and depend only on the constructed artifact.

**Schema and formatting conformance.**   Each turn must satisfy the prompt/output schema required by its task type (e.g., a single option letter for multiple-choice; executable code block where applicable; or a single short factual span), as defined by the prompt templates. Violations are treated as HARDFAIL and trigger regeneration under the per-turn retry budget.

**Self-contained prompts.**   A turn is rejected if it relies on external hyperlinks, hidden files, or unstated private context.

**Unambiguous supervision.**   We reject turns that do not admit a stable supervision signal, including: (i) multiple-choice items with multiple plausible correct options, (ii) factual questions whose answer is underspecified by the prompt, (iii) code tasks with missing I/O contract or unverifiable behavior.

**Content safety and policy compliance.**   We remove unsafe or disallowed content in accordance with standard dataset release norms (e.g., instructions for wrongdoing, explicit personal data). These are treated as HARDFAIL and are never regenerated from the same seed.

### E.3   Layered quality assurance and error handling

QC is executed in a layered manner, mirroring the construction loop described above. We clarify *where* each check is applied and *what* is logged for auditability, without repeating the validation logic already specified in Part A.

**Turn-level validation (commit gate).**   The turn-level validation gates (format correctness, topic coherence, difficulty consistency, and mixture consistency) are applied as described above. For bookkeeping, each rejection is tagged with a single *primary failure reason* drawn from {FORMATFAIL, PARSEFAIL, DIFFGATEFAIL, MIXGATEFAIL, COHERENCEFAIL, SAFETYFAIL} to support later QC reporting.

**Session-level validation (pattern/regime confirmation).**   The session-level validation (pattern/regime confirmation and coherence checks) is applied as described above. Violations trigger bounded session-level repair (resample affected turns or targets) up to $R_{\text{ep}}$.

**Post-hoc cleaning and batch health checks.**   After a batch of sessions is produced, we run deterministic cleaning to remove construction artifacts (e.g., placeholder tokens, accidental metadata tags, spurious answer prefixes), followed by distributional sanity checks over core metadata (e.g., length, pattern counts, entropy bins; see below for reported summaries). Degenerate clusters (e.g., unusually repetitive outputs or abnormal failure-reason spikes) are flagged for audit sampling (see below).

### E.4   Arbitration and judge agreement for curation

Some QC decisions consume construction-time LLM judgments (e.g., difficulty rubric components and mixture attribution). We do *not* restate how dispersion/agreement is computed; we only specify the *curation actions* taken when those signals indicate instability.

*Table 5.* Turn-level statistics by task type and domain order.

| Task Type | 1-domain | 2-domain | 3-domain | 4-domain | Total | Avg Q Len | Avg A Len |
|---|---|---|---|---|---|---|---|
| Factual | 5,534 | 8,273 | 4,715 | 1,531 | 20,053 | 19.4 | 8.7 |
| Reasoning | 769 | 14,639 | 6,080 | 8,360 | 29,848 | 31.2 | 21.5 |
| Code | 553 | - | - | - | 553 | 27.0 | 43.7 |
| Multiple Choice | 2,856 | 4,302 | 5,550 | 1,728 | 14,436 | 30.8 | 12.1 |
| Total | 9,712 | 27,214 | 16,345 | 11,619 | 52,582 | 26.9 | 15.5 |

**When is arbitration triggered?** A candidate is marked UNSTABLE when the agreement/dispersion gates defined above fail, or when cross-signal consistency gates indicate a contradiction between LLM attribution and deterministic evidence. By default, unstable items are *regenerated* under the standard retry policy (Algorithm 2). Arbitration is invoked primarily for (i) quota-critical slices, or (ii) systematic instability patterns discovered in batch monitoring.

**Arbitration actions.** Given a quota-critical UNSTABLE candidate, we apply:

1. **Re-judge (same rubric, stricter schema).** Re-run the same judging rubric but enforce a shorter, schema-only output (no free-form rationale), to reduce parsing/verbosity variance.

2. **Regenerate (preferred).** If the case is near acceptance boundaries or exhibits semantic misalignment, regenerate the turn/session and re-apply the standard validators (Algorithm 2).

3. **Manual audit escalation (rare).** Escalate only when repeated regeneration fails and the slice is still quota-critical, or when a systematic issue is suspected (see below).

### E.5 Audit sampling

**Human audit procedure.** We conducted a comprehensive human audit of the dataset to ensure quality and identify systematic issues. Six PhD students from diverse academic backgrounds (covering the domains represented in the benchmark) performed manual inspection of 50% of the dataset content, checking each sample individually for: (i) coherence and realism, (ii) supervision validity, and (iii) alignment with intended drift patterns. This audit identified problematic samples at a rate of 0.4%. Based on the patterns observed in these problematic samples, we then applied targeted screening criteria to the remaining 50% of the data, enabling efficient identification and removal of similar issues.

**Audit checklist.** The audit used a fixed checklist with pass/fail fields: format compliance, unambiguous answerability, no hidden dependencies, no leakage-like memorized strings, and pattern plausibility.

### E.6 Contamination and leakage safeguards

We implement a lightweight but conservative set of safeguards to reduce contamination risks.

**Deduplication within the benchmark.** We remove exact duplicates by hashing normalized prompts and answers. We also perform near-duplicate detection using embedding similarity on normalized text, dropping items with a similarity above the threshold to prevent inflated coverage.

**Seed isolation and split hygiene.** We maintain disjoint seed pools for construction vs. evaluation suites and enforce that no session shares the same normalized prompt across splits. All split assignments are deterministic and keyed by stable IDs to avoid accidental drift across releases.

## F Dataset Statistics and Coverage

This section summarizes the scale and slice coverage of the released benchmark, complementing the construction pipeline defined above, the QC and auditing policy, and the released data schema (see below). We report (i) the underlying single-turn QA pools used for session construction, (ii) scenario-level coverage over domain order and drift factors, and (iii) turn-level composition statistics.

### F.1 Overall scale

XDOMAINBENCH contains **8,598** multi-turn scenarios with a total of **52,582** turns (QA pairs). Turn-level counts and average lengths are summarized in Table 5. Scenario counts by composition order and task type are reported in Table 7. Code turns are instantiated only in 1-domain settings in the current release.

*Table 6.* Data Sources (Single-turn Pool Sizes) by Domain

| Category | Domain | Total | Data Sources (Sample Count) | License/ Notes |
|---|---|---|---|---|
| **Arts** | Art | 572 | RACE (218), TriviaQA (354) | RACE: non-commercial research only; TriviaQA: Apache-2.0 |
| | Music | 1105 | RACE (247), TriviaQA (858) | RACE: non-commercial research only; TriviaQA: Apache-2.0 |
| **Humanities** | History | 459 | MMLU (48), TriviaQA (411) | MMLU: MIT; TriviaQA: Apache-2.0 |
| | Literature | 311 | MMLU (2), RACE (309) | MMLU: MIT; RACE: non-commercial research only |
| | Philosophy | 417 | CommonsenseQA (114), MMLU (131), OpenBookQA (30), TriviaQA (142) | CSQA/MMLU: MIT; OpenBookQA: Apache-2.0 (repo); TriviaQA: Apache-2.0 |
| **Social Sci.** | Economics | 481 | SciQ (1), TriviaQA (480) | SciQ: CC BY-NC 3.0; TriviaQA: Apache-2.0 |
| | Finance | 822 | CommonsenseQA (2), MMLU (1), OpenBookQA (7), RACE (26), SciQ (251), TriviaQA (535) | CSQA/MMLU: MIT; OpenBookQA: Apache-2.0 (repo); RACE: non-commercial; SciQ: CC BY-NC 3.0; TriviaQA: Apache-2.0 |
| | Geography | 2070 | SciQ (56), TriviaQA (2014) | SciQ: CC BY-NC 3.0; TriviaQA: Apache-2.0 |
| | Law | 713 | LexGLUE (ECtHR) (516), MMLU (197) | ECtHR (LexGLUE subset): CC BY-NC-SA 4.0; MMLU: MIT |
| | Psychology | 497 | CommonsenseQA (372), MMLU (72), OpenBookQA (53) | CSQA/MMLU: MIT; OpenBookQA: Apache-2.0 (repo) |
| | Sociology | 436 | CommonsenseQA (136), MMLU (15), OpenBookQA (19), TriviaQA (266) | CSQA/MMLU: MIT; OpenBookQA: Apache-2.0 (repo); TriviaQA: Apache-2.0 |
| **STEM** | Biology | 958 | MMLU (47), OpenBookQA (765), SciQ (146) | MMLU: MIT; OpenBookQA: Apache-2.0 (repo); SciQ: CC BY-NC 3.0 |
| | Chemistry | 890 | OpenBookQA (85), SciQ (805) | OpenBookQA: Apache-2.0 (repo); SciQ: CC BY-NC 3.0 |
| | Engineering | 914 | ARC Challenge (15), SciQ (205), TriviaQA (694) | ARC: CC BY-SA; SciQ: CC BY-NC 3.0; TriviaQA: Apache-2.0 |
| | Mathematics | 675 | GSM8K (603), MMLU (72) | GSM8K: MIT; MMLU: MIT |
| | Medicine | 436 | MMLU (209), MedMCQA (187), SciQ (40) | MMLU: MIT; MedMCQA: MIT; SciQ: CC BY-NC 3.0 |
| | Physics | 561 | ARC Challenge (86), OpenBookQA (153), SciQ (322) | ARC: CC BY-SA; OpenBookQA: Apache-2.0 (repo); SciQ: CC BY-NC 3.0 |
| | Computer Sci. | 395 | Custom / Internal (395) | Internal/custom (not redistributed) |
| | Technology | 947 | CommonsenseQA (866), RACE (55), SciQ (4), TriviaQA (22) | CSQA: MIT; RACE: non-commercial; SciQ: CC BY-NC 3.0; TriviaQA: Apache-2.0 |
| **General** | Sports | 1164 | TriviaQA (1164) | TriviaQA: Apache-2.0 |

## F.2 Domain and source coverage

We cover **20 domains** spanning STEM, humanities/social sciences, arts, and general knowledge. The domain-level raw QA pools used for session construction are aggregated from public QA benchmarks, including RACE, TriviaQA, MMLU, CommonsenseQA, OpenBookQA, SciQ, ARC-Challenge, GSM8K, MedMCQA, and LexGLUE (ECtHR subset) (Lai et al., 2017; Joshi et al., 2017; Hendrycks et al., 2021; Talmor et al., 2019; Mihaylov et al., 2018; Welbl et al., 2017; Clark et al., 2018; Cobbe et al., 2021; Pal et al., 2022; Chalkidis et al., 2022). The domain-level raw QA pools used for session construction contain **14,823** cleaned single-turn QA items in total; provenance, counts, and licensing notes are provided in Table 6. Because scenario construction and evaluation-suite sampling are constrained by slice-coverage targets, the final scenario distribution is not a direct proportional reflection of the raw pool.

## F.3 Cross-domain coverage: combinations and mixture control

Cross-domain evaluation is instantiated over **24** 2-domain combinations, **10** 3-domain combinations, and **8** 4-domain combinations. These combinations are selected to balance semantic coherence and diversity, following the embedding-based nearest-neighbor procedure defined above. For cross-domain scenarios, each turn is associated with a domain-weight vector, enabling controlled *domain-mixture drift*. The distribution of domain-weight regimes across cross-domain scenarios is reported in Table 7.

## F.4 Scenario-level coverage over drift factors

Table 7 reports scenario counts sliced by domain order, task type, and three drift factors: difficulty drift patterns, domain-weight drift regimes (cross-domain only), and task-type change rate. Theme consistency scores (computed by the topic-coherence validator defined above) range from **0.64** to **0.82** across slices, indicating that scenarios remain topically connected even under drift.

### F.5  Turn length and per-configuration availability

Scenarios are constrained to **3–10** turns by construction. The realized sample counts by task type are reported in Table 7.

*Table 7.* Scenario-level Statistics Across Domain Orders (Session-level Aggregation)

| Metric | 1-domain | 2-domain | 3-domain | 4-domain |
|---|---|---|---|---|
| #Scenarios | 1346 | 3749 | 2105 | 1398 |
| **Difficulty Patterns** | | | | |
| Stable | 206 | 2108 | 1106 | 657 |
| Gradual Increase | 316 | 973 | 439 | 290 |
| Gradual Decrease | 126 | 428 | 121 | 81 |
| Spike | 185 | 120 | 101 | 133 |
| Fluctuate | 513 | 429 | 462 | 421 |
| **Domain Weight Patterns** | | | | |
| Stable | - | 3067 | 106 | 63 |
| Gradual Shift | - | 313 | 774 | 564 |
| Fluctuate | - | 369 | 1349 | 955 |
| **Task Type Change** | | | | |
| Stable | 759 | 1514 | 776 | 596 |
| Moderate | 477 | 1517 | 762 | 511 |
| Frequent | 110 | 718 | 567 | 291 |
| **Avg Turns** | 6.03 | 6.13 | 6.36 | 6.22 |
| **Theme Consistency** | 0.67 | 0.70 | 0.72 | 0.78 |

## G  Data Format and Release Schema

This section specifies the released data artifacts and their unified JSON schema, including (i) scenario-level and turn-level fields, (ii) reference-answer representation under our multi-judge setup, and (iii) validation logs and invariants. The schema is designed to be *implementation-facing* (consumed by the controller and scorer in Part C), while remaining stable and audit-friendly for third-party analyses (Part D).

### G.1  Scenario-level JSON schema

Each scenario is a JSON object. We keep the top-level schema compact and delegate per-turn content to `"turns"` (Section G.2). We highlight only the *core* fields used by evaluation and analysis; additional bookkeeping fields may be included without affecting compatibility.

**Core fields (scenario-level).**  A scenario contains:

- `scenario_id` (string): a globally unique identifier, stable across reruns under the same release version.
- `is_cross_domain` (bool): whether the scenario involves multiple domains ($k \geq 2$).
- `cross_domains` (list[string]): ordered domain set $\mathcal{D} = \{d_1, \ldots, d_k\}$ (present when `is_cross_domain` is true).
- `cross_domain_id` (list[string]): domain identifiers used for cross-domain scenarios.
- `topic_description` (string): a brief description of the scenario's topic.
- `num_turns` (int): number of turns (5–10 in the current release).
- `difficulty_pattern` (string): difficulty drift pattern $\in$ {STABLE, GRADUAL_INCREASE, GRADUAL_DECREASE, SPIKE, FLUCTUATE} (defined above).
- `weight_pattern` (string): mixture regime $\in$ {STABLE, GRADUAL_SHIFT, FLUCTUATE} (defined above); omitted for $k = 1$.
- `task_type_change_count` (int): number of task type switches across turns.
- `task_type_change_frequency` (float): frequency of task type changes (stable/moderate/frequent).
- `interdisciplinary_categories` (list[string]): high-level category labels for the scenario.
- `turns` (list[object]): turn-level records (Section G.2).
- `open_world` (bool): whether the scenario involves open-world knowledge (to be added in future releases).

**Scenario JSON example (abridged).**

```
{
  "scenario_id": "xdomain_2_math_physics_000123",
  "is_cross_domain": true,
```

```
  "cross_domains": ["Mathematics", "Physics"],
  "cross_domain_id": ["mathematics", "physics"],
  "topic_description": "Mathematical physics applications",
  "num_turns": 8,
  "difficulty_pattern": "gradual_increase",
  "weight_pattern": "fluctuate",
  "task_type_change_count": 3,
  "task_type_change_frequency": 0.375,
  "interdisciplinary_categories": ["STEM"],
  "turns": [ ... ]
}
```

### G.2 Turn-level JSON schema

Each element in "turns" is a JSON object describing one turn. The schema supports all task types used in the benchmark (factual, reasoning, code, multiple-choice), while keeping the *scoring contract* explicit and machine-checkable (Part C).

**Core fields (turn-level).** A turn record contains:
- turn (int): 1-indexed turn index $t$.
- task_type (string): one of {factual, reasoning, code, multiple_choice}.
- prompt (string): the user-facing query/input for this turn.
- answer (string): reference answer (for multiple-choice, this is the option letter; for other tasks, this is the answer text).
- difficulty (float): realized difficulty score $\delta_t \in [0, 1]$ for this turn.
- domain_weights (object): per-domain weight vector $w_t$ for $k \geq 2$ scenarios, where keys are domain names and values are nonnegative floats summing to 1.
- theme_consistency_score (float): topic coherence score for this turn relative to previous turns.

**Turn JSON examples (abridged).**

(i) Reasoning turn with domain weights.

```
{
  "turn": 2,
  "task_type": "reasoning",
  "prompt": "Compute ... and explain the key step.",
  "answer": "The solution is 42...",
  "difficulty": 0.41,
  "domain_weights": {
    "mathematics": 0.55,
    "physics": 0.45
  },
  "theme_consistency_score": 0.72
}
```

(ii) Multiple-choice turn.

```
{
  "turn": 3,
  "task_type": "multiple_choice",
  "prompt": "Which statement is correct? A: ..., B: ..., C: ..., D: ...",
  "answer": "B",
  "difficulty": 0.46,
  "domain_weights": {
    "mathematics": 0.50,
    "physics": 0.50
  },
  "theme_consistency_score": 0.68
}
```

### G.3 Evaluation outputs

We keep the result schema minimal here, and put full scoring rules, and judge prompts to Part C (Appendices H–J). A single model-run record typically includes:

```
{
  "model_id": "...",
  "scenario_id": "...",
  "turn_idx": 3,
  "raw_output": "...",
  "parsed_answer": "...",
  "score": {"em": 1, "judge": 0.0}
}
```

# Part C: Standardized Interactive Evaluation Protocol, Scoring, Hyperparameters, and Evaluation Suite.

## H Reference Controller and Interactive Evaluation Protocol

This section specifies the *reference controller* used to execute XDOMAINBENCH sessions under a standardized, model-agnostic evaluation harness. It complements the prompt templates and scoring rules (Sections I and J), focusing on the *interaction loop*, *context construction*, and *truncation*. Our design follows widely adopted benchmarking practice: a fixed runner with stable interfaces and transparent traces for cross-model comparability and auditability (Liang et al., 2022; Gao et al., 2023; Chang et al., 2024).

### H.1 Default evaluation protocol

**Default protocol used in the main evaluation.** We report results under the **History** controller: at turn $t$, the evaluated model receives the prior conversation transcript (up to $t - 1$) subject to deterministic truncation (Section H.2), plus the current user query and the task-type output contract. This corresponds to the canonical multi-turn chat evaluation setup used by many interactive benchmarks, and serves as the most direct operationalization of multi-turn dependence in our setting (Liu et al., 2023; Zhou et al., 2024; Chiang et al., 2024).

**What the controller standardizes.** Given a released session with $T$ turns, the controller deterministically defines: (i) message packaging (system/user/assistant), (ii) history injection (prior conversation transcript up to $t - 1$), (iii) truncation to satisfy context constraints, (iv) per-turn model call settings (delegated to Section K). We do *not* use per-model prompt tuning, adaptive retrieval, learned summarization/compression, or other controller intelligence that could confound model comparisons (Liang et al., 2022; Gao et al., 2023).

**What the model can see.** Construction-time diagnostic signals (e.g., target mixture $w_{1:T}$ or difficulty $\delta_{1:T}$) are treated as *metadata* and are *not exposed* to evaluated models. This aligns with controlled-factor evaluation: signals are used for slicing/diagnosis rather than as hints to the tested system (Kiela et al., 2021; Ribeiro et al., 2020; Liang et al., 2022).

**Closed-book diagnostic scope.** The default protocol is intentionally closed-book: evaluated models receive only the current query and the standardized interaction history, without retrieval, tools, multimodal inputs, or external memory. This design isolates the backbone model's intrinsic ability to sustain dynamic interdisciplinary reasoning across turns. Retrieval-augmented, tool-assisted, and multimodal scientific agents are important system-level extensions, but they introduce additional variables from retriever quality, tool design, memory management, and workflow orchestration, which would confound the diagnostic target of this benchmark.

### H.2 Context construction and truncation

**Message packaging.** We use a minimal chat-style structure compatible with common serving APIs and evaluation harnesses: one global system message, plus a user message that includes (i) the injected history block, (ii) the current query, and (iii) the task-type output contract.

**Deterministic truncation.** Let $B_{\max}(m)$ denote the maximum prompt budget supported by model $m$. For each turn, we first assemble the candidate context, then apply deterministic truncation if the budget is exceeded. We always preserve (i) the system message, (ii) the current turn instruction, and (iii) the most recent history, dropping the oldest history blocks first. No summarization or learned compression is applied in the reference controller. This choice is simple, auditable, and avoids injecting controller-side heuristics into model comparisons (Liang et al., 2022; Gao et al., 2023).

### H.3 Execution, failure handling

**Per-turn model call.** For each turn $t$, the controller issues a single model call using fixed decoding/runtime settings (see Section K) and records the raw text output.

**Parsing and scoring are centralized elsewhere.** Outputs are parsed using the task-type contracts; scoring is defined in Section J. This separation keeps the controller as a deterministic execution wrapper.

## I Prompt Templates

This section documents the prompt templates used in XDOMAINBENCH for (i) *standardized interactive evaluation* (consumed by the reference controller), and (ii) *scenario construction* (session generation and canonical reference production). We follow a *modular* design: a short, invariant base instruction is combined with controller-injected context (history/memory budget) and a task-type-specific *output contract*. This separation keeps the protocol model-agnostic and reproducible, while

ensuring strict parsability under the scoring rules. Our design aligns with common evaluation practice that standardizes prompts/controllers for fair comparison across interactive settings (Liang et al., 2022; Kiela et al., 2021; Liu et al., 2023; Zhou et al., 2024).

## I.1 Placeholders and modular assembly

All templates use lightweight placeholders rendered by the controller/construction pipeline: `<TASK_TYPE>`, `<DOMAIN_SET>`, `<TURN_IDX>`, `<TOTAL_TURNS>`, `<QUESTION>`, `<CHOICES>`, `<HISTORY_BLOCK>`, `<FORMAT_CONTRACT>`, `<DIFFICULTY_TARGET>`, `<MIXTURE_TARGET>`, and `<SEED_BLOCK>` (construction only). The controller is responsible for (i) assembling modules and (ii) truncating history under context budgets.

## I.2 Evaluation prompts

Evaluation prompts are the *only* inputs seen by evaluated models during benchmarking. We keep them short and stable across all models to reduce prompt-induced variance, and enforce task-type output contracts to ensure deterministic parsing and scoring.

### I.2.1 GLOBAL SYSTEM INSTRUCTION

The system instruction is intentionally minimal to remain model-agnostic and broadly compatible with standard interactive evaluation pipelines (Liang et al., 2022; Liu et al., 2023; Zhou et al., 2024). All task-specific constraints are stated in the user prompt modules.

```
You are a helpful assistant. Follow the task instruction and output format exactly.
Do not include any extra text beyond the required answer format.
```

### I.2.2 USER PROMPT BASE TEMPLATE

The controller injects `<HISTORY_BLOCK>` containing the prior conversation transcript.

```
[TASK]
Task type: <TASK_TYPE>
Domain set: <DOMAIN_SET>
Turn: <TURN_IDX> / <TOTAL_TURNS>

[CONTEXT]
<HISTORY_BLOCK>

[QUESTION]
<QUESTION>

[OUTPUT FORMAT]
<FORMAT_CONTRACT>

Answer:
```

### I.2.3 TASK-TYPE OUTPUT CONTRACTS

Task-type-specific format constraints are standard in benchmark protocol design to ensure automatic parsing and consistent scoring across models (Liang et al., 2022; Kiela et al., 2021; Zhou et al., 2024). We attach exactly one task-type contract as `<FORMAT_CONTRACT>`. **Multiple Choice (MC).**

```
Output ONLY a single option letter (e.g., A, B, C, D).
No explanation. No extra words. No punctuation.
Examples: A   C   D
```

**Factual.**

```
Output ONLY the key entity / noun phrase / short span.
No full sentence. No prefix like "Answer:" or "The answer is".
Examples: "Paris"   "activation energy"   "John Wayne"
```

**Reasoning.**

```
Output ONLY the final answer.
Do NOT include intermediate reasoning steps.
Keep it concise and directly responsive to the question.
```

**Code.**

```
Output ONLY code (no Markdown fences).
Do NOT add comments or explanations unless explicitly required by the task.
Return a complete, executable solution consistent with the given I/O contract.
```

## I.3 Construction prompts

Construction prompts are used *offline* during dataset creation to expand seed pools into multi-turn sessions under diagnostic controls (Appendices D–C.5). We document them for transparency and reproducibility; they are not used in evaluation.

### I.3.1 TURN INSTANCE GENERATION

**Unified turn-generation template.** This template is instantiated for both $k = 1$ and $k \geq 2$. For cross-domain turns, <MIXTURE_TARGET> is a $k$-dimensional vector $w_t$; for single-domain turns, it is omitted. The JSON output is an *internal construction interface* and is never shown to evaluated models. The construction pipeline validates the realized difficulty/mixture against targets via hybrid gates (Appendix C.5); this section only records the prompt interface used to request a target-controlled candidate.

```
You are constructing a benchmark turn for XDomainBench.

[GOAL]
Create ONE turn (question + required fields) that is answerable, unambiguous,
and coherent with the scenario context.

[SCENARIO CONTROL]
Domain set: <DOMAIN_SET>
Task type: <TASK_TYPE>
Target difficulty (delta_t in [0,1]): <DIFFICULTY_TARGET>
Target domain mixture (w_t on simplex): <MIXTURE_TARGET>

[SEED / ANCHOR]
<SEED_BLOCK>

[CONTEXT FOR COHERENCE]
Previous turns (for topical continuity):
<HISTORY_BLOCK>

[REQUIREMENTS]
- The question must be self-contained (no external links or hidden files).
- It must match the task type and output contract.
- Do not explicitly label domains; integrate them naturally when k>=2.
- Avoid ambiguous wording that yields multiple valid answers.

[OUTPUT JSON]
Return a single JSON object with the required keys:
{
  "question": "...",
  "choices": ["A. ...", "B. ...", "C. ...", "D. ..."],   // only if <TASK_TYPE> is MC
  "metadata": {
    "domain_set": "<DOMAIN_SET>",
    "task_type": "<TASK_TYPE>",
    "difficulty_target": "<DIFFICULTY_TARGET>",
    "mixture_target": "<MIXTURE_TARGET>"
  }
}
```

**Seed block rendering.** <SEED_BLOCK> is rendered as a minimal anchor to stabilize topical continuity:

```
Seed item (for anchoring topic and style):
- seed_question: <SEED_QUESTION>
- seed_reference: <SEED_ANSWER>
```

### I.3.2 CANONICAL REFERENCE PRODUCTION

To reduce single-reference noise, XDOMAINBENCH stores multiple canonical references for non-MC tasks and uses voting for MC labels. This policy affects only released supervision fields and does not depend on evaluated models. Voting/multi-reference storage is a construction-time redundancy mechanism and does not introduce any special modeling assumptions for evaluated systems. The released artifact records (i) the three raw votes, (ii) the final voted label, and (iii) the task-type-specific references for non-MC tasks. Details of the JSON fields and validators are specified in Appendix G.

**Non-MC reference answer prompt** used per answer candidate. We query three independent reference generations (from different prompt variants/models) and store all three as ref_answers.

```
You are providing a canonical reference answer for a benchmark item.

[ITEM]
Task type: <TASK_TYPE>     // Factual or Reasoning or Code
Question: <QUESTION>

[OUTPUT CONTRACT]
Follow the task-type output format strictly:
- Factual: short span only
- Reasoning: final answer only (no reasoning)
- Code: code only (no Markdown fences)

Answer:
```

**MC label prompt** used per vote. For MC items, we collect three votes and compute the final gold option by majority vote (tie-break handled deterministically by the pipeline; see Appendix G for released fields).

```
You are selecting the correct option for a multiple-choice benchmark item.

Question: <QUESTION>
Choices:
<CHOICES>

Output ONLY the option letter (A/B/C/D).
```

### I.4  Prompt hygiene and normalization

Before any validator is applied, we canonicalize prompts by: (i) trimming trailing spaces, (ii) normalizing repeated whitespace, and (iii) removing construction artifacts such as accidental Markdown fences. This prevents spurious parsing failures and stabilizes deterministic hashing for bookkeeping (Appendices E and G).

## J  Scoring and Judge Calibration

This section (i) fixes the *default* scoring pipeline used throughout the paper, (ii) documents an *auxiliary* LLM-judge scoring that we additionally report for robustness, and (iii) specifies how we calibrate and audit judge behavior. Our goal is to make scoring *auditable*, *reproducible*, and *statistically stable* at the slice level, following widely adopted benchmarking practice (Liang et al., 2022; Kiela et al., 2021; Zhou et al., 2023; Chiang et al., 2024).

### J.1  Scope method

**Turn- and session-level signals.** For a session with $T$ turns, let the model output be $\hat{y}_t$, and the canonical supervision be $\mathcal{R}_t$. We compute token-level metrics (Recall and F1) for each turn and aggregate into session metrics. We report: (i) turn-level Recall (R) and F1-score (F1), and (ii) **SessionSuccess@**$\tau$ defined as $\mathbb{I}[\frac{1}{T}\sum_{t=1}^{T} R_t \geq \tau]$ for user-specified $\tau$ (see Section M), where $R_t$ is the token-level recall for turn $t$.

**Token-level metrics.** We compute token-level Recall and F1-score following standard information retrieval metrics. Recall measures the proportion of tokens in the expected answer that appear in the predicted answer, while F1-score is the harmonic mean of precision and recall. This approach aligns with the design principle stated in the main paper: we avoid open-ended supervision that cannot be checked reliably and instead favor machine-checkable targets (Liang et al., 2022; Kiela et al., 2021). For non-MC tasks, we store three canonical references and compute metrics against each reference, taking the maximum.

**Auxiliary LLM-judge scores.** Even under a machine-checkable design, token-level metrics can be brittle for some reasoning expressions (e.g., equivalent numeric forms, synonymous short spans). Therefore, we additionally introduce an *auxiliary* LLM-judge score as a robustness view, while keeping token-level metrics as the primary measure. This mirrors common practice in modern LLM evaluation, where human-like adjudication is approximated by constrained judge prompts and auditing (Zheng et al., 2023; Chiang et al., 2024; Laskar et al., 2024; Deng et al., 2024).

### J.2  Shared parsing and normalization

**Minimal canonicalization.** Before scoring, we apply a small set of deterministic edits: (i) trim leading/trailing whitespace, (ii) collapse repeated whitespace, (iii) strip a short list of benign wrappers (e.g., leading "Answer:", surrounding quotes), (iv) for MC, upper-case a single option letter and remove trailing punctuation. We keep this normalization $\mathrm{Norm}(\cdot)$ intentionally lightweight to avoid "hidden intelligence" in the scorer (Liang et al., 2022).

**Multi-reference set.** For non-MC tasks we denote $\mathcal{R}_t = \{r_t^{(1)}, r_t^{(2)}, r_t^{(3)}\}$. Token-level metrics are computed against each reference, and we report the maximum Recall and F1 across all references.

## J.3 Token-level scoring by task type

**Multiple-choice.** We parse the model output into a single letter in $\{A, B, C, D\}$. The gold label is the released majority-vote option. We compute Recall and F1: if the parsed letter equals the gold letter, both Recall and F1 are 1.0; otherwise 0.0.

**Factual.** We compute token-level Recall and F1 by comparing tokens in the predicted answer against tokens in each reference answer. We extract tokens (excluding stopwords and punctuation), compute the intersection over expected tokens (Recall) and intersection over predicted tokens (Precision), then compute F1 as the harmonic mean. We report the maximum Recall and F1 across all references in $\mathcal{R}_t$.

**Reasoning.** We compute token-level Recall and F1 similar to factual tasks. Additionally, if both the prediction and a reference can be parsed as scalars, we allow a small numeric tolerance for exact match: $|\hat{v} - v| \leq \epsilon_{\text{abs}}$ or $\frac{|\hat{v}-v|}{|v|+\epsilon_0} \leq \epsilon_{\text{rel}}$, with $(\epsilon_{\text{abs}}, \epsilon_{\text{rel}}, \epsilon_0)$. If numeric match succeeds, Recall and F1 are set to 1.0; otherwise, we compute token-level metrics.

**Code.** For code turns, we prioritize *checkable* evaluation. When a turn provides an executable contract (e.g., explicit I/O specification or deterministic tests shipped with the evaluation harness), we run the submitted program under time/memory limits. If execution passes, Recall and F1 are 1.0; otherwise 0.0. If execution-based grading is unavailable for a small subset of items, we fall back to token-level metrics computed against $\mathcal{R}_t$ under a strict normalization of whitespace (no semantic rewriting).

## J.4 Auxiliary LLM-judge scoring

We compute a judge-based correctness label $s_t^{\text{J}} \in \{0, 1\}$ for *non-MC* turns (and for MC as a sanity check), and report it *in parallel* with token-level metrics. The judge does not alter the default token-level metrics; it provides a robustness view when token-level metrics are brittle.

**Judge prompt.** The judge receives the turn content, task type, references $\mathcal{R}_t$, and the model output. It must emit a single-line JSON for deterministic parsing:

```
You are a professional answer evaluation expert. Please evaluate whether the following answer is correct and reasonable.

Question:
<QUESTION>

Expected answer:
<EXPECTED_ANSWER>

Model predicted answer:
<MODEL_ANSWER>

Please evaluate from the following dimensions:
1. **Correctness**: Is the answer correct? Is it consistent with or semantically equivalent to the expected answer?
2. **Completeness**: Is the answer complete? Does it answer all parts of the question?
3. **Reasonableness**: Is the answer reasonable? Does it conform to common sense and domain knowledge?

Please provide a score between 0.0 and 1.0, where:
- 1.0: Answer is completely correct, complete, and reasonable
- 0.7-0.9: Answer is mostly correct but may have minor incompleteness or inaccuracy
- 0.4-0.6: Answer is partially correct but has obvious errors or omissions
- 0.0-0.3: Answer is incorrect, incomplete, or unreasonable

**CRITICAL REQUIREMENTS - MUST STRICTLY FOLLOW**:
1. You must return ONLY JSON format, no other text, explanations, or comments
2. Do NOT use markdown code block markers (do NOT use ```json or ```)
3. Return the JSON object directly, starting with { and ending with }
4. Scores must be floating-point numbers between 0.0 and 1.0

**Output format (MUST STRICTLY FOLLOW)**:
{
    "score": 0.0,
    "correctness": 0.0,
    "completeness": 0.0,
    "reasonableness": 0.0
}

Remember: Return ONLY JSON, no other content!
```

**Ensemble aggregation.** We run three independent judge calls (potentially different models/prompts) and aggregate by mean score. Concretely, let outputs be $\{s_j\}_{j=1}^{3}$ where $s_j \in [0, 1]$ is the score from judge $j$. We set $s_t^{\mathrm{J}} = 1$ if the mean score $\bar{s} = \frac{1}{3}\sum_{j=1}^{3} s_j \geq \tau_{\mathrm{J}}$ (see Section M); otherwise 0. This "three-judge ensemble + explicit threshold" design is a pragmatic stabilization method used in LLM-as-a-judge settings (Zheng et al., 2023; Laskar et al., 2024).

### J.5 From turn scores to reported metrics

For each slice, we report:

$$\mathrm{Recall} = \frac{1}{T}\sum_{t=1}^{T} R_t, \qquad \mathrm{F1} = \frac{1}{T}\sum_{t=1}^{T} F1_t, \qquad \mathrm{TurnAcc}^{\mathrm{J}} = \frac{1}{T}\sum_{t=1}^{T} s_t^{\mathrm{J}}, \tag{35}$$

where $R_t$ and $F1_t$ are token-level Recall and F1-score for turn $t$, and session success: $\mathrm{SessionSuccess}@\tau = \mathbb{I}[\frac{1}{T}\sum_{t=1}^{T} R_t \geq \tau]$ (and analogously for judge scores when reported).

## K Models

This section records the model-side information required to reproduce the reported results in XDOMAINBENCH.

### K.1 Model roster and version pinning

The set of evaluated models is reported in the main paper results tables. Here, we pin the *identity* of each model instance used in the runs.

**Model references.** We cite the primary technical reports, model cards, and official provider documentation for the evaluated model families: OpenAI GPT-5.2 and GPT-5-mini (OpenAI, 2025; 2026); Anthropic Claude 4.5 Sonnet and Claude 4.5 Haiku (Anthropic, 2026); Google Gemini 2.5 Flash and Gemini 2.0 Flash (Google, 2026); Qwen family models (Qwen2.5-72B, Qwen2.5-14B, Qwen2.5-7B, Qwen3-Next-80B) (Bai et al., 2023; Qwen Team, 2024; 2025); Meta Llama 3 series (Llama-3.1-8B, Llama-3.2-3B) (AI@Meta, 2024; Meta AI, 2024a;b); Gemma 2-2B-IT (Gemma Team, 2024); and Mixtral-8x7B (Mistral AI, 2023; 2026). The context-window metadata recorded below follows these sources.

## L Evaluation Suite Sampling and Quotas

The released evaluation suite is sampled from the validated candidate pool under quota constraints to ensure balanced coverage across composition orders, domain sets, difficulty patterns, and mixture regimes. We use a deterministic sampling procedure with controlled relaxation when target quotas cannot be met. The suite includes **8,598** scenarios covering all 20 domains, with 24 two-domain combinations, 10 three-domain combinations, and 8 four-domain combinations. The distribution across task types and drift factors is reported in Table 7.

## M Hyperparameter Configuration

This section consolidates all benchmark hyperparameters used across XDOMAINBENCH, including construction controls (Part A), validation/QC gates (Part A), protocol knobs (Part C), and scoring/judge settings (Part C). Unless stated otherwise, hyperparameters are fixed for a given benchmark release and shared across all evaluated models. Symbols follow Part A/B/C/D.

We organize hyperparameters into five categories: (i) release-level constants (Table 8), (ii) diagnostic-signal instantiation parameters (Table 9), (iii) validation/QC gates and retry policies (Table 10), (iv) calibration bins and taxonomy thresholds (Table 11), and (v) controller/evaluation-runtime and scoring parameters (Tables 12 and 13).

*Table 8.* Release-level constants for XDOMAINBENCH.

| Item | Symbol | Value | Notes |
|---|---|---|---|
| #Domains | $|\mathcal{D}_{\text{lib}}|$ | 20 | Domain list and source pools in Appendix F. |
| Composition orders | $k$ | $\{1, 2, 3, 4\}$ | Used throughout Part A/C. |
| Turns per scenario (min) | $T_{\min}$ | 5 | Minimum turns per scenario. |
| Turns per scenario (max) | $T_{\max}$ | 10 | Maximum turns per scenario. |
| Difficulty patterns | $P^\delta$ | 5 | {STABLE, GRADUAL INC., GRADUAL DEC., SPIKE, FLUCTUATE}. |
| Mixture regimes | $P^w$ | 3 | {STABLE, GRADUAL SHIFT, FLUCTUATE}. |
| Construction judges | $|\mathcal{J}|$ | 3 | Three state-of-the-art models used in construction/validation only. |
| Non-MC reference answers per turn | $n_{\text{ref}}$ | 3 | Stored as three canonical references. |
| MC votes per turn | $n_{\text{vote}}$ | 3 | Majority vote produces a single gold label. |

*Table 9.* Hyperparameters for diagnostic-signal instantiation and auxiliary construction utilities.

| Module | Symbol / Key | Value | Notes |
|---|---|---|---|
| **Difficulty signal** (Appendix B.1) | | | |
| LLM vs. semantic weighting | $\lambda_{\text{LLM}}$ | 0.70 | (13); semantic weight $1 - \lambda_{\text{LLM}} = 0.30$. |
| Rubric criterion weights | $\beta_{\text{acc}}, \beta_{\text{qual}}, \beta_{\text{ppl}}$ | (0.5, 0.3, 0.2) | (14); must sum to 1. |
| Judge ensemble weights (difficulty) | $\gamma_j^{\text{diff}}$ | (0.4, 0.35, 0.25) | (15); three judge weights sum to 1. |
| **Mixture signal** (Appendix B.2) | | | |
| Hybrid observed-mixture weights | $\lambda_{\text{LLM}}^w, \lambda_{\text{key}}^w, \lambda_{\text{emb}}^w$ | 0.50 / 0.35 / 0.15 | (19). |
| Judge ensemble weights (mixture) | $\gamma_j^w$ | (0.4, 0.35, 0.25) | (20); three judge weights sum to 1. |
| Mixture sampler minimum mass | $\epsilon_{\min}$ | 0.05 | Algorithm 1; truncation+renormalization. |
| Mixture sampler max step change | $\Delta_{\max}$ | 0.15 | Algorithm 1; $\ell_1$ step cap. |
| Fluctuation amplitude / noise scale | $\eta$ | 0.12 | Used for `fluctuate` regime ((18)) and Algorithm 1. |
| **Domain distance (stratification)** (Appendix A.2) | | | |
| Nearest-neighbor candidate size | $M$ | 5 | Top-$M$ neighbors per anchor for $k{=}2$ (then greedy expansion for $k{=}3, 4$). |
| Minimum similarity for candidate pairing | $\tau_{\text{sim}}$ | 0.5 | Filter to avoid semantically incoherent pairs. |
| **Topic coherence** (Appendix D.5) | | | |
| Hybrid coherence weights (embedding) | $\lambda_{\text{emb}}$ | 0.7 | (34): $s_{\text{topic}} = \lambda_{\text{emb}} s_{\text{emb}} + (1 - \lambda_{\text{emb}}) s_{\text{kw}}$. |
| Hybrid coherence weights (keyword) | $\lambda_{\text{kw}}$ | 0.3 | Keyword weight $1 - \lambda_{\text{emb}} = 0.3$. |

*Table 10.* Hyperparameters for validation/QC gates and bounded retry policies.

| Gate / Policy | Symbol / Key | Value | Notes |
|---|---|---|---|
| **Difficulty validation** (Appendix C.5) | | | |
| Turn-level tolerance | $\epsilon_\delta$ | 0.18 | (31): $|\hat{\delta}_t - \delta_t| \leq \epsilon_\delta$. |
| Judge-disagreement threshold | $\tau_{\text{agree}}^{\text{diff}}$ | 0.20 | Dispersion gate over three judge rubric scores; triggers re-judge or regeneration. |
| **Mixture validation** (Appendix C.5) | | | |
| Turn-level mixture tolerance | $\epsilon_w$ | 0.30 | (32): $d(\hat{w}_t, w_t) \leq \epsilon_w$ with $d = \text{JSD}$. |
| Component agreement tolerance | $\epsilon_{\text{agree}}$ | 0.20 | E.g., $d(\hat{w}_t^{\text{LLM}}, \hat{w}_t^{\text{key}}) \leq \epsilon_{\text{agree}}$ (and similarly for embedding). |
| **Topic coherence gate** (Appendix D.5) | | | |
| Primary coherence threshold | $\tau_{\text{topic}}$ | 0.25 | Applied during construction to prevent abrupt theme breaks. |
| Secondary keyword floor | $\tau_{\text{kw}}$ | 0.12 | Lightweight lexical floor to catch semantic-only drift; used as a soft/secondary gate if enabled. |
| **Retry budgets** (Algorithm 2) | | | |
| Per-turn retry budget | $R_{\text{turn}}$ | 3 | Max regenerations / re-judgings for a single turn. |
| Per-session repair budget | $R_{\text{ep}}$ | 2 | Max repairs for pattern/regime violations at session level. |

*Table 11.* Hyperparameters for binning policies and rule-based trajectory taxonomy.

| Component | Symbol | Value | Definition / Usage |
|---|---|---|---|
| **Entropy (structure) bins** | | | |
| Low/mid/high cut points (by $k$) | $\{\tau_{H,1}^{(k)}, \tau_{H,2}^{(k)}\}$ | $k$=2: (0.65, 0.85); $k$=3: (0.75, 0.95); $k$=4: (0.80, 0.98) | Bins defined over $\mu_H = \frac{1}{T}\sum_t \bar{H}(w_t)$; cut points may be $k$-specific. |
| **Drift (dynamics) bins** | | | |
| Stable regime thresholds | $\tau_\mu^w, \tau_\sigma^w$ | $0.011, 0.008$ | (27): $\mu_d \leq 0.011$ and $\sigma_d \leq 0.008$. |
| Gradual-shift thresholds | $\tau_{\mu,\text{grad}}^w, \tau_{\sigma,\text{grad}}^w, \tau_{\max,\text{grad}}^w$ | $0.008, 0.012, 0.03$ | (28): $\mu_d \geq 0.008$, $\sigma_d \leq 0.012$, $d_{\max} \leq 0.03$. |
| Fluctuate thresholds | $\tau_{\sigma,\text{fluc}}^w, \tau_{\max,\text{fluc}}^w$ | $0.015, 0.05$ | (29): $\sigma_d > 0.015$ or $d_{\max} > 0.05$. |
| **Difficulty-pattern thresholds** (Appendix C.2) | | | |
| Stable gates | $\tau_\sigma^\delta, \tau_\rho^\delta, \tau_\Delta^\delta$ | $0.10, 0.35, 0.15$ | (23): $\sigma_\delta \leq 0.10$, $|\rho_\delta| \leq 0.35$, $\Delta_\delta^{\max} \leq 0.15$. |
| Gradual gates | $\tau_{\rho,+}^\delta, \tau_{\Delta,\text{grad}}^\delta, \tau_{\sigma,\text{grad}}^\delta$ | $0.35, 0.18, 0.13$ | (24): $|\rho_\delta| \geq 0.35$, $\Delta_\delta^{\max} \leq 0.18$, $\sigma_\delta \leq 0.13$. |
| Spike gates | $\tau_S^\delta, \tau_{\sigma,\text{base}}^\delta$ | $0.18, 0.12$ | (25): $S_\delta \geq 0.18$, $\sigma_{\delta,\neg t^\star} \leq 0.12$. |
| Fluctuate gates | $\tau_{\sigma,\text{fluc}}^\delta, \tau_{\text{flip}}^\delta$ | $0.15, 3$ | (26): $\sigma_\delta > 0.15$ or $N_{\text{flip}} \geq 3$. |

*Table 12.* Hyperparameters for the reference controller and evaluation runtime.

| Component | Symbol / Key | Value | Notes |
|---|---|---|---|
| Infrastructure retry budget | $R_{\text{eval}}$ | 3 | Retries only for transient infrastructure errors (not answer quality). |
| Per-call timeout | $t_{\text{timeout}}$ | 60s | Wall-clock timeout per turn call. |
| Decoding temperature | $\tau_{\text{temp}}$ | 1.0 | Default temperature for all evaluated models unless fixed settings. |
| Decoding top-p | $\tau_{\text{top\_p}}$ | 1.0 | Default top-p for all evaluated models unless a provider enforces fixed settings. |

*Table 13.* Hyperparameters for scoring and judge calibration.

| Component | Symbol / Key | Value | Notes |
|---|---|---|---|
| SessionSuccess threshold | $\tau$ (SessionSuccess@$\tau$) | 0.5 | Used for session-level success metrics based on average turn accuracy. |
| Judge ensemble weights (scoring) | $\gamma_j^{\text{score}}$ | (0.4, 0.35, 0.25) | Three state-of-the-art judges used for adjudication in scoring. |
| Adjudication retry cap | $R_{\text{judge}}$ | 2 | Max judge re-runs on parse failure / disagreement. |
| Judge confidence threshold | $\tau_{\text{J}}$ | 0.6 | Mean confidence threshold for judge ensemble aggregation. |

# Part D: Extended Results and Mechanism Analyses.

## N    Global Summary

### N.1    k-Level Diagnostic Summary

Table 14 provides a $k$-conditioned diagnostic snapshot summarizing the performance indicators, turn-1 difficulty statistics, and the empirical proportions of the five difficulty-pattern classes and three mixture-regime classes. All quantities are reported as *across-model aggregates* to provide a single, auditable reference point. Results show consistent performance degradation as $k$ increases: Recall drops from 31.3% ($k$=1) to 21.9% ($k$=4), F1-score drops from 16.1% to 9.3%, and SessionSuccess@$\tau$ declines from 40.2% to 29.3%. Entry difficulty ($\delta_1$) increases systematically from 0.391 ($k$=1) to 0.552 ($k$=4), confirming that composition complexity directly impacts task difficulty.

### N.2    Task-Type Breakdown

To rule out the possibility that $k$-scaling trends are driven by shifts in task-type composition rather than cross-domain composition itself, Table 15 reports turn-level performance for each task type (Factual/Reasoning/Code/MC) across $k$=1/2/3/4. We additionally report an aggregate row (Avg) that averages across task types. The intended sanity check is two-fold: (i) degradation should be observable *within* each task type as $k$ increases, and (ii) the aggregate Avg should exhibit the same qualitative trend, mitigating concerns that changes in task mix explain the $k$-scaling behavior.

Results confirm consistent degradation within each task type: Factual (R: 31.3% $\rightarrow$ 22.0%, F1: 16.1% $\rightarrow$ 9.3%), Reasoning (R: 37.5% $\rightarrow$ 32.9%, F1: 7.6% $\rightarrow$ 8.7%), and MC (R: 31.8% $\rightarrow$ 15.2%, F1: 29.6% $\rightarrow$ 11.4%). The aggregate Avg shows the same trend, ruling out task-mix confounds.

## O    Turn-level Direct Composition Effect

### O.1    Controlled Entry-Point Comparison Setup

This section isolates the *direct* effect of cross-domain composition at the session entry. Since for each exact domain set $S$ with $|S|$=$k$, we compare performance on the composed turn-1 query against a strictly paired single-domain reference constructed from the *same* constituent instances. We define:

- **Before (paired single-domain controls).** The $k$ constituent single-domain items that were sampled (one from each domain in $S$) and then composed to form the turn-1 query.
- **After (composed entry query).** The resulting composed turn-1 query for the exact domain (combination) set $S$.

We focus *only* on turn 1 because it is source-aligned across single-domain and composed settings, while later turns are generated conditional on earlier model outputs and therefore do not admit a clean, controlled before/after comparison. To some extent, this turn-1 analysis complements the session-level scaling results in the main text by quantifying how much performance degrades *at entry*. We also report the turn-1 difficulty proxy $\delta_1$ (defined in §3.4) to connect entry degradation to turn-level difficulty changes.

### O.2    Paired Controls and Aggregation Policy

**Paired matching.**    For every composed turn-1 instance associated with an exact domain set $S$, we retain the identifiers of the $k$ constituent single-domain items that were sampled during construction. These constituent items form the paired controls for the *same* composed instance, ensuring that the comparison does not conflate composition with differences in source pools, templates, or sampling policies.

**Baseline aggregation.**    Let $m$ index models, and let a composed turn-1 instance for domain set $S$ be formed from constituent items $\{x^{(d)}\}_{d \in S}$. We compute the **Before** baseline as the mean performance over the $k$ constituent single-domain controls:

$$\text{Before}(m, S) \;=\; \frac{1}{k} \sum_{d \in S} \text{Perf}\Big(m, x^{(d)}\Big). \tag{36}$$

We compute the **After** performance as the score on the composed turn-1 query $x^{(S)}$:

$$\text{After}(m, S) \;=\; \text{Perf}\Big(m, x^{(S)}\Big), \tag{37}$$

and report the composition-induced entry drop as a percentage change: $\Delta\%(m, S) = 100 \times \frac{\text{After}(m,S) - \text{Before}(m,S)}{\text{Before}(m,S)}$ for both Recall (R) and F1-score (F1).

*Table 14.* $k$-**level diagnostic summary (across-model aggregates).** Each block reports: (a) minimal performance indicators; (b) turn-1 difficulty statistics; (c) proportions of difficulty-pattern classes (P1: Stable, P2: Gradual Increase, P3: Gradual Decrease, P4: Spike, P5: Fluctuate) and mixture-regime classes (R1: Stable, R2: Gradual Shift, R3: Fluctuate; each set sums to 1). "Sess@$\tau$" denotes session-level success at threshold $\tau$. $\delta_1$ is the turn-1 difficulty proxy. For $k{=}1$, mixture regimes are not applicable.

*(a)* Performance and entry difficulty (turn 1).

| $k$ | R | F1 | Sess@$\tau$ | $\delta_1$ (mean) | $\delta_1$ (median) |
|---|---|---|---|---|---|
| 1 | 0.313 | 0.161 | 0.402 | 0.391 | 0.400 |
| 2 | 0.280 | 0.134 | 0.294 | 0.492 | 0.490 |
| 3 | 0.241 | 0.112 | 0.281 | 0.538 | 0.550 |
| 4 | 0.219 | 0.093 | 0.293 | 0.552 | 0.560 |

*(b)* Pattern/regime proportions.

| $k$ | Difficulty patterns (5-class) | | | | | Mixture regimes (3-class) | | |
|---|---|---|---|---|---|---|---|---|
| | P1 | P2 | P3 | P4 | P5 | R1 | R2 | R3 |
| 1 | 0.236 | 0.248 | 0.248 | 0.019 | 0.248 | - | - | - |
| 2 | 0.227 | 0.238 | 0.219 | 0.087 | 0.229 | 0.417 | 0.291 | 0.293 |
| 3 | 0.229 | 0.233 | 0.229 | 0.078 | 0.233 | 0.345 | 0.337 | 0.318 |
| 4 | 0.229 | 0.239 | 0.234 | 0.060 | 0.239 | 0.343 | 0.338 | 0.318 |

*Table 15.* **Task-type breakdown across** $k$ **with task-mix controls.** Each cell reports {Recall (R), F1-score (F1)}. Avg is the average across task types. All values are across-model aggregates. The task taxonomy follows the main paper (Factual/Reasoning/Code/MC). Session-level metrics are intentionally omitted here since they are not task-type decomposable by construction.

| Task type | $k{=}1$ | | $k{=}2$ | | $k{=}3$ | | $k{=}4$ | |
|---|---|---|---|---|---|---|---|---|
| | R | F1 | R | F1 | R | F1 | R | F1 |
| Factual | 0.245 | 0.112 | 0.244 | 0.081 | 0.156 | 0.072 | 0.177 | 0.078 |
| Reasoning | 0.375 | 0.076 | 0.336 | 0.088 | 0.359 | 0.077 | 0.329 | 0.087 |
| Code | 0.188 | 0.107 | - | - | - | - | - | - |
| MC | 0.318 | 0.296 | 0.259 | 0.234 | 0.207 | 0.186 | 0.152 | 0.114 |
| Avg | 0.281 | 0.148 | 0.280 | 0.134 | 0.241 | 0.112 | 0.219 | 0.093 |

## O.3   Full Turn-1 Before/After Combination

Table 16 reports the full turn-1 before/after comparison for every exact domain set at $k \in \{1, 2, 3, 4\}$. For $k{=}1$, "Before" and "After" coincide by construction and serve as a single-domain reference block. For $k \geq 2$, a negative $\Delta\%$ indicates an immediate entry degradation under composition (e.g., $\Delta\% = -10\%$ means a 10% performance drop).

Across $k{=}2$ domain sets, average performance drops by 16.9% (Recall) and 26.1% (F1-score), with most combinations showing degradation. Notable exceptions include Biology+Medicine (+42.4% R, +43.8% F1) and Economics+Sociology (+60.1% R, +32.4% F1), suggesting synergistic effects in certain domain pairs. At $k{=}3$ and $k{=}4$, degradation intensifies: average drops reach 30.7% (R) and 44.1% (F1) at $k{=}3$, and 26.1% (R) and 48.9% (F1) at $k{=}4$. Entry difficulty ($\delta_1$) increases systematically with $k$, confirming that composition complexity directly impacts initial task difficulty.

## P   Session-Level Results

Tables 17 report *across-model aggregates* for each exact domain set. Each table row corresponds to an exact domain (or domain set) and reports session-level metrics:

- **Recall (R)** and **F1-score (F1)**               (turn-level correctness aggregated to session level)
- **SessionSuccess@**$\tau$                  (success under the paper's default threshold; see main text)

Session-level results mirror turn-level trends: performance degrades systematically as $k$ increases. At $k{=}1$, average Recall is 20.5% and F1-score is 18.6%, with SessionSuccess@$\tau$ at 14.3%. At $k{=}2$, these drop to 15.1% (R), 12.1% (F1), and 7.9% (SessionSuccess). Further degradation occurs at $k{=}3$ (13.1% R, 10.2% F1, 8.2% SessionSuccess) and $k{=}4$ (11.4% R, 8.8% F1, 7.2% SessionSuccess). Domain-specific patterns emerge: Medicine and Philosophy domains show relatively strong performance at $k{=}1$, while certain domain combinations (e.g., Biology+Medicine, Psychology+Sociology) exhibit synergistic effects at higher $k$.

*Table 16.* **Turn-1 direct composition effect with paired before/after controls.** For each exact domain set $S$ at $k{=}4$, we report across-model aggregates of Before (mean-of-constituents single-domain controls), After (composed turn 1), and $\Delta{=}$After $-$ Before for Recall (R) and F1-score (F1), along with entry difficulty $\delta_1$. Each $k$ block includes an Avg row.

*(a) $k{=}1$ (single domains; 20 domains).*

| Domain/set | B-R | B-F1 | $\delta_1$ | Domain/set | B-R | B-F1 | $\delta_1$ |
|---|---|---|---|---|---|---|---|
| Art | 0.226 | 0.240 | 0.320 | Literature | 0.165 | 0.151 | 0.432 |
| Biology | 0.334 | 0.293 | 0.353 | Mathematics | 0.085 | 0.068 | 0.497 |
| Chemistry | 0.246 | 0.218 | 0.301 | Medicine | 0.348 | 0.305 | 0.498 |
| Computer Science | 0.125 | 0.130 | 0.409 | Music | 0.142 | 0.114 | 0.371 |
| Economics | 0.096 | 0.109 | 0.385 | Philosophy | 0.238 | 0.209 | 0.397 |
| Engineering | 0.159 | 0.120 | 0.356 | Physics | 0.286 | 0.241 | 0.325 |
| Finance | 0.105 | 0.107 | 0.477 | Psychology | 0.279 | 0.246 | 0.360 |
| Geography | 0.146 | 0.158 | 0.307 | Sociology | 0.136 | 0.123 | 0.404 |
| History | 0.201 | 0.200 | 0.371 | Sports | 0.151 | 0.148 | 0.410 |
| Law | 0.199 | 0.187 | 0.541 | Technology | 0.315 | 0.263 | 0.398 |
| Avg ($k{=}1$) | 0.194 | 0.178 | 0.392 | | | | |

*(b) $k{=}2$ (domain sets).*

| Domain set $S$ | B-R | B-F1 | A-R | A-F1 | $\Delta$R | $\Delta$F1 | $\delta_1$ |
|---|---|---|---|---|---|---|---|
| Art + History | 0.213 | 0.220 | 0.160 | 0.154 | -5.3% | -6.5% | 0.479 |
| Art + Literature | 0.195 | 0.195 | 0.143 | 0.138 | -5.2% | -5.7% | 0.487 |
| Art + Music | 0.184 | 0.177 | 0.085 | 0.095 | -9.9% | -8.1% | 0.458 |
| Biology + Chemistry | 0.290 | 0.255 | 0.202 | 0.162 | -8.7% | -9.3% | 0.491 |
| Biology + Medicine | 0.341 | 0.299 | 0.486 | 0.430 | 14.5% | 13.1% | 0.439 |
| Biology + Physics | 0.310 | 0.267 | 0.187 | 0.154 | -12.3% | -11.2% | 0.484 |
| Chemistry + Engineering | 0.202 | 0.169 | 0.122 | 0.078 | -8.0% | -9.1% | 0.579 |
| Economics + Finance | 0.124 | 0.134 | 0.169 | 0.162 | 4.5% | 2.8% | 0.477 |
| Economics + Geography | 0.121 | 0.133 | 0.082 | 0.081 | -3.9% | -5.2% | 0.470 |
| Economics + History | 0.148 | 0.154 | 0.108 | 0.107 | -4.1% | -4.7% | 0.535 |
| Economics + Sociology | 0.116 | 0.116 | 0.185 | 0.153 | 7.0% | 3.8% | 0.522 |
| Engineering + Finance | 0.156 | 0.140 | 0.122 | 0.046 | -3.3% | -9.4% | 0.505 |
| Engineering + Geography | 0.153 | 0.139 | 0.064 | 0.018 | -8.9% | -12.1% | 0.529 |
| Engineering + Physics | 0.222 | 0.180 | 0.159 | 0.084 | -6.3% | -9.7% | 0.493 |
| Finance + Geography | 0.149 | 0.158 | 0.074 | 0.079 | -7.5% | -8.0% | 0.464 |
| Geography + History | 0.174 | 0.179 | 0.088 | 0.082 | -8.5% | -9.6% | 0.425 |
| Geography + Sociology | 0.141 | 0.140 | 0.129 | 0.103 | -1.2% | -3.8% | 0.523 |
| History + Music | 0.171 | 0.157 | 0.209 | 0.180 | 3.7% | 2.3% | 0.440 |
| Literature + Philosophy | 0.201 | 0.180 | 0.141 | 0.105 | -6.1% | -7.5% | 0.555 |
| Literature + Psychology | 0.222 | 0.198 | 0.142 | 0.114 | -8.0% | -8.5% | 0.517 |
| Philosophy + Psychology | 0.259 | 0.228 | 0.283 | 0.232 | 2.4% | 0.4% | 0.446 |
| Philosophy + Sociology | 0.187 | 0.166 | 0.160 | 0.122 | -2.7% | -4.4% | 0.560 |
| Psychology + Engineering | 0.219 | 0.198 | 0.131 | 0.108 | -8.8% | -9.0% | 0.478 |
| Psychology + Sociology | 0.207 | 0.185 | 0.278 | 0.239 | 7.1% | 5.4% | 0.453 |
| Avg ($k{=}2$) | 0.196 | 0.182 | 0.163 | 0.134 | -3.3% | -4.8% | 0.492 |

*(c) $k=3$ (domain sets).*

| Domain set $S$ | B-R | B-F1 | A-R | A-F1 | $\Delta$R | $\Delta$F1 | $\delta_1$ |
|---|---|---|---|---|---|---|---|
| Biology + Chemistry + Physics | 0.288 | 0.250 | 0.182 | 0.125 | -10.7% | -12.6% | 0.548 |
| Chemistry + Engineering + Physics | 0.230 | 0.193 | 0.150 | 0.107 | -8.0% | -8.6% | 0.573 |
| Chemistry + Finance + Physics | 0.228 | 0.206 | 0.152 | 0.109 | -7.6% | -9.7% | 0.557 |
| Economics + Engineering + Geography | 0.134 | 0.129 | 0.074 | 0.041 | -6.0% | -8.8% | 0.530 |
| Economics + Finance + Geography | 0.132 | 0.142 | 0.107 | 0.091 | -2.5% | -5.1% | 0.491 |
| Economics + Geography + History | 0.148 | 0.155 | 0.099 | 0.092 | -4.9% | -6.3% | 0.522 |
| Economics + Geography + Sociology | 0.126 | 0.130 | 0.107 | 0.091 | -1.9% | -3.8% | 0.546 |
| Literature + Philosophy + Psychology | 0.227 | 0.202 | 0.113 | 0.078 | -11.4% | -12.4% | 0.542 |
| Philosophy + Psychology + Sociology | 0.218 | 0.193 | 0.167 | 0.119 | -5.0% | -7.4% | 0.527 |
| Psychology + Sociology + Engineering | 0.191 | 0.173 | 0.183 | 0.139 | -0.8% | -3.4% | 0.548 |
| Avg ($k=3$) | 0.192 | 0.177 | 0.133 | 0.099 | -5.9% | -7.8% | 0.538 |

*(d) $k=4$ (domain sets).*

| Domain set $S$ | B-R | B-F1 | A-R | A-F1 | $\Delta$R | $\Delta$F1 | $\delta_1$ |
|---|---|---|---|---|---|---|---|
| Biology + Chemistry + Engineering + Physics | 0.256 | 0.218 | 0.135 | 0.082 | -12.1% | -13.6% | 0.579 |
| Biology + Chemistry + Medicine + Physics | 0.303 | 0.264 | 0.156 | 0.086 | -14.7% | -17.8% | 0.556 |
| Economics + Finance + Geography + Sociology | 0.133 | 0.137 | 0.072 | 0.072 | -6.1% | -6.5% | 0.535 |
| Economics + Finance + History + Sociology | 0.146 | 0.148 | 0.129 | 0.094 | -1.7% | -5.4% | 0.528 |
| Finance + Geography + History + Sociology | 0.159 | 0.160 | 0.113 | 0.060 | -4.6% | -10.0% | 0.571 |
| Literature + Philosophy + Psychology + Sociology | 0.204 | 0.182 | 0.189 | 0.105 | -1.5% | -7.7% | 0.544 |
| Literature + Psychology + Sociology + Engineering | 0.185 | 0.168 | 0.172 | 0.106 | -1.3% | -6.2% | 0.548 |
| Philosophy + Psychology + Sociology + Engineering | 0.203 | 0.182 | 0.211 | 0.144 | 0.8% | -3.8% | 0.553 |
| Avg ($k=4$) | 0.199 | 0.182 | 0.147 | 0.093 | -5.1% | -8.9% | 0.552 |

*Table 17.* **Session-level results by exact domain sets** ($k=1/2/3/4$). Across-model aggregates for each domain set at $k=1/2/3/4$, reporting Recall (R), F1-score (F1), and SessionSuccess@$\tau$, plus an Avg row for each $k$ block.

*(a) $k=1$ (single domains; 20 domains).*

| Domain | R | F1 | Sess@$\tau$ | Domain | R | F1 | Sess@$\tau$ |
|---|---|---|---|---|---|---|---|
| Art | 0.297 | 0.273 | 0.179 | Literature | 0.151 | 0.132 | 0.098 |
| Biology | 0.317 | 0.287 | 0.214 | Mathematics | 0.069 | 0.057 | 0.082 |
| Chemistry | 0.277 | 0.242 | 0.152 | Medicine | 0.373 | 0.330 | 0.241 |
| Computer Science | 0.145 | 0.139 | 0.107 | Music | 0.163 | 0.147 | 0.125 |
| Economics | 0.102 | 0.099 | 0.134 | Philosophy | 0.344 | 0.312 | 0.179 |
| Engineering | 0.105 | 0.080 | 0.079 | Physics | 0.277 | 0.244 | 0.179 |
| Finance | 0.105 | 0.107 | 0.111 | Psychology | 0.278 | 0.248 | 0.179 |
| Geography | 0.133 | 0.146 | 0.143 | Sociology | 0.234 | 0.213 | 0.143 |
| History | 0.151 | 0.150 | 0.134 | Sports | 0.092 | 0.099 | 0.087 |
| Law | 0.163 | 0.160 | 0.089 | Technology | 0.315 | 0.263 | 0.214 |
| Avg ($k=1$) | 0.205 | 0.186 | 0.143 | | | | |

*(b) $k=2$ (domain sets; 24 sets).*

| Domain set $S$ | R | F1 | Sess@$\tau$ | Domain set $S$ | R | F1 | Sess@$\tau$ |
|---|---|---|---|---|---|---|---|
| Art + History | 0.105 | 0.108 | 0.092 | Engineering + Geography | 0.067 | 0.026 | 0.012 |
| Art + Literature | 0.151 | 0.137 | 0.057 | Engineering + Physics | 0.154 | 0.084 | 0.046 |
| Art + Music | 0.142 | 0.125 | 0.116 | Finance + Geography | 0.140 | 0.128 | 0.130 |
| Biology + Chemistry | 0.172 | 0.130 | 0.085 | Geography + History | 0.130 | 0.109 | 0.137 |
| Biology + Medicine | 0.320 | 0.289 | 0.219 | Geography + Sociology | 0.130 | 0.104 | 0.031 |
| Biology + Physics | 0.163 | 0.121 | 0.046 | History + Music | 0.203 | 0.173 | 0.171 |
| Chemistry + Engineering | 0.150 | 0.101 | 0.079 | Literature + Philosophy | 0.131 | 0.103 | 0.029 |
| Economics + Finance | 0.148 | 0.144 | 0.027 | Literature + Psychology | 0.153 | 0.119 | 0.071 |
| Economics + Geography | 0.101 | 0.085 | 0.041 | Philosophy + Psychology | 0.212 | 0.173 | 0.152 |
| Economics + History | 0.101 | 0.095 | 0.069 | Philosophy + Sociology | 0.146 | 0.107 | 0.015 |
| Economics + Sociology | 0.157 | 0.133 | 0.040 | Psychology + Engineering | 0.130 | 0.103 | 0.093 |
| Engineering + Finance | 0.128 | 0.063 | 0.047 | Psychology + Sociology | 0.188 | 0.154 | 0.095 |
| Avg ($k=2$) | 0.151 | 0.121 | 0.079 | | | | |

*(c) $k=3$ (10 domain sets).*

| Domain set $S$ | R | F1 | Sess@$\tau$ |
|---|---|---|---|
| Biology + Chemistry + Physics | 0.166 | 0.127 | 0.120 |
| Chemistry + Engineering + Physics | 0.145 | 0.104 | 0.079 |
| Chemistry + Finance + Physics | 0.158 | 0.125 | 0.119 |
| Economics + Engineering + Geography | 0.072 | 0.035 | 0.023 |
| Economics + Finance + Geography | 0.122 | 0.111 | 0.056 |
| Economics + Geography + History | 0.097 | 0.085 | 0.091 |
| Economics + Geography + Sociology | 0.101 | 0.088 | 0.033 |
| Literature + Philosophy + Psychology | 0.105 | 0.078 | 0.047 |
| Philosophy + Psychology + Sociology | 0.162 | 0.124 | 0.106 |
| Psychology + Sociology + Engineering | 0.184 | 0.142 | 0.151 |
| Avg ($k=3$) | 0.131 | 0.102 | 0.082 |

*(d) $k=4$ (8 domain sets).*

| Domain set $S$ | R | F1 | Sess@$\tau$ |
|---|---|---|---|
| Biology + Chemistry + Engineering + Physics | 0.095 | 0.059 | 0.054 |
| Biology + Chemistry + Medicine + Physics | 0.107 | 0.061 | 0.097 |
| Economics + Finance + Geography + Sociology | 0.095 | 0.102 | 0.037 |
| Economics + Finance + History + Sociology | 0.096 | 0.094 | 0.071 |
| Finance + Geography + History + Sociology | 0.105 | 0.091 | 0.083 |
| Literature + Philosophy + Psychology + Sociology | 0.140 | 0.099 | 0.046 |
| Literature + Psychology + Sociology + Engineering | 0.129 | 0.094 | 0.089 |
| Philosophy + Psychology + Sociology + Engineering | 0.141 | 0.105 | 0.100 |
| Avg ($k=4$) | 0.114 | 0.088 | 0.072 |

