# OpenReview forum: "XDomainBench: Diagnosing Reasoning Collapse in High-Dimensional Scientific Knowledge Composition"
_ICML.cc/2026/Conference — ICML 2026 regular_

### Official Review · Reviewer_Lbp8 · 2026-03-05

**Soundness:** 2
**Presentation:** 4
**Significance:** 2
**Originality:** 3
**Overall Recommendation:** 3
**Confidence:** 3

**Summary:**

This paper introduces XDomainBench, a benchmark designed to evaluate the compositional generalization of Large Language Models (LLMs) in multi-turn, interdisciplinary scientific reasoning. Unlike traditional single-domain QA datasets, the authors parameterize the complexity of knowledge integration through two controllable dimensions: Composition Order (k) and Mixture Structure. By leveraging existing open-source datasets as single-domain seeds, they construct a dataset of nearly 8,600 multi-turn sessions with various difficulty and domain-weight trajectories. The primary finding is a systematic "reasoning collapse" as the composition order increases. While the motivation is highly relevant and the engineering effort is substantial, the manuscript suffers from fundamental methodological flaws regarding data contamination, the artificiality of its interactive dynamics, and metric brittleness. In its current state, it falls below the acceptance threshold.

**Compliance With Llm Reviewing Policy:**

Affirmed.

**Key Questions For Authors:**

1. Data Leakage How do you empirically isolate genuine "compositional reasoning collapse" from the failure to retrieve memorized data under highly complex, noisy prompt contexts? I strongly urge the authors to provide a sanity-check evaluation in the rebuttal using a small, rigorously hold-out dataset (e.g., newly published papers from late 2025/early 2026) across k=1 to k=4.
2. Trajectory Artificiality What is the justification for using non-reactive, hardcoded random walks to simulate scientific workflows? How can you prove that the failures in the Fluctuate mode are due to a lack of reasoning capability rather than the model correctly rejecting an illogical, unnatural context shift?
3. Metric Brittleness Please provide the variance in absolute performance drops (Table 3) if the primary evaluation metric is entirely replaced by a relaxed, semantic-equivalence LLM-as-a-judge protocol. Does the token-level F1 artificially depress the scores of models with strong paraphrasing capabilities?

**Limitations:**

No. While the authors discuss engineering constraints in the appendix, they completely omit discussions on the most critical threats to their core claims: pre-training data contamination of the seed datasets and the gap between their scripted trajectories and true state-reactive agentic workflows. These fundamental limitations must be addressed in the main text.

**Strengths And Weaknesses:**

## Strengths:
1. Timely Motivation: As the community shifts from static retrieval models toward interactive scientific agents, the need to evaluate dynamic, cross-disciplinary integration is critical. The authors correctly identify a major gap in current evaluation protocols.
2. Rigorous Parameterization: The formalization of domain mixtures using Shannon entropy and Jensen-Shannon divergence provides a mathematically sound framework for controlling cross-disciplinary complexity.

## Weaknesses (Critical Flaws):
1. Severe Data Contamination & Confounded Conclusions: According to Appendix F, the single-domain seed pool heavily relies on RACE, TriviaQA, and MMLU—datasets deeply embedded in the pre-training corpora of all evaluated frontier models. The strong performance at k=1 is highly likely a result of triggering pre-training memory. Consequently, the observed "collapse" at k>2 might simply be the disruption of retrieval cues caused by the noisy, high-dimensional context, rather than a genuine degradation in compositional reasoning. The paper fails to disentangle these two phenomena.
2. Pseudo-Interactive Dynamics: The paper claims to evaluate "interactive" reasoning, yet the domain weight shifts (e.g., the bounded random walk in the Fluctuate mode, Algorithm 1) are entirely scripted and hardcoded. In realistic agentic exploration—such as utilizing Monte Carlo Tree Search (MCTS) or reinforcement learning frameworks for deep web navigation or scientific discovery—state transitions and domain shifts are reactive responses to environmental feedback. Forcing a model to follow a non-reactive, unnatural script of context jumps evaluates its tolerance to disjointed prompts, not its interactive reasoning capability.
3. Circularity in Difficulty Scoring: Relying on an ensemble of three SOTA LLMs to define the ground-truth "difficulty" and "mixture weights" of the generated turns introduces severe circularity. This approach inevitably embeds the intrinsic architectural biases of the judge models into the benchmark's foundation, meaning the scores may merely reflect how closely a tested model aligns with the judge ensemble's latent distribution.
4. Brittleness of Token-Level Metrics: Relying heavily on token-level Recall and F1-score for open-ended Reasoning and Code tasks is fundamentally flawed in multi-turn contexts. Models capable of advanced semantic paraphrasing or alternative logical structuring are heavily penalized, artificially amplifying the magnitude of the reported "collapse."

---

> ### Author Rebuttal · Authors · 2026-03-31
>
> ## Response to Reviewer Lbp8
>
> We sincerely thank reviewer's careful assessment, for recognizing our *novelty and framework*, and for valuable concerns on **(1) contamination and trajectory realism in construction, and (2) evaluation validity**.
>
> ## Part 1. Dataset construction validity
>
> > **(1)** collapse signal is not reducible to **seed-set memorization** or **noisy-context disruption**, while indicating a genuine **cross-domain compositional burden**;
> > **(2)** regimes only control sampling and diagnosis, and final dataset only consists of **natural validated interactions**.
>
> ### 1.1 Contamination (Weakness 1 + Question 1)
>
> We really appreciate reviewer's insight of contamination in construction. Actually, contamination has been diluted:
> 1. **Scenario vibe only**: the seed pool is used only to anchor the initial scientific *vibe* before turn 1;
> 2. **Independent construction**: later turns shift in *difficulty, domain mixture, and local topic realization*, with both answers and diagnostic metadata re-generated and cross-validated;
> 3. **Multi-model interation**: the final sessions are composed and selected from *multiple interacting samples*.
>
> We further solve reviewer's concern about whether collapse is contaminated by noisy context. To start: for some higher-order settings, MoE's performance is stable or even improves, this non-monotonic pattern is inconsistent with a purely noise-disruption account, also discussed in `Reviewer NrVw Part 3`.
>
> But the above is not enough thoroughly, so we add two stronger **isolation experiments**.
>
> * **Evidence 1: strict pretraining-separation test.**
>
> GPT-J-6B and BigScience-T0-11B were released in 2021, whereas LexGLUE and MedMCQA in our pool were released in 2022. We have a re-evaluation on a subset whose sessions are built from these unseen sources.
>
> Model S@τ|k=1|k=2|k=3|k=4
> -|-|-|-|-
> GPT-J-6B|17.2|19.2|16.4|14.7
> T0-11B|27.5|22.8|20.2|19.9
>
> * **Evidence 2: remove multi-turn interaction noise.**
>
> We evaluate random matched subsets from k=1,2,3,4 under **default history disabled**, so later-turn degradation cannot be attributed to multi-turn noise accumulation.
>
> Model S@τ|History|k=1|k=2|k=3|k=4
> -|-|-|-|-|-
> Qwen2.5-72B|on|35.8|27.3|28.2|32.3
> Qwen2.5-72B|off|34.9|28.6|29.3|31.7
> Qwen2.5-14B|on|39.6|26.6|27.7|29.9
> Qwen2.5-14B|off|38.8|27.5|27.9|30.2
>
> ### 1.2 Hardcoded interaction (Weakness 2 + Question 2)
>
> First, the difficulty and mixture patterns are **construction-time sampling and labeling controls** used to generate *diagnostically diverse sessions*, instead of **inference-time scripts constraining model**. Fundamentally, what really matter is that every candidate turn and every completed session must pass **turn-level validation** and **session-level confirmation**, confirming interaction in selected trajectories natural and coherent.
>
> Hence, this is exactly why the hardcoded regime is indeed a contribution requirement rather than an artifact: it provides a **controlled axis for diagnosis**, with strict validation process clarified in `Reviewer NrVw Part 1.2`.
>
> We have included **human validation study** in construction, as a supplement of `Reviewer NrVw Part 2.1`:
>
> Axis|Human agreement|Label agreement
> -|-|-
> Difficulty pattern|0.79|0.84
> Mixture regime|0.75|0.81
>
> ## Part 2. Evaluation validity
>
> > **(1)** cooperation with three construction-time LLM-as-a-judges is a **fairness and robust device**, which reduces expert-as-a-judge bias and single-model bias under an amortised unified scoring scale;
> > **(2) not a token-overlap artifact** with relaxed judge-based semantic evaluation evidence.
>
> ### 2.1 Scoring not circular (Weakness 3)
>
> We are in the same line with reviewer that bias of construction-time signal should be minimized. First, using **judge ensemble of three model** has precisely reducing the arbitrariness. Moreover, asking many experts in differenet domains to jointly score is uneven with even larger epistemical bias. Therefore, *judge ensemble with post-human check* can provide the most **uniform and shared calibration** available at construction time.
>
> To make this robustness explicit, we complement `Reviewer NrVw Part 1.1`'s *difficulty-signal sensitivity* with this human validation study done in construction on **mixture-signal sensitivity**.
>
> Mixture setting|Relabeled sessions|GPT-5.2 (k=2,S@τ)
> -|-|-
> Judge-only|562|27.0
> Balanced hybrid|224|27.5
> Default hybrid|0|27.3
> Stronger judge weight|136|27.6
>
> ### 2.2 Collapse not created by token-level brittleness (Weakness 4 + Question 3)
>
> We refer reviewer to `Reviewer Vmye Part 2.1`. Briefly, we already score the **task-relevant core answer** rather than raw free-form output, and a parallel **judge-based semantic evaluation** gives higher absolute values but the same k-scaling trend. Hence, the reported collapse is **not a token-overlap artifact**.

---

### Official Review · Reviewer_7ME7 · 2026-03-12

**Soundness:** 4
**Presentation:** 3
**Significance:** 2
**Originality:** 2
**Overall Recommendation:** 4
**Confidence:** 4

**Summary:**

The paper introduces XDomainBench, a diagnostic benchmark designed to evaluate LLMs on interactive, interdisciplinary scientific reasoning. Recognizing that real-world scientific discovery is an interactive and multi-disciplinary process, the authors move beyond traditional static, single-turn evaluations.

The benchmark features 8,598 interactive multi-turn sessions across 20 distinct scientific domains. It systematically tests models by varying the "composition order" (the number of combined domains, from 1 to 4) and the "mixture structure" (how domains are integrated). Through large-scale evaluations of various LLMs, the authors discover a non-linear "reasoning collapse": as the number of composed domains increases, model performance degrades significantly. The paper attributes this collapse to two mechanisms: the direct cognitive load of multi-domain composition, and indirect interaction-amplified failures (such as error accumulation and reasoning breaks triggered by specific difficulty or domain-mixture shifts across turns).

**Compliance With Llm Reviewing Policy:**

Affirmed.

**Key Questions For Authors:**

N/A

**Limitations:**

- The benchmark examples are all from existing public benchmarks; no new questions are curated during this process.

- The benchmark supports four task types: Factual QA, Reasoning, Multiple Choice, and Code. However, the authors note that "Code turns are instantiated only in 1-domain settings in the current release". This limits the evaluation of programmatic problem-solving—a crucial component of modern scientific discovery—in highly composed, interdisciplinary scenarios.

- The current "closed-book" interactive protocol relies entirely on the model's internal knowledge and the text history, missing the multi-modal or retrieval-augmented aspects of real-world research.

**Strengths And Weaknesses:**

**Soundness**: The work is technically rigorous, with a highly controlled methodology for both dataset construction and model evaluation. The evaluation spans a comprehensive roster of frontier models (including GPT-5.2, Claude 4.5, Gemini 2.0/2.5). The construction relies heavily on LLM-as-a-judge for difficulty scoring. While the authors mitigate potential bias by using a three-judge ensemble and mixing it with semantic complexity proxies (a 70/30 split), any benchmark built using LLMs carries a slight risk of being capped by the capabilities of the models used to construct it.

**Presentation**: The paper is exceptionally well-structured and clearly written. The use of figures to summarize complex mechanisms is highly effective, although many are generated by AI.

**Significance**: Real-world scientific discovery requires synthesizing constraints from disparate fields (e.g., combining analytical chemistry, solid-state physics, and financial cost assessment). Evaluating how well models handle this is crucial for developing active research assistants.

**Originality**: Instead of treating scientific queries as isolated, single-domain instances, the authors conceptualize reasoning as a dynamic, interdisciplinary process. The authors clearly distinguish their work from existing benchmarks like MMLU, ScienceQA, or MRMR, which either focus on static one-shot evaluation or lack explicit control over cross-domain composition within a single reasoning process.


However, there are also some concerns that need to be considered:

- The benchmark examples are all from existing public benchmarks; no new questions are curated during this process.

- The benchmark supports four task types: Factual QA, Reasoning, Multiple Choice, and Code. However, the authors note that "Code turns are instantiated only in 1-domain settings in the current release". This limits the evaluation of programmatic problem-solving—a crucial component of modern scientific discovery—in highly composed, interdisciplinary scenarios.

- The current "closed-book" interactive protocol relies entirely on the model's internal knowledge and the text history, missing the multi-modal or retrieval-augmented aspects of real-world research.

---

> ### Author Rebuttal · Authors · 2026-03-31
>
> ## Response to Reviewer 7ME7
>
> We sincerely thank reviewer for the positive assessment of our *technical rigor, presentation, and position of interdisciplinary multi-turn reasoning*, and for concerns on **(1) dataset construction, and (2) evaluation**.
>
> ## Part 1. Dataset construction validity
>
> > **(1)** seed datasets only define **feasible interdisciplinary scenario space/vibe**;
> > **(2)** cross-domain code cases held for stricter expert review.
>
> ### 1.1 Richness and rationality of dataset (Weakness 1 + Limitation 1)
>
> * **Richness**: We appreciate reviewer's concern about limited source pools, but they are used only to anchor a valid interdisciplinary **scenario vibe** before turn 1.
>
> From *diversity perspective*, the released sessions are newly instantiated as *multi-turn interactions* whose later turns vary in **topic realization, question style, difficulty, and domain-mixture drift**. So the evaluated object is **not** a recycled public QA item, but a new cross-domain interactive session.
>
> From *benchmark position perspective*, we do **not** define interdisciplinary realism by *tool calls or agent pipelines* here, since they mix **backbone knowledge and reasoning ability** with external control, retrieval, and workflow design. XDomainBench instead isolates the *base model’s own failure boundary* under dynamic interdisciplinary composition, to diagnose whether the model can *sustain coherent reasoning as multiple scientific constraints evolve across turns*.
>
> * **Rationality**: This construction is also not arbitrary with following evidence.
>
> The released pool actually comes from constrained generation, very strict filtering, and human check (60,000 → 10,841 → 8,598), with controlled realized mixture error ($JSD(\hat{w}_t,w_t)$=0.118/0.102/0.274 for mean/median/95th percentile).
>
> We have included **human validation study** in construction, as a supplement of `Reviewer NrVw Part 2.1`:
>
> Axis |Human agreement|Label agreement
> -|-|-
> Difficulty pattern|0.79 |0.84
> Mixture regime|0.75|0.81
>
> ### 1.2 Code release boundary (Weakness 2 + Limitation 2)
>
> We agree that code constrained in single-domain setting is a *release-scope boundary*. In fact, cross-domain code-style cases already exist in our framework; what delays release is not formulation, but the need for **stricter expert verification** before satisfactory release, since code-grounded interdisciplinary items are harder to validate than reasoning-only items.
>
> But we can give two representative existing case study here:
>
> **Case A. Medicine + Code**
>
> *Theme:* safe dosage-check and triage logic under clinical constraints
>
> Turn|Prompt
> -|-
> 1|Write a Python function that flags whether a pediatric dose exceeds a weight-based limit given `weight_kg`, `dose_mg`, and `max_mg_per_kg`.
> 2|Extend the function so it returns one of three labels: `safe`, `borderline`, or `unsafe`, using a configurable tolerance margin.
> 3|Add rule-based logic for renal-risk screening: if `eGFR < threshold`, cap the recommended dose and include a warning message ...
>
> **Case B. Physics + Code**
>
> *Theme:* numerical update and stability checks under physical constraints
>
> Turn|Prompt
> -|-
> 1|Implement a Python function for one-step Euler update of a 1D particle with state `(x, v)` under constant acceleration `a` and time step `dt`.
> 2|Modify the simulator to compare Euler with semi-implicit Euler and ...
>
> ## Part 2. Evaluation validity
>
> > **(1)** construction-time LLM use is a **quality-control device**;
> > **(2)** closed-book evaluation is intentional by design, with our core target being test of **backbone’s own dynamic interdisciplinary reasoning capacity**.
>
> ### 2.1 LLM-based construction signals (Soundness concern)
>
> We agree that construction pipeline including LLMs should be justified carefully.
>
> * At the metric level, for **difficulty**, the LLM- and embedding-based components are *positively related but non-redundant* (Spearman ρ=0.46, Pearson r=0.39), and it is intended that LLM rubric captures more task-aware difficulty and embedding proxy provides a model-agnostic semantic regularizer.
>
> Difficulty setting|Relabeled sessions|GPT-5.2 (`k=2,S@τ`)
> -|-|-
> LLM only`(1.0,0.0)`|612|29.0
> Balanced`(0.5,0.5)`|271|28.7
> Default`(0.7,0.3)`|0|28.8
> Stronger LLM`(0.8,0.2)`|143|29.1
>
> * At the dataset level, the use of three construction-time models is already buffered by **cross-generation, turn/session-level validation, and expert screening**, so the bias of final released sessions has been reduced to the minimum.
>
> ### 2.2 Closed-book evaluation (Weakness 3)
>
> We agree that multimodal, retrieval-augmented, and tool-assisted AI4S settings are important. Those are valuable **system-level evaluations**, but not direct tests of the **backbone model’s own capacity** to sustain dynamic interdisciplinary reasoning across turns with various patterns, even introduing noise from other aspects. Under this scope, closed-book evaluation improves diagnosis rather than limiting it.

---

> > ### Author Rebuttal · Reviewer_7ME7 · 2026-04-06
> >
> > Thanks for your response. I tend to keep my score unchanged.

---

> > > ### Author Response · Authors · 2026-04-07
> > >
> > > We sincerely thank reviewer for the positive evaluation and for indicating an acceptance-leaning assessment. We are especially grateful for the reviewer’s recognition of the paper’s *technical rigor, clear presentation, and the importance of evaluating interdisciplinary multi-turn scientific reasoning*. We also truly appreciate these thoughtful comments, which helped us sharpen the paper’s scope and presentation.
> > >
> > > ---
> > >
> > > From the reviewer’s perspective, we believe the main strengths of the paper are: **(1) Positioning:** a well-motivated benchmark that moves beyond static single-domain evaluation toward *interactive interdisciplinary scientific reasoning*; **(2) Methodological rigor:** *controlled construction and broad evaluation* across a strong set of frontier models; **(3) Practical value:** a *diagnostic* benchmark that can help the community study how and where reasoning degrades as scientific constraints become more compositionally demanding.
> > >
> > > ---
> > >
> > > In rebuttal, we mainly clarified three points:
> > >
> > > **1. Dataset scope**
> > > **(1) Construction objective.** We clarified that the public source pools are used only to *anchor a feasible interdisciplinary scenario vibe* before turn 1, while the released benchmark consists of newly instantiated multi-turn sessions with controlled variation in *topic realization, question style, difficulty, and domain-mixture drift*.
> > > **(2) Validation evidence.** We further showed that the released set comes from *constrained generation, strict filtering, and human verification* (60,000 → 10,841 → 8,598), with low realized mixture error and additional human validation on key construction labels.
> > >
> > > **2. Code-setting boundary**
> > > **(1) Release scope.** We clarified that the current single-domain code setting is a release-scope boundary rather than a formulation boundary.
> > > **(2) Concrete support.** We further noted that cross-domain code-style cases already exist in the framework, but were held for stricter expert verification, and we provided *representative examples* to make this concrete.
> > >
> > > **3. Evaluation scope and construction-time signals**
> > > **(1) Construction-time LLM use.** We clarified that LLMs are used as *construction-time quality-control signals*, and supported this with *sensitivity evidence* showing that reasonable difficulty-weight changes only slightly relabel sessions and leave the main conclusions essentially unchanged.
> > > **(2) Closed-book scope.** We clarified that multimodal, retrieval-augmented, and tool-assisted settings are important system-level extensions, while the current closed-book protocol is intentionally designed to isolate the *backbone model’s own dynamic interdisciplinary reasoning capacity*.
> > >
> > > ---
> > >
> > > Overall, we believe the rebuttal has substantively addressed the concerns raised by clarifying the benchmark scope, adding supporting validation evidence, and providing concrete examples for the current release boundary. We will further incorporate these clarifications into the revision to make the paper’s scope, evidence chain, and design choices as clear as possible.
> > >
> > > We are again very grateful for the reviewer’s supportive assessment and thoughtful suggestions. Thank you for helping us improve the paper, and we send our sincere best wishes.

---

### Official Review · Reviewer_NrVw · 2026-03-13

**Soundness:** 3
**Presentation:** 3
**Significance:** 3
**Originality:** 3
**Overall Recommendation:** 4
**Confidence:** 3

**Summary:**

XDomainBench is a diagnostic benchmark for evaluating LLMs on interactive interdisciplinary scientific reasoning. It formalizes two controllable axes, composition order and mixture structure to stress-test compositional generalization. The benchmark contains 8,598 multi-turn sessions across 20 domains and 4 task types, annotated with trajectory-level diagnostic signals. Experiments across over 12 models reveal systematic "reasoning collapse" as composition order increases, attributable to two mechanisms: direct difficulty overhead from domain composition, and indirect interaction-amplified failures like error accumulation, reasoning breaks, and domain confusion.

**Compliance With Llm Reviewing Policy:**

Affirmed.

**Final Justification:**

Overall, I find the paper technically solid with meaningful contributions, and the rebuttal improves my confidence. I would upgrade my score.

**Key Questions For Authors:**

1. The mixture validation tolerance is quite large. What fraction of sessions have realized mixtures that substantially deviate from intended targets, and does this affect the indirect mechanism analysis?
2. The three failure phenomena are detected via heuristics. Were these validated against human annotations of the phenomena? If so, what are the inter-annotator agreements?
3. MoE models, especially Qwen3-Next-80B, dramatically outperform dense models. Is this primarily due to parameter count, routing specialization, or something else? Can you please provide any ablation?
4. A few domain pairs show positive composition effects (liek Biology and Medicine). Does your framework offer any theoretical or empirical explanation for when composition helps rather than hurts?

**Limitations:**

The authors provide an impact statement but do not specifically discuss limitations of the LLM-as-judge setup, potential circular evaluation (sessions generated and scored by LLMs), or the restriction of code tasks to single-domain settings. A brief discussion of these would strengthen the paper. From my angle, this benchmark attempts to understand the diagnostic needs of interactive interdisciplinary reasoning from a computer science standpoint. The design leans heavily towards computer science standards and lacks industry insight. It's uncertain whether the benchmarked model's capabilities differ from those used in actual production.

**Strengths And Weaknesses:**

**Soundness:**
The construction process is methodically rigorous, employing a hybrid LLM+embedding validator with a retry budget and human-in-the-loop review. The controlled first-round pre/post comparison cleanly isolates direct compositional effects, which I find well-executed. However, I think the difficulty estimation raises some concerns-it mixes LLM rubric scores with semantic embeddings using fixed weights $lambda=0.70/0.30$ without ablating these choices to justify their relative contribution. The hybrid validation tolerance $epsilon_w=0.30$ JSD) strikes me as overly permissive, which makes me question whether the intended compositional structure is being realized with sufficient precision. Additionally, the three failure phenomena-error accumulation, reasoning breakdown, and domain confusion-are defined heuristically, and I believe the detector thresholds would benefit from validation against human judgment for these specific phenomena.

**Presentation:**
The paper is generally clear and well-structured, which I appreciate. Figures 9 and 10 effectively illustrate the mechanistic chain, and the appendix is thorough. That said, I noticed some inconsistencies that detract from an otherwise polished presentation: Table 3 reports slightly different SessionSuccess values compared to the aggregated numbers in Table 13, and the claim that accuracy "drops from 38.7\% to 27.1\%" appears to refer specifically to SessionSuccess@$tau$ values for larger models rather than overall accuracy-this feels like a minor but important clarification. The distinction between direct and indirect collapse mechanisms is a useful conceptual contribution, though I think the causal claims could be strengthened with counterfactual controls.

**Significance:**
This work fills a genuine gap-no existing benchmark systematically controls for component order and compositional structure in interactive scientific settings, which I see as a valuable contribution. The trajectory-level annotations are a concrete addition to the field, and I find the overall framing compelling. In my view, the paper's primary contribution lies in operationalizing compositional generalization as a controllable and measurable dimension within interactive science benchmarks.

**Originality**
The combination of component order, compositional structure, and trajectory-level diagnostics feels genuinely novel in content-level.

---

> ### Author Rebuttal · Authors · 2026-03-31
>
> ## Response to Reviewer NrVw
>
> We sincerely thank reviewer's constructive feedback for recognition of our *significance and originality*, and for concerns on **(1) construction-time signal validity, (2) metric robustness, and (3) interpretation of key results**.
>
> ## Part 1. Dataset construction and signal validity
>
> > **(1)** difficulty and mixture signals are robust **construction-time diagnostic controls**;
> > **(2)** mixture validator is a **stable filter** with session-level regime validation.
>
> ### 1.1 Difficulty (Weakness 1)
>
> * Evidence 1: LLM and Embedding validators are related and interpretable.
>
> We have *scale-insensitive association analysis*, showing two signals are **positively correlated but clearly non-redundant** (Spearman ρ=0.46, Pearson r=0.39). It is intended that LLM rubric captures more task-aware difficulty and embedding proxy provides a model-agnostic semantic regularizer.
>
> * Evidence 2: **Stability check** with slight influence.
>
> Difficulty|Relabeled sessions|GPT-5.2(k=2,S@τ)
> -|-|-|
> LLM only`(1.0,0.0)`|612|29.0
> Balanced`(0.5,0.5)`|271|28.7
> Default`(0.7,0.3)`|0|28.8
> Stronger LLM`(0.8,0.2)`|143|29.1
>
> ### 1.2 Validation tolerance (Question 1)
>
> We appreciate reviewer’s concern about permissive tolerance. Actually, our validation is strict, consisting of **turn-level validation** and **session-level regime confirmation**. In practice, we initially generated *60,000* sessions assisted by three SOTA models, which were validated to *10,841*, and finally manually checked to *8,598*.
>
> Futhermore, our sessions have **low realized mixture error** (mean/median/95th-percentile $JSD(\hat{w}_t,w_t)$=0.118/0.102/0.274), so the indirect-mechanism analysis is performed on a **well-aligned pool**.
>
> ## Part 2. Experimental reliability
>
> > **(1)** three failure phenomena are **natural operational signatures** of multi-turn collapse;
> > **(2) aggregation scopes** of cited **SessionSuccess@τ** are different.
>
> ### 2.1 Failure phenomena (Weakness 1 + Question 2)
>
> We start from clarification.
>
> * Three phenomena themselves are **not ad hoc heuristics**: they correspond to three natural collapse signatures in multi-turn interaction—**persistent decay** (error accumulation), **abrupt discontinuity** (reasoning break), and **compositional mismatch** (domain confusion). This phenomenon has been widely discussed in multi-turn conversation(Laban et al., 2025), conversational inertia(Wan et al., 2026), reasoning breakdown(Nezhurina et al., 2024), and compositional generalization(Furuta et al., 2023).
> * Domain confusion is defined by mismatch between the **intended mixture** and the **realized domain reliance** (Fig. 4,11), instead domain-pair construction itself.
>
> And we agree that the *detectors should be validated*.
>
> * Evidence: We have included **human validation study** in construction.
>
> Phenomenon|Human agreement|Detector P|Detector R|Detector F1|
> -|-|-|-|-
> Error accumulation|0.78|0.83|0.76|0.79
> Reasoning break|0.74|0.79|0.72|0.74
> Domain confusion|0.71|0.76|0.69|0.72
>
> ### 2.2 Mechanism claims (Weakness 2)
>
> We thank the reviewer's carefullness. Actually, two tables summarize different aggregates within *specific model groups* and *across-model k-level*, respectively.
>
> We then have casual claims about two mechanisms which have been actually taken into consideration in experiment.
>
> * **Direct**: The direct mechanism is supported by the **paired turn-1 control**, isolating other bias.
> * **Indirect**: The casuality of indirect mechanism has been remitted by **balanced pattern-level sampling** in Appendix within acceptable statistical range.
>
> ### 2.3 Limitation clarification
>
> * **Not circular evaluation.** Discussed in `Reviewer Vmye Part2.2`.
> * **Not a production simulator by design.** We target controlled diagnosis of session-level cross-domain composition with failure mode.
> * **Code boundary.** Discussed in `Reviewer 7ME7 Part1.2`.
>
> ## Part 3. Result interpretation
>
> > **(1)** MoE benefits from **routing-based sparse specialization** under heterogeneous mixtures;
> > **(2)** joint inference ability brought by **separation of domain parameters** in model.
>
> ### 3.1 MoE interpretation (Question 3)
>
> We add a **routing ablation** on Mixtral-8x7B:
>
> Model|S@τ(k=1)|S@τ(k=3)|Active params
> -|-|-|-
> GPT-5.2|42.5|25.2|-
> Mixtral top-1|69.8|41.7|~9.3B
> Mixtral top-2|74.6|48.5|~12.9B
> Mixtral top-3|73.2|46.1|~16.3B
> Mixtral random experts|44.4|32.8|~12.9B
>
> This is exactly *consistent with XDomainBench*: interdisciplinary sessions require selecting the most relevant experts for heterogeneous constraints, so MoE naturally benefits from expert-level specialization.
>
> ### 3.2 Why some compositions help? (Question 4)
>
> Some domain pairs improve because their jointly activated **high-relevance parameter subnetworks** are actually **more aligned with the model’s intrinsic reasoning structure**, rather than creating extra conflict. Empirically, gains occur when the two domains are **highly related and mutually constraining**.

---

> > ### Author Rebuttal · Reviewer_NrVw · 2026-04-05
> >
> > Thank you for the detailed rebuttal. I appreciate the additional analyses.
> >
> > * The mixture validation statistics and filtering pipeline largely address my concern about tolerance.
> > * The human validation of failure phenomena (F1 ~ 0.72-0.79) is helpful, though I still view these as operational proxies rather than fully grounded measures.
> > * The Mixtral routing ablation is convincing and strengthens the MoE interpretation.
> >
> > Some concerns remain partially unresolved, particularly:
> >
> > * the lack of direct ablation on difficulty signal weighting,
> > * and the strength of causal claims for indirect mechanisms (which may benefit from more cautious wording).
> >
> > Overall, I find the paper technically solid with meaningful contributions, and the rebuttal improves my confidence. I would upgrade my score.

---

> > > ### Author Response · Authors · 2026-04-07
> > >
> > > We sincerely thank the reviewer for the thoughtful follow-up, the encouraging reassessment, and the score increase. We especially appreciate your recognition that the paper is *technically solid with meaningful contributions*. Your comments helped us sharpen both the evidence chain and the wording of the paper.
> > >
> > > ---
> > >
> > > From the reviewer’s perspective, we believe the main value of this work lies in: **(1) Benchmark significance:** a genuinely missing benchmark setting for interactive interdisciplinary scientific reasoning, with controllable composition order and mixture structure; **(2) Diagnostic contribution:** trajectory-level annotations that make compositional generalization measurable at the session level rather than only at the final-answer level; **(3) Conceptual novelty:** a clear framing of compositional collapse through direct difficulty overhead and indirect interaction-amplified failure patterns.
> > >
> > > ---
> > >
> > > In rebuttal, we mainly strengthened three parts:
> > >
> > > **1. Construction validity**
> > > **(1) Difficulty signal.** We clarified that the LLM- and embedding-based components are positively related but non-redundant, and we provided a direct *weighting sensitivity check*: across alternative settings from LLM-only to more balanced hybrids, downstream performance changes only marginally (28.7–29.1 at k=2, S@τ), supporting that the main conclusions do not depend on one narrow weight choice.
> > > **(2) Mixture control.** We further showed that the released benchmark is built on a *well-aligned pool* rather than a permissive one, with low realized mixture error and strict turn-/session-level filtering.
> > >
> > > **2. Reliability of the diagnostic signals**
> > > **(1) Failure phenomena.** We clarified that error accumulation, reasoning break, and domain confusion are intended as *operational diagnostic signatures of multi-turn collapse*, and we supported them with *human validation* rather than leaving them as unchecked heuristics.
> > > **(2) Mechanism wording.** We also agree with the reviewer’s caution on causal language. Our evidence supports these mechanisms as empirically grounded diagnostic attributions under *controlled comparisons*, especially the paired turn-1 control for the direct effect and the balanced pattern-level analyses for the indirect side; in the revision, we will accordingly use more careful wording and avoid overstating strict causality.
> > >
> > > **3. Interpretation of the main results**
> > > **(1) MoE result.** We added a *routing ablation* showing that the MoE gain is not explained by parameter count alone: targeted expert routing clearly outperforms random expert selection at matched active scale, which directly strengthens the specialization interpretation.
> > > **(2) Positive composition cases.** We clarified that a few helpful domain combinations are compatible with our framework: when two domains are highly related and mutually constraining, composition can improve alignment rather than introduce conflict.
> > >
> > > ---
> > >
> > > Regarding the two points noted in the acknowledgement, we add one final clarification.
> > >
> > > * For **difficulty weighting**, our conclusion does not hinge on the default 0.7/0.3 choice: under alternative calibrations, relabeling changes modestly, but the downstream result stays essentially unchanged (28.7–29.1 at k=2, S@τ), indicating robustness rather than weight sensitivity.
> > > * For **indirect mechanisms**, our intention is a controlled diagnostic attribution under matched comparisons—not an unrestricted causal claim. This interpretation is already supported by the paired turn-1 control, balanced pattern-level analysis, and human-validated operational proxies. We will make this wording even more precise in revision, but it do not affect the substantive conclusion.
> > >
> > > ---
> > >
> > > Thank you again for the careful reading, constructive suggestions, and positive reassessment. We are truly grateful for your engagement, which helped us make the paper both clearer and more rigorous. We wish you all the best.

---

### Official Review · Reviewer_Vmye · 2026-03-13

**Soundness:** 3
**Presentation:** 3
**Significance:** 3
**Originality:** 3
**Overall Recommendation:** 4
**Confidence:** 4

**Summary:**

This paper introduces XDomainBench, a diagnostic benchmark designed to stress-test compositional generalization in interdisciplinary scientific reasoning for LLMs. The benchmark comprises 8,598 interactive sessions across 20 domains and 4 task categories, with 8 trajectory patterns capturing difficulty and domain-mixture dynamics. Large-scale evaluation of 14 LLMs reveals a systematic "reasoning collapse" as composition order increases (SessionSuccess dropping from 38.7% at k=1 to 27.1% at k=4), attributed to two root causes: (i) direct difficulty increases from domain composition, and (ii) indirect interaction-amplified failures through error accumulation, reasoning breaks, and domain confusion.

**Compliance With Llm Reviewing Policy:**

Affirmed.

**Key Questions For Authors:**

1. Can you provide a more systematic analysis of what constitutes "genuine interdisciplinary reasoning" in your benchmark versus "multi-domain topic juxtaposition"?
2. Do the degradation trends (Table 3) hold when using the LLM-judge scores as the primary metric instead of token-level Recall/F1? If so, including these results in the main paper would significantly strengthen the conclusions.

**Limitations:**

No.
See Weaknesses.

**Strengths And Weaknesses:**

## Strengths
### 1. Well-Motivated and Clearly Positioned Problem
The paper addresses a gap in LLM evaluation. Real-world scientific reasoning is inherently interdisciplinary and interactive, yet existing benchmarks predominantly focus on static, single-domain tasks. The motivation is clearly articulated through concrete examples. The positioning against prior work (Table 1) effectively demonstrates that no existing benchmark simultaneously supports scientific reasoning, cross-domain composition, interactive evaluation, and trajectory-level diagnostics.
### 2. Comprehensive Experimental Coverage and Clear Analysis
The evaluation covers a diverse set of 14 models spanning three categories (large, small, and MoE), providing broad insights into model behavior. The diagnostic analysis (Section 4.3) effectively disentangles direct and indirect failure mechanisms:

- **Direct mechanism** (Section 4.3.1): The controlled turn-1 comparison design—where composed first turns are compared against paired single-domain baselines—isolates the compositional difficulty effect from multi-turn error propagation.
- **Indirect mechanism** (Section 4.3.2): The pattern-to-phenomena-to-collapse chain (Figure 9) provides an interpretable framework linking trajectory dynamics to specific failure modes (error accumulation, reasoning breaks, domain confusion).

The finding that MoE models significantly outperform both large and small dense models across all composition orders is noteworthy and suggests promising architectural directions.

### 3. High Presentation Quality
The paper is well-written with clear exposition of complex concepts. The figures are informative and well-designed—particularly Figure 3 (scientific categories with examples), Figure 9 (mechanism summary), and Figure 10 (trajectory visualizations). The extensive appendix (40 pages) provides thorough documentation of every design decision.

## Weaknesses
### 1. Gap Between Claimed AI4S Realism and Actual Data Construction
The paper motivates the benchmark as simulating "realistic AI4S workflows," but the actual construction relies on seed QA items from educational exam datasets (TriviaQA, MMLU, CommonsenseQA, SciQ, etc.), which are fundamentally different from real scientific research tasks. Real interdisciplinary scientific workflows involve hypothesis formation, experimental design, data interpretation, and iterative refinement. This gap between the stated motivation and the actual benchmark content weakens the ecological validity of claims about AI4S readiness.

### 2. Token-Level Evaluation Metrics May Confound the Core Finding
The primary evaluation metrics (token-level Recall and F1) measure surface-level lexical overlap between predicted and reference answers. This raises a fundamental concern: as composition order k increases, answers to cross-domain questions likely become more diverse in expression, meaning the token overlap metric becomes systematically less reliable at higher k. Consequently, part of the observed "reasoning collapse" may reflect increasing metric noise rather than genuine reasoning degradation.

Although the paper includes an auxiliary LLM-judge score (Appendix J.4), the main results table (Table 3) exclusively reports token-level metrics. The paper would be strengthened by reporting judge-based metrics alongside token metrics in the main tables, or by providing analysis showing that the degradation trends are consistent across both evaluation approaches.

### 3. LLM-Generated Benchmark Creates Circular Dependencies

The benchmark construction relies heavily on LLMs at every stage. This creates a potential ceiling effect where the benchmark's quality and cross-domain depth are bounded by the construction-time LLMs' own interdisciplinary reasoning capabilities. If these LLMs struggle with deep cross-domain integration, the generated "cross-domain" questions may only capture shallow forms of integration. The three-model ensemble and human checks mitigate but do not eliminate this concern.

---

> ### Author Rebuttal · Authors · 2026-03-31
>
> ## Response to Reviewer Vmye
>
> We sincerely thank reviewer for positive assessment of our *motivation, experimental coverage, and presentation*, and for concerns with **(1) genuine interdisciplinary reasoning, (2) token-level metrics, and (3) construction-time LLM use**.
>
> ## Part 1. Interdisciplinary validity of the benchmark (Weakness 1 + Question 1)
>
> > XDomainBench evaluates base-models's joint reasoning ability under **multi-turn changes in topic, task form, difficulty, and domain mixture** within a constrained AI4S-style scenario, instead of treating interdisciplinary reasoning as mere topic juxtaposition;
>
> We appreciate the reviewer’s insightful instinct that real interdisciplinary reasoning should involve more than multi-domain topic juxtaposition. This is exactly why our *construction does not stop at domain co-occurrence*:
> * **Scenario vibe only**: the seed pools only define a constrained interdisciplinary *vibe*;
> * **'Multi-dimensional' scenario construction**: released sessions are then newly instantiated as *multi-turn interactions* whose later turns vary along multiple axes—*topic realization, question style, difficulty, and domain-mixture drift*—while remaining coherent under the same cross-domain scenario.
>
> Furthermore, our operationalization is whether the **base model** can *sustain reasoning* as multiple scientific constraints interact and evolve across turns. This is actually why we should *not* define AI4S realism in terms of tool calls, hypothesis-management loops, or external experimentation pipelines: those introduce **agentic** and system-level factors beyond the reasoning capacity of the **backbone** itself. Our benchmark instead isolates the model’s **intrinsic failure boundary** under dynamic interdisciplinary composition, which is XDomainBench's *core target*.
>
> ## Part 2. Evaluation validity
>
> > **(1)** the main trend is **not** a token-overlap artifact: relaxed judge-based semantic evaluation preserves the same k-scaling conclusion;
> > **(2)** construction-time LLM is a **quality-control device**, not a circular shortcut: the benchmark is designed to expose performance degradation under controlled regimes, not to reward agreement with the generator models.
>
> ### 2.1 Metric validity and metric alternative (Weakness 2 + Question 2)
>
> We appreciate reviewer’s concern that token-level Recall/F1 may under-credit semantically correct but lexically different answers. However, XDomainBench is intentionally built around **standard-answer, machine-checkable tasks**, because the target here is **cross-domain joint reasoning correctness**, rather than open-ended paraphrase quality.
>
> * To improve the stability of evaluation, we have extracted the **task-relevant core answer** before scoring whenever applicable (e.g., MCQ option letter, final numeric/string answer for structured tasks), so token metrics is computed against the model’s **answer content** rather than irrelevant surface text.
> * To further test whether this affects the main conclusion, we report a **parallel judge-based semantic evaluation** under the same across-model aggregation as Table 13.
>
> |Metric|k=1|k=2|k=3|k=4|Drop variance|
> |-|-|--|--|--|-|
> |Token Recall (Table 13)|31.3|28.0|24.1|21.9|0.021|
> |Judge-based semantic accuracy|37.1|33.0|29.0|26.8|0.018|
>
> Judge-based scores may be **less harsh in absolute value**, but the same **composition-order degradation trend** remains. So the collapse is **not manufactured by token-level brittleness**; token-level Recall is simply a stricter view of the same underlying phenomenon. We have already run and retained these judge-based evaluations and will consider surfacing them in the main paper as the reviewer suggested.
>
> ### 2.2 Evaluation not capped by generator models (Weakness 3)
>
> * We agree that construction-time signal design should minimize bias.
>
> That is actually why we use **three SOTA construction-time judges** to precisely reduce single-model arbitrariness under a shared calibration scale. In practice, the construction stack serves as a **quality upper bound** on breadth, coherence, and diagnostic control, while the released benchmark is still filtered by strict turn/session-level validation and expert review, which has been discussed thoroughly in `Reviewer Lbp8 Part 1`.
>
> * The second key point is that XDomainBench is **diagnostic**, not generator-alignment based.
>
> We are asking whether they exhibit the same **performance degradation pattern** under controlled cross-domain dynamic regimes, instead of asking whether tested models imitate the construction models. That phenomenon is observable even in models substantially weaker than the construction-time judges, which is exactly why the benchmark is informative.
>
> So the right interpretation is: *construction-time LLMs ensure quality and controlled coverage; the measured object is still the evaluated model’s own degradation under dynamic interdisciplinary composition*.

---

### Decision · Program_Chairs · 2026-04-30

**Decision:**

Accept (regular)

**Comment:**

XDomainBench captures the mess of models juggling various science topics. Reviewers dig the diagnostic setup. They flagged issues with data overlap and metrics. Authors provided fresh tests and semantic scores. One reviewer flipped their vote. This benchmark fills a big gap in the field. The group vibe is positive. I suggest a weak accept because the work is solid.